# Maximizing the Potential of Synthetic Data: Insights from Random Matrix Theory

**Aymane El Firdoussi**[*,1]**, Mohamed El Amine Seddik**[*,1]**, Soufiane Hayou**[2]**, Reda Alami**[1]**, Ahmed Alzubaidi**[1] **& Hakim Hacid**[1]
[1]Technology Innovation Institute, Abu Dhabi, UAE
[2]Simons Institute, Berkeley, USA
[*]Equal contribution
`firstname.lastname@tii.ae`

## Abstract

Synthetic data has gained attention for training large language models, but poor-quality data can harm performance (see, e.g., Shumailov et al. (2023); Seddik et al. (2024)). A potential solution is data pruning, which retains only high-quality data based on a score function (human or machine feedback). Previous work Feng et al. (2024) analyzed models trained on synthetic data as sample size increases.

Using random matrix theory, we generalize this analysis and derive the performance of a binary classifier trained on a mix of real and pruned synthetic data in a high dimensional setting. Our findings identify conditions where synthetic data could improve performance, focusing on the quality of the generative model and verification strategy. We also show a smooth phase transition in synthetic label noise, contrasting with prior works on sharp transition in infinite sample limits. Our extensive experimental setup validates our theoretical results.

## 1 Introduction

The landscape of large language models (LLMs) is evolving rapidly, with a growing trend towards training models on a combination of real and synthetic data. This synthetic data is often generated by previously trained models (Allal et al., 2024; Ben Allal et al., 2024; Abdin et al., 2024). However, the quality of these generators can significantly impact the performance of newly trained models, potentially leading to model collapse (Shumailov et al., 2023), a phenomenon in which the model drastically degrades in performance.

Model collapse has been extensively studied, both empirically (Guo et al., 2023) and theoretically (Seddik et al., 2024), highlighting the potential risks associated with training on synthetic data. To mitigate these risks, researchers have proposed various strategies, including the verification of AI-synthesized data (Feng et al., 2024). This approach aligns with the widely adopted Reinforcement Learning from Human Feedback (RLHF) technique (Kaufmann et al., 2023). Feng et al. (2024) provided theoretical support for this strategy by analyzing synthetic data as Gaussian mixtures with noisy labels, using linear binary classifiers and scalar parameters to control verifier quality. Their findings reveal a sharp performance transition: *under infinite synthetic sample size conditions, model accuracy shifts from zero accuracy (due to errors in synthetic data and verification) to optimal performance as these errors decrease.*

While current theoretical studies primarily focus on label noise in synthetic data (Dohmatob et al., 2024a; Gerstgrasser et al., 2024; Feng et al., 2024), they often overlook potential distribution shifts in the feature space between real and synthetic data. This gap is particularly relevant in practical scenarios where generative models are trained on finite real data sets, potentially leading to imperfect learning of the underlying distribution.

Our paper addresses this gap by proposing a statistical model that accounts for both distribution shifts in the feature space and label noise. In our model, we induce distribution shifts in the feature space by supposing that the statistics of synthetic data are empirical estimates of the underlying real data statistics. In a finite sample size regime, these estimates may be biased, resulting in distribution

shifts between real and synthetic data. Leveraging random matrix theory, we derive the theoretical performance of a binary classifier trained on a mixture of real and pruned (i.e., verified) synthetic data in a high-dimensional setting. Our analysis provides conditions under which synthetic data improves performance, emphasizing the critical roles of the generative model's quality and the efficacy of the synthetic data verification strategy. Lastly, we show that the sharp phase transition phenomenon identified in (Feng et al., 2024) in the infinite sample size limit is a particular case of a general result, where smooth phase transitions can take place.

**Summary of contributions.** Our contributions are four fold:

- We introduce a statistical model for studying synthetic data that accounts for label and feature noise, extending beyond previous models that only consider label noise.
- By leveraging random matrix theory, we characterize the performance of a binary classifier trained on a mixture of real and synthetic data in a high-dimensional setting.
- When training only on synthetic data, we find a smooth phase transition in classifier performance, generalizing the work of Feng et al. (2024) on sharp transitions in infinite sample size limit.
- We validate our results with extensive experiments (toy example and realistic LLM setups).

## 2 RELATED WORK

The use of synthesized data for model training has gained significant traction in recent years, particularly with the widespread adoption of large language models (LLMs) that rely on large amounts of data in their training stages. Several studies have explored the impact of synthesized data on model performance, revealing both its advantages and limitations. A primary concern is the phenomenon of model collapse Shumailov et al. (2023), where the iterative use of generated data for model training results in a degradation of model quality. This issue has been explored theoretically and empirically across multiple studies (e.g. LeBrun et al. (2021); Alemohammad et al. (2023); Bohacek & Farid (2023); Bertrand et al. (2023); Jain et al. (2024); Seddik et al. (2024); Dohmatob et al. (2024a;b;c)).

Seddik et al. (2024) investigated model collapse in recursive training settings, where new models are trained on data generated by previous models. They demonstrate that recursive training on purely synthetic data inevitably leads to performance degradation. However, they show that mixing real and synthetic data can attenuate model collapse, though the proportion of real data must remain high to maintain model performance. Their findings support the idea that synthesized data alone cannot sustain model quality across iterations without a significant quantity of real data.

Gerstgrasser et al. (2024) argue that model collapse can be avoided entirely if data is accumulated rather than replaced across iterations. Their empirical studies on language models, diffusion models, and variational autoencoders indicate that accumulating both real and synthetic data helps maintain model performance over time, breaking the recursive degradation loop that leads to collapse. Additionally, Jain et al. (2024) introduced a weighted empirical risk minimization (ERM) approach to better integrate synthetic data to the training pipeline, leading to a significant reduce in the test risk.

The most relevant work to our study is Feng et al. (2024) where the authors examined the effects of synthesized data on model performance in a non-recursive setting, using the concept of reinforcement through feedback to select high quality synthetic data. Their theoretical results, based on Gaussian mixture models, showed that adding feedback significantly improves the robustness of models trained on synthesized data. However, their setup assumes that only labels, not features, are noisy. Additionally, their focus is primarily on scenarios where only the number of data points, $n$, grows to infinity. Other (practical) scenarios where for instance the feature dimension, $p$, grows at a fixed ratio with $n$ are not covered.

Our work extends the Gaussian mixture model setup to include both noisy features and labels, which is a more realistic scenario when training on synthesized data. Additionally, we consider a high-dimensional regime where both $p$ and $n$ grow to infinity with a fixed ratio, a setup often used in Random Matrix Theory (RMT). This allows us to study the interaction between feature dimension, pruner error, and data size in a more comprehensive manner. Our approach also accounts for the presence of mixed data—original and synthetic—providing a more realistic framework for studying the effect of synthetic data in practical applications.

## 3 THEORETICAL SETUP

**Real data.** We suppose that real data consists of $n$ $p$-dimensional i.i.d. vectors $\boldsymbol{x}_1, \ldots, \boldsymbol{x}_n \in \mathbb{R}^p$ sampled from a Gaussian mixture of two distinct isotropic clusters $\mathcal{C}_1$ and $\mathcal{C}_2$ of means $\pm\boldsymbol{\mu}$ with $\boldsymbol{\mu} \in \mathbb{R}^p$. Essentially, for $a \in \{1, 2\}$, each data vector $\boldsymbol{x}_i \in \mathcal{C}_a$ has a corresponding label $y_i = (-1)^a$ and is sampled as:

$$\boldsymbol{x}_i = y_i\boldsymbol{\mu} + \boldsymbol{z}_i, \quad \boldsymbol{z}_i \sim \mathcal{N}(\mathbf{0}, \mathbf{I}_p). \tag{1}$$

**Generative model.** To generate synthetic data, we consider the generative model corresponding to maximum likelihood which consists of estimating the underlying first and second-order statistics of the real data with their empirical estimates. In particular, we suppose that we are given a subset $\hat{n} \leq n$ of the real dataset $(\boldsymbol{x}_i, y_i)_{i=1}^n$ on which we can estimate the statistics. This setup allows us to model a situation where new real data samples might be available to train next-generation models and the parameter $\hat{n}$ offers control over the generative model quality. The statistics for generating synthetic data are therefore computed using the following estimates

$$\hat{\boldsymbol{\mu}} = \frac{1}{\hat{n}} \sum_{i=1}^{\hat{n}} y_i\boldsymbol{x}_i, \quad \hat{\mathbf{C}} = \frac{1}{\hat{n}} \sum_{i=1}^{\hat{n}} (y_i\boldsymbol{x}_i - \hat{\boldsymbol{\mu}}) (y_i\boldsymbol{x}_i - \hat{\boldsymbol{\mu}})^\top. \tag{2}$$

**Synthetic data.** We consider that synthetic data is generated as $m$ i.i.d. vectors $\tilde{\boldsymbol{x}}_1, \ldots, \tilde{\boldsymbol{x}}_m \in \mathbb{R}^p$ with corresponding (noisy) labels $\tilde{y}_1, \ldots, \tilde{y}_m = \pm 1$ such that $\tilde{\boldsymbol{x}}_i \in \tilde{\mathcal{C}}_a$ with true label $\bar{y}_i = (-1)^a$ for $a \in \{1, 2\}$ is sampled as ($\tilde{\mathcal{C}}_1$ and $\tilde{\mathcal{C}}_2$ denote the synthetic clusters)

$$\tilde{\boldsymbol{x}}_i = \bar{y}_i\hat{\boldsymbol{\mu}} + \hat{\mathbf{C}}^{\frac{1}{2}}\tilde{\boldsymbol{z}}_i, \quad \tilde{\boldsymbol{z}}_i \sim \mathcal{N}(\mathbf{0}, \mathbf{I}_p), \tag{3}$$

and the labels $\tilde{y}_i$ are generated such that $\mathbb{P}\{\tilde{y}_i = \bar{y}_i\} = 1 - \varepsilon$ where $\varepsilon \geq 0$ controls label noise. Essentially, the **quality** of synthetic data depends on the **sample size $\hat{n}$ and the label noise rate** $\varepsilon$.

In the asymptotic regime where $\hat{n} \to \infty$ with $\frac{p}{\hat{n}} \to 0$, we can generate synthetic samples that follow asymptotically the exact same distribution as of the real ones, and therefore only label noise is relevant to the quality of the synthetic data. However, in the regime when both $\hat{n}, p \to \infty$ with $\frac{p}{\hat{n}} \to \hat{\eta} > 0$, while the estimation of $\boldsymbol{\mu}$ with $\hat{\boldsymbol{\mu}}$ remains consistent, the estimation of the covariance is not. In fact, in this regime $\|\hat{\mathbf{C}} - \mathbf{I}_p\| \not\to 0$ and the eigenvalues of $\hat{\mathbf{C}}$ spread in the vicinity of 1 which is described in the limit by the Marchenko-Pastur law (Marchenko & Pastur, 1967) as depicted in Fig. 1. Eventually, such inconsistency in estimating the second moment in high dimensions yields a distribution shift between synthetic and real data, which might cause a drop in performance when training a new model on synthetic data generated with $\hat{\boldsymbol{\mu}}$ and $\hat{\mathbf{C}}$. In the remainder, we describe precisely how the performance of a simple classifier is affected in this regime.

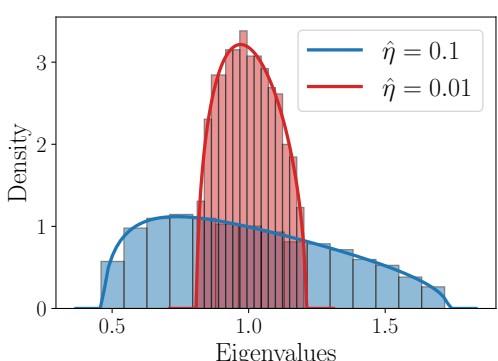

Figure 1: Illustration of the Marchenko-Pastur law: The histogram of eigenvalues of the empirical covariance matrix $\hat{\mathbf{C}}$ (as per equation (2)) using different values of $\hat{n}$. The histograms correspond to $p = 500$ and $\hat{n} = 5 \times 10^3$ (in blue) and $\hat{n} = 5 \times 10^4$ (in red). The line plots depict the limiting Marchenko-Pastur law. As $\hat{n}$ grows, the distribution of eigenvalues shrinks towards 1.

**Objective.** Our goal throughout the paper is to study the effect of synthetic data when training on a mixture of the $n$ real and $m$ synthetic data described above, i.e., with the following objective function:

$$\mathcal{L}(\boldsymbol{w}) := \underbrace{\frac{1}{n+m} \sum_{i=1}^{n} \ell(\boldsymbol{x}_i, y_i; \boldsymbol{w})}_{\text{real data}} + \underbrace{\frac{1}{n+m} \sum_{i=1}^{m} q_i \ell(\tilde{\boldsymbol{x}}_i, \tilde{y}_i; \boldsymbol{w})}_{\text{synthetic data}}, \tag{4}$$

where $\ell$ is some convex loss function and the $q_i$'s are data pruning parameters ($q_i \in \{0, 1\}$), indicating whether to select or drop the $i^{th}$ synthetic sample ($\tilde{\boldsymbol{x}}_i, \tilde{y}_i$). In particular, the $q_i$'s are Bernoulli random variables conditionally on $\tilde{y}_i \neq \bar{y}_i$ or $\tilde{y}_i = \bar{y}_i$ (we recall that $\bar{y}_i$'s denote the true labels of the synthetic samples) with conditional probabilities

$$\rho := \mathbb{P}\{q_i = 1 \mid \tilde{y}_i \neq \bar{y}_i\}, \quad \phi := \mathbb{P}\{q_i = 1 \mid \tilde{y}_i = \bar{y}_i\}, \tag{5}$$

which control the pruner accuracy (as discussed in (Feng et al., 2024)). As we mentioned previously, we suppose training on $n \geq \hat{n}$ real data, modeling a situation where new real samples are available with $\hat{n}$ controlling the generative model quality in generating faithful synthetic features[1].

$L^2$**-loss.** In the remainder of the paper we take $\ell$ to be the regularized squared loss as it allows us to obtain a closed-form solution for the optimization problem in equation 4, hence, a more tractable analysis. Specifically, we take $\ell(\boldsymbol{x}, y; \boldsymbol{w}) = (\boldsymbol{w}^\top \boldsymbol{x} - y)^2 + \gamma \|\boldsymbol{w}\|^2$ where $\gamma \geq 0$ is a regularisation parameter, which yields the following closed-form solution

$$\boldsymbol{w} = \frac{1}{N}\mathbf{Q}\mathbf{X}\boldsymbol{y}, \quad \mathbf{Q} = \left(\frac{1}{N}\mathbf{X}\mathbf{X}^\top + \gamma \mathbf{I}_p\right)^{-1}. \tag{6}$$

where $N = n + m$, the matrix $\mathbf{X} = (\boldsymbol{x}_1, \ldots, \boldsymbol{x}_n, q_1\tilde{\boldsymbol{x}}_1, \ldots, q_m\tilde{\boldsymbol{x}}_m) \in \mathbb{R}^{p \times N}$ is the concatenation of both real and (pruned) synthetic features, and the vector $\boldsymbol{y} = (y_1, \ldots, y_n, \tilde{y}_1, \ldots, \tilde{y}_m) \in \mathbb{R}^N$ is the concatenation of real and (noisy) synthetic labels.

## 4 MAIN RESULTS

In this section, we present and discuss the main results obtained through the analysis of the classifier model defined in equation 6. We start by specifying the supposed growth rate assumptions.

**Assumption 4.1** (Growth Rate). *We consider a high-dimensional regime where $p, n, \hat{n}, m \to \infty$ and we recall $N = n + m$ such that:*

*1)* $\frac{p}{N} \to \eta \in [0, \infty)$,     *2)* $\frac{p}{\hat{n}} \to \hat{\eta} \in [0, \infty)$,     *3)* $\frac{n}{N} \to \pi \in [0, 1]$,     *4)* $\|\boldsymbol{\mu}\| = \mathcal{O}(1)$.

**Role of the assumptions.** The above assumptions are central to understanding the nuances between real and synthetic data (as constructed above) in a high-dimensional regime. Essentially,

- Assumptions 1), 2), and 3) define the scaling of data dimension $p$ and the different sample sizes ($n$ real data, $m$ synthetic data, and $\hat{n}$ real samples used to train the generative model). In particular, we suppose that all these dimensions scale linearly relative to each other, which corresponds to the classical RMT regime. This setting is more general than the infinite sample size regime in the sense that the former can be recovered by taking $\eta, \hat{\eta} \to 0$. Specifically, the parameter $\hat{\eta}$ controls the generative model quality, where lower values indicate better generative model quality. Plus, the parameter $\pi$ corresponds to the proportion of the real samples in the data mixture. For instance, $\pi = 0$ models a setting where the training is done only on synthetic samples, and $0 < \pi < 1$ highlights the fact that the number $n$ of real and $m$ of synthetic samples are of the same order, therefore, making our results scalable to any possible proportion $\pi$.

- The fourth condition about the magnitude of the mean vector $\boldsymbol{\mu}$ reflects the fact that the classification problem should neither be trivial ($\|\boldsymbol{\mu}\| \gg 1$) nor impossible ($\|\boldsymbol{\mu}\| \to 0$) as the dimension of data grows large. For instance, assuming $\|\boldsymbol{\mu}\|$ of order $O(\sqrt{p})$ would not be relevant as $p \to \infty$ since the classification problem becomes trivial in this regime. We refer the reader to (Couillet & Benaych-Georges, 2016) for a more general formulation and justifications of this assumption under an extended $k$-class Gaussian mixture model.

Having stated the main assumptions, we are now in place to present our main technical findings on the performance of the classifier model trained on a mixture of real and synthetic data. As a corollary, we also cover the case where the model is trained solely on synthetic data and showcase a generalization of the result obtained by Feng et al. (2024).

---

[1]Technically, our results hold irrespective of the statistical dependencies between the data used to train the generative model in equation 2 or the classifier in equation 6.

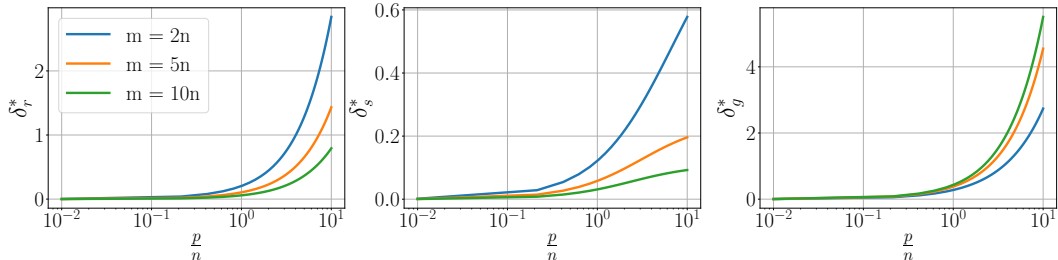

Figure 2: Behavior of $(\delta_r^*, \delta_s^*, \delta_g^*)$ in terms of the ratio $\frac{p}{n}$. For small ratio $\frac{p}{n}$, the values of $\delta_r^*, \delta_s^*, \delta_g^*$ are close to 0. $(\delta_r^*, \delta_s^*, \delta_g^*)$ are computed by iterating the system 7 starting from random values.

## 4.1 PARTIALLY SYNTHETIC: MIXTURE OF REAL AND SYNTHETIC DATA

We start by analyzing the general case of training on a mix of real and synthetic data. As we described in the previous section, the statistics of synthetic data are empirical estimates of the ones of real data. Under Assumption 4.1, the estimation of $\boldsymbol{\mu}$ with $\hat{\boldsymbol{\mu}}$ remains consistent, while the estimation of the underlying real data covariance (i.e., $\mathbf{I}_p$ in our setting) with $\hat{\mathbf{C}}$ is inconsistent as we previously discussed. As a result, studying the theoretical performance of the classifier in equation 6 demands deploying tools from random matrix theory that refines the estimation of scalar quantities depending on large random matrices. In our case, the scalar quantity of interest corresponds to the model's accuracy which depends on the random matrices $\hat{\mathbf{C}}$ and $\mathbf{X}\mathbf{X}^\top$ as per equation 6.

In our analysis of the classifier's theoretical performance, we found that the effect of high-dimension (and that of distribution shift between real and synthetic samples) is described by three scalar quantities $(\delta_r^*, \delta_s^*, \delta_g^*)$ which are defined as the unique solution of the following fixed point system which is derived from Lemma D.1 in the Appendix.

$$\delta_g = \frac{\alpha(1-\pi)}{1+\delta_s} \cdot \frac{\hat{\eta}}{\gamma + \frac{\pi}{1+\delta_r} + \frac{\alpha(1-\pi)}{(1+\delta_s)(1+\delta_g)}}, \quad \delta_r = \frac{\eta}{\hat{\eta}} \cdot \frac{1+\delta_s}{\alpha(1-\pi)}\delta_g, \quad \delta_s = \frac{\alpha\delta_r}{1+\delta_g}. \quad (7)$$

where $\alpha = \phi(1-\varepsilon) + \rho\varepsilon$. These quantities will be used subsequently in our results. Intuitively, $\delta_r^*$ captures the contribution of real data, $\delta_s^*$ corresponds to the contribution of synthetic data, and $\delta_g^*$ corresponds to the influence of the generative model. In an infinite sample size regime where $n, m, \hat{n} \to \infty$ while the dimension $p$ is kept fixed, $(\delta_r^*, \delta_s^*, \delta_g^*) = (0,0,0)$ as per Fig. 2, while under Assumption 4.1 these quantities are non zero yielding a counterintuitive behavior in high-dimension. For convenience, we further define a set of scalar quantities that will prove useful in the next result.

$$\alpha = \mathbb{E}[q_i] = \phi(1-\varepsilon) + \rho\varepsilon, \quad \lambda = \mathbb{E}[q_i \tilde{y}_i] = \phi(1-\varepsilon) - \rho\varepsilon,$$

$$a = \frac{\pi}{1+\delta_r^*} + \frac{\alpha(1-\pi)}{1+\delta_s^*}, \quad b = \gamma + \frac{\pi}{1+\delta_r^*} + \frac{\alpha(1-\pi)}{(1+\delta_s^*)(1+\delta_g^*)}, \quad c = \frac{\pi}{1+\delta_r^*} + \frac{\lambda(1-\pi)}{1+\delta_s^*},$$

$$a_1 = \frac{\pi\eta}{(1+\delta_r^*)^2 h_2 b^2}, \quad b_1 = \frac{\alpha(1-\pi)\eta}{(1+\delta_s^*)^2(1+\delta_g^*)^2 h_2 b^2}, \quad b_2 = \frac{\alpha(1-\pi)\eta}{(1+\delta_s^*)^2(1+\delta_g^*)^4 h_2 b^2},$$

$$h_1 = 1 - a_1 - b_2, \quad h_2 = 1 - \left(\frac{\alpha(1-\pi)}{(1+\delta_s^*)(1+\delta_g^*)}\right)^2 \frac{\hat{\eta}}{b^2}.$$

The first set of parameters $(\alpha, \lambda, a, b, c)$ pop out from the expectation of the classifier's decision function while the remaining quantities are related to second-order statistics. Essentially, the main relevant quantities to our analysis are $\hat{\eta}$ and $\varepsilon$ which characterize the quality of synthetic data, with $\phi$ and $\rho$ characterizing the verification process. In an idealized scenario, we would have $\hat{\eta} = \varepsilon = 0$ which reflects the fact that there is no distribution shift nor label noise, while $\phi = 1 - \rho = 1$ corresponds to a perfect (oracle) verification process. Our main goal is to study how these parameters influence the classifier's performance hence providing the conditions that make synthetic data relevant for performance boost. The main result brought by this paper is therefore stated as follows.

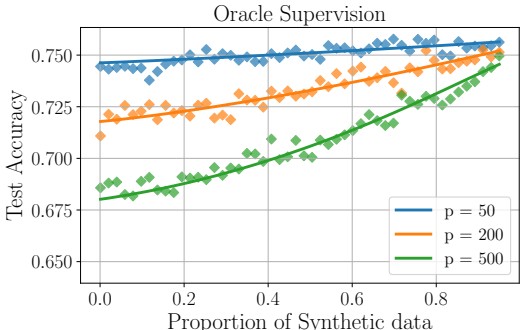 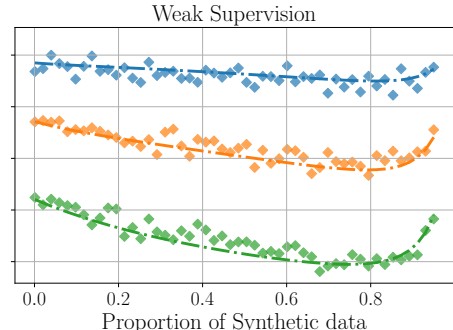

Figure 3: Scatter plots correspond to empirical test accuracy while lines correspond to the theoretical counterpart as per Theorem 4.2. The parameters used in this experiments are: $n = \hat{n} = 1000$, $\|\boldsymbol{\mu}\| = 0.7$ and $\gamma = 1$, $(\rho, \phi) = (0, 1)$ for Oracle supervision and $(\rho, \phi) = (1, 0.5)$ for the Weak supervision. The parameter $\varepsilon$ is variable depending on the proportion of synthetic data by taking it equal to the misclassification error corresponding to training a classifier on synthetic data only. As theoretically anticipated, a boost of performance is observed with synthetic data supervision while distribution shift affects negatively the performance.

**Theorem 4.2** (Theoretical performance). *Let $\boldsymbol{w}$ be the Ridge classifier as defined in equation 6 and suppose that Assumption 4.1 holds. The decision function $\boldsymbol{w}^\top \boldsymbol{x}$, on some (real) test sample $\boldsymbol{x} \in \mathcal{C}_a$, with corresponding label $y = (-1)^a$ and independent of $\mathbf{X}$, satisfies*

$$\boldsymbol{w}^\top \boldsymbol{x} \xrightarrow{\mathcal{D}} \mathcal{N}\left(y \cdot \mu, \nu - m^2\right),$$

*where $\mu = \frac{c\|\boldsymbol{\mu}\|^2}{b + a\|\boldsymbol{\mu}\|^2}$ and*

$$\nu = \frac{c\|\boldsymbol{\mu}\|^2}{h_1(b + a\|\boldsymbol{\mu}\|^2)^2}\left(c(1 + b_1 - b_2)\|\boldsymbol{\mu}\|^2 + \frac{c}{h_2} - 2\left(a_1 + \frac{\lambda b_1}{\alpha}\right)(b + a\|\boldsymbol{\mu}\|^2)\right) + \frac{a_1 + b_1}{h_1}.$$

*Moreover, the asymptotic test accuracy of the classifier is given by $\Phi\left((\nu - m^2)^{-\frac{1}{2}}m\right)$ where $\Phi(x) = \frac{1}{\sqrt{2\pi}}\int_{-\infty}^{x} e^{-\frac{t^2}{2}}\,dt$.*

Theorem 4.2 states that the decision function of the classifier in equation 6 is asymptotically equivalent to the thresholding of two monovariate Gaussian random variables with respective means $\mu$ and $-\mu$ and standard deviation $\nu$, where the statistics $\mu$ and $\nu$ are expressed in terms of the scalar quantities defined above. Here, $\mu$ represents the signal strength while $\nu$ highlights the classifier's uncertainty or dispersion. To provide some insights into the implications of this theorem, we start by examining it in a low-dimensional regime where $p$ is kept fixed while $n, m, \hat{n} \to \infty$. In this case, we have $\eta, \hat{\eta} \to 0$ and $\delta_r^*, \delta_s^*, \delta_g^* \to 0$ which yields

$$a = \pi + \alpha(1 - \pi), \quad b = \gamma = \pi + \alpha(1 - \pi), \quad c = \pi + \lambda(1 - \pi),$$

and $a_1 = b_1 = b_2 = 0$ with $h_1 = h_2 = 1$. As such, the accuracy of the classifier increases with $\lambda$, i.e., when the synthetic labels are verified (large $\frac{\phi}{\rho}$) or less noisy (small $\varepsilon$). This is in line with the findings of Feng et al. (2024) while extended by our result to training on a mix of real and synthetic data. However, when the dimension scales linearly with the different sample sizes, the values of $\delta_r^*, \delta_s^*, \delta_g^* \nrightarrow 0$ yielding a lower signal strength $\mu$ and higher variance $\nu^2$. This highlights the fact that in high-dimension, even if the synthetic labels are not noisy or equivalently well verified, there is a performance drop due to the feature distribution shift between real and synthetic data.

Fig. 3 depicts the empirical test accuracy and the theoretical prediction as per Theorem 4.2 when varying the proportion of synthetic data. As theoretically anticipated, adding synthetic data does not boost the classifier's performance unless it is verified accurately (oracle supervision versus weak supervision). Moreover, our results show the effect of the distribution shift which heavily affects performance in the case of weak supervision (Fig. 3 right).

## 4.2 FULLY SYNTHETIC: TRAINING ON SYNTHETIC DATA

In this section, we study the fully synthetic setting which corresponds to training solely on synthetic data (i.e. $n = 0$ in equation 6). For simplicity, we consider only label noise and ignore feature noise in the synthetic data. Essentially, this allows us to exhibit the smooth phase transition of the classifier's accuracy in terms of label noise, which extends the result of Feng et al. (2024). Specifically, we obtain the following corollary of theorem 4.2.

**Corollary 4.3** (Performance when training only on synthetic data). *Let $w_s$ be the Ridge classifier described in equation 6 trained only on synthetic data with only label noise (i.e., $\hat{\mathbf{C}} = \mathbf{I}_p$). Under Assumption 4.1, the decision function $w_s^\top x$ on a test sample $x \in \mathcal{C}_a$ with corresponding label $y = (-1)^a$ and independent of $\mathbf{X}$, satisfies*

$$w_s^\top x \xrightarrow{\mathcal{D}} \mathcal{N}\left(y \cdot \mu_s, \nu_s - \mu_s^2\right),$$

*where*

$$\mu_s = \frac{\phi(1-\varepsilon) - \rho\varepsilon}{\alpha\|\boldsymbol{\mu}\|^2 + \alpha + \gamma(1+\delta_s)}\|\boldsymbol{\mu}\|^2,$$

$$\nu_s = \frac{\lambda^2\|\boldsymbol{\mu}\|^2}{h(\alpha\|\boldsymbol{\mu}\|^2 + \alpha + \gamma(1+\delta_s))}\left(\frac{\|\boldsymbol{\mu}\|^2 + 1}{\alpha\|\boldsymbol{\mu}\|^2 + \alpha + \gamma(1+\delta_s)} - \frac{2(1-h)}{\alpha}\right) + \frac{1-h}{h},$$

*with*

$$\eta_s = \lim_{p\to\infty} \frac{p}{m}, \quad h = 1 - \frac{\alpha\eta_s}{(\alpha + \gamma(1+\delta))^2}, \quad \delta_s = \frac{\eta_s\alpha - \alpha - \gamma + \sqrt{(\alpha + \gamma - \eta_s\alpha)^2 + 4\eta_s\alpha\gamma}}{2\gamma}.$$

Corollary 4.3 provides an explicit formulation of Theorem 4.2 with synthetic data only and ignoring distribution shift (yielding an explicit expression of $\delta_s$). This setting provides a clearer interpretation of the effect of label noise since the classifier's performance is directly related to the quantity $\lambda = \phi(1-\varepsilon) - \rho\varepsilon$. The breaking point of the classifier's performance occurs at $\lambda = 0$, which corresponds to the accuracy of random guessing, yielding to the critical value of label noise $\varepsilon^* = (1 + \frac{\rho}{\phi})^{-1}$. This critical value is equivalent to the one obtained by Feng et al. (2024), however, we extend their result to the high-dimensional setting which exhibits a smoother phase transition as depicted in Fig. 4. Essentially, the sharp phase transition of Feng et al. (2024) is covered by our result by taking $\eta_s \to 0$. In this sense, the predicted smooth transition better mirrors real-world scenarios where finite sample sizes introduce gradual changes in performance rather than abrupt shifts. This makes our theoretical findings more applicable and reliable for practical scenarios.

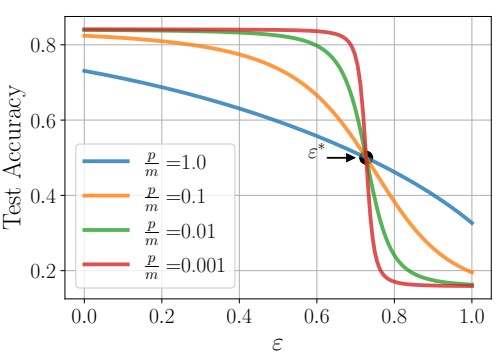

Figure 4: Phase transition in terms of label noise as predicted by Corollary 4.3. The critical value for $\varepsilon$ is predicted at $\varepsilon^* = (1 + \frac{\rho}{\phi})^{-1}$. We fix $p = 100$ and vary $m$. The remaining parameters are $\|\boldsymbol{\mu}\| = 1$, $\rho = 0.3$ and $\phi = 0.8$, i.e. $\varepsilon^* = 0.73$.

## 5 EXPERIMENTS

In this section, we present our experiments conducted on different real-world tasks and datasets in order to illustrate our theoretical findings presented in the previous section.

### 5.1 EXPERIMENTAL SETTINGS

**Amazon Reviews.** We use the Amazon Reviews datasets (Blitzer et al. (2007)) which include several binary classification tasks corresponding to positive versus negative reviews of `books`,

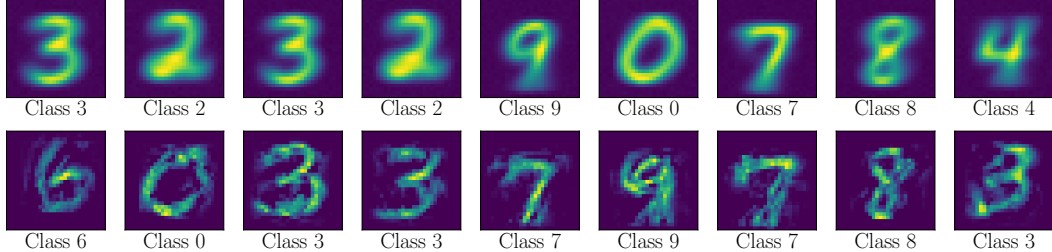

| Class 3 | Class 2 | Class 3 | Class 2 | Class 9 | Class 0 | Class 7 | Class 8 | Class 4 |

| Class 6 | Class 0 | Class 3 | Class 3 | Class 7 | Class 9 | Class 7 | Class 8 | Class 3 |

Figure 5: Illustration of two different generation schemes for the MNIST data. *Top figure:* Generating MNIST-like data samples by only estimating the mean of each class $\hat{\boldsymbol{\mu}}_a$ for $a \in [10]$ and without estimating the covariance matrix, i.e samples here are generated through the distribution $\mathcal{N}(\hat{\boldsymbol{\mu}}_a, \mathbf{I}_p)$. *Bottom figure:* Generating samples by estimating both the mean and covariance of each class, as of our considered generative model defined in equation 2.

`electronics` and `kitchen`. We apply the standard scaler from `sklearn` (Pedregosa et al., 2011) and estimate $\|\boldsymbol{\mu}\|$ with the normalized data. The synthetic data is generated following the described generative scheme (see equation 2). We use the Ridge classifier in equation 6 for this data.

**MNIST.** We also conducted experiments on the MNIST (LeCun & Cortes (2010)) dataset to illustrate our theoretical insights, by training a simple neural network with one-hidden layer and ReLU activation function. Concerning the synthetic data, we used different values of $\hat{n}$ to generate new samples in order to highlight the importance of the generation quality, and introduced a label noise $\varepsilon$ to highlight the importance of the pruning. Figure 5 shows some examples of MNIST-like synthetic data that has been generated and used in our experiments.

**LLM Safety Alignment.** We also investigated the impact of synthetic text data for the task of alignment of LLMs with direct preference optimization on safety datasets, using the same approach as in (Alami et al., 2024). We finetune the `Falcon 2-11B` Instruct model (Malartic et al., 2024) on $n = 5000$ human data from Anthropic's *HH-RLHF* dataset[2], which correspond to real data, while synthetic data are extracted from the PKU safe RLHF dataset[3] which are generated using `Alpaca3-70B`[4]. We increase the amount of synthetic data by injecting gradually five batches of 7000 samples per batch, to study the performance of the fine-tuned model as we add more synthetic data. In this experiment, we focus only on label noise by randomly perturbing the synthetic dataset. Each entry from the synthetic dataset includes a prompt $x^{(j)}$, a safe response $y_{s_w}^{(j)}$ (safety-accepted response), and a less safe response $y_{s_l}^{(j)}$ (safety-rejected response). We, therefore, perturbed this dataset by swapping safe and less safe responses with a probability $\varepsilon$ (label noise), and selecting the prompts according to a verifier of parameters $\rho$ and $\phi$ described earlier in this paper.

For the evaluation, we use the ALERT dataset[5] (Tedeschi et al. (2024)) to test the safety of responses of the finetuned model after being judged by `LLama-Guard-3-8B` (Dubey et al., 2024). As in (Alami et al., 2024), we compute the safety score as the percentage of safe answers labeled by `Llama-Guard-3-8B`. We report the results in figure 8 for strong supervision $(\rho, \phi) = (0.2, 0.9)$ and weak supervision $(\rho, \phi) = (0.5, 0.5)$ for both $\varepsilon = 0.1$ and $\varepsilon = 0.5$.

**LLM Q&A Safety Generation.** This experiment aims to evaluate the impact of synthetically generated prompts (i.e. feature noise). To construct the generative model for this experiment, we fine-tune an LLM ($M$) with supervised fine-tuning (SFT) on pairs of question-answer (Q&A) sentences extracted from a safety dataset. Initially, we fine-tune $M$ on $12k$ human annotated Q&A as safe or unsafe (Ji et al., 2024), yielding a fine-tuned model on human data denoted as $M_h$. Then, $M_h$ is considered as the generative model to generate a large dataset of synthetic Q&A prompts

---

[2] `https://huggingface.co/datasets/yimingzhang/hh-rlhf-safety`

[3] `https://huggingface.co/datasets/PKU-Alignment/PKU-SafeRLHF`

[4] `https://huggingface.co/PKU-Alignment/alpaca-70b-reproduced-llama-3`

[5] `https://github.com/Babelscape/ALERT/blob/master/data/alert.jsonl`

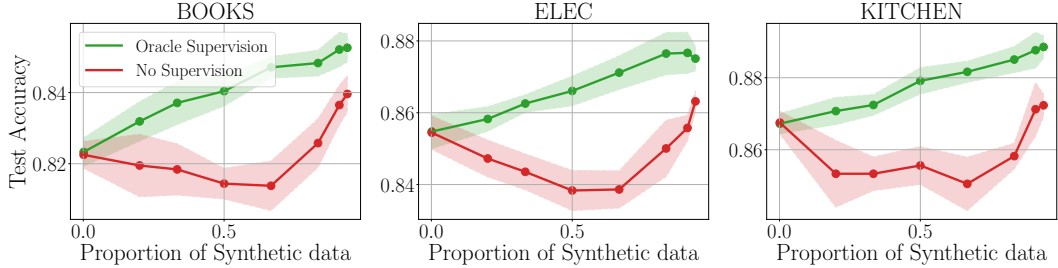

Figure 6: **Results of the Amazon Reviews setting**: Test Accuracy with the proportion of synthetic data evaluated on Amazon Review Blitzer et al. (2007) dataset. The number of real data sample used is $n = 800$, the dimension is $p = 400$, $\gamma = 10^{-1}$ and $\varepsilon = 0.2$ (fixed). The pruning parameters are $(\rho, \phi) = (0, 1)$ for Oracle supervision and $(\rho, \phi) = (1, 1)$ for No supervision.

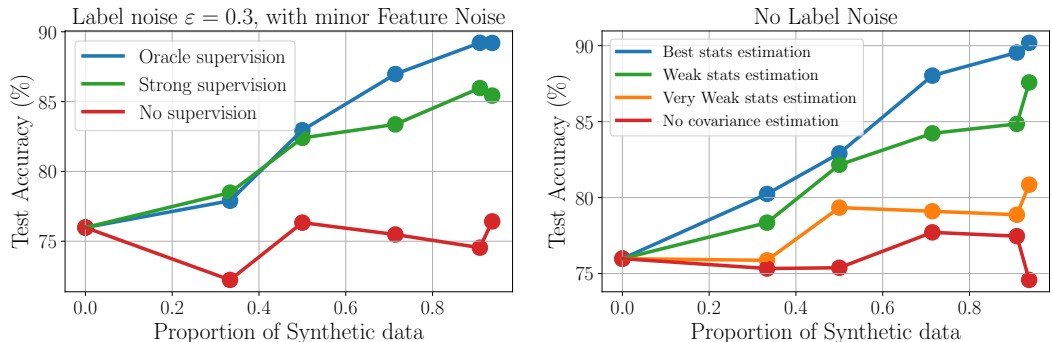

Figure 7: **Results of the MNIST setting**: Training an NN with one hidden layer and ReLU activation function on a mixture of real ($n = 500$) and varying the proportion of synthetic Gaussian data.

(around $120k$ samples) that were further annotated as safe/unsafe using `Mistral-Nemo`[6] and `Qwen2-7B-Instruct` (Yang et al., 2024), which incorporate a further label noise. To verify the data, we use `Llama-Guard-3.1` (Dubey et al., 2024). We conducted this experiment using two LLMs (M) of different sizes (to vary the generative model quality) which are the `Llama-3.1-8B` and `Gemma-2-2B-it` (Team et al., 2024) instruct models.

## 5.2 EFFECT OF LABEL NOISE

Figures 6, 7 (left plot) 8 reflect the effect of label noise. Essentially, as theoretically anticipated, the trained models do not benefit from synthetic data unless it is accurately verified. Specifically, in the case of weak supervision, model performance drops significantly, and the improvement from using synthetic data is only visible with very high synthetic sample sizes. On the contrary, with strong supervision, we observe a monotonous performance boost as the proportion of synthetic data increases.

## 5.3 EFFECT OF FEATURE NOISE

In this section, we discuss the experiments related to feature noise. In Fig. 7 (right), we depict the performance of a one-hidden layer MLP trained on a mix of real and synthetic MNIST data following our theoretical framework. As we can observe from the figure, the performance boost from synthetic data heavily depends on the generative model quality as predicted by our theoretical results. We further observe the same trend using LLMs as depicted in Fig. 9, where we observe that the synthetic data generated by `Llama3.1-8B-Instruct` yields a better performance boost compared to `Gemma-2-2B-it` as we increase the amount of synthetic samples, which means

---

[6]`https://mistral.ai/news/mistral-nemo/`

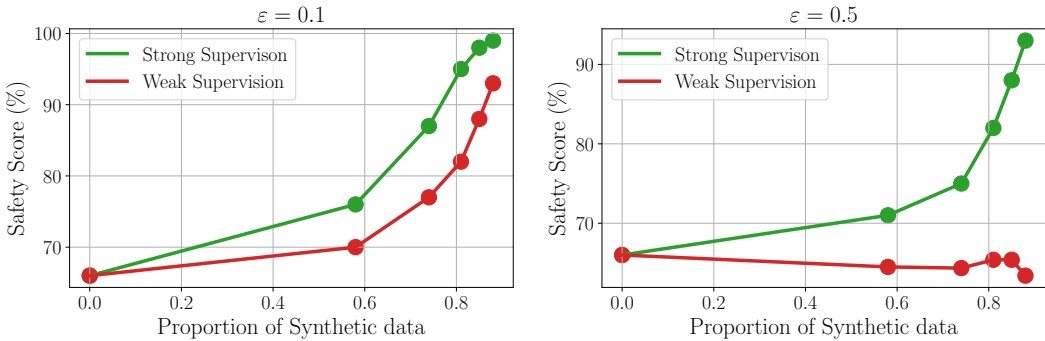

Figure 8: **Results of LLM Safety Alignment**: Strong supervision corresponds to $(\rho, \phi) = (0.2, 0.9)$ and weak supervision to $(\rho, \phi) = (0.5, 0.5)$.

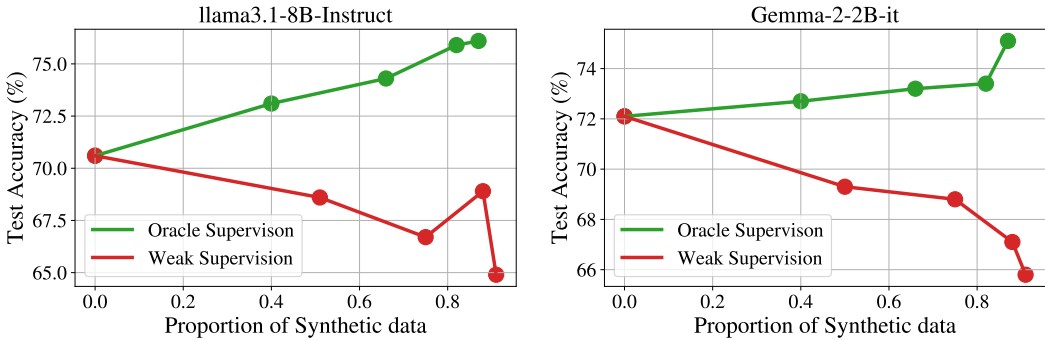

Figure 9: **Results of LLM Q&A Safety Generation**: Evaluation of two LLMs trained as presented in section 5.1 is depicted for both **(left)** $M=$ `Llama3.1-8B-Instruct` and **(right)** $M=$ `Gemma-2-2B-it`. The test accuracy is computed over the testing dataset extracted from Ji et al. (2024), with $2.8k$ Q&A samples. The results shown are the average over 3 runs.

that `Llama3.1-8B-Instruct` generates better synthetic samples (less distribution shift) than `Gemma-2-2B-it`.

## 6 CONCLUSION AND LIMITATIONS

In this work, we conducted a comprehensive theoretical and empirical analysis of models trained on a mixture of real and synthetic data with verification. By leveraging random matrix theory, we identified critical factors such as distribution shifts and label noise that significantly impact model performance. Our findings demonstrate that synthetic data can enhance model accuracy under specific conditions, particularly when the generative model is of high quality and the verification process is accurate. Additionally, we extended previous research by showing that performance transitions are smooth rather than sharp when synthetic data is incorporated in high-dimensional settings.

Despite these advancements, our current setting is limited to label verification of synthetic data. Incorporating feature verification represents a promising extension for future research, which could provide further insights into the reliability and effectiveness of synthetic data in model training. Another possible extension of our work is to study distributions beyond the Gaussian model and analyze how higher-order statistics can be incorporated into our current framework.

In conclusion, this work provides a foundational understanding of the conditions under which synthetic data can be beneficial for model training in high-dimensional settings. By integrating both theoretical insights and empirical validations, this study provides new insights into the effective utilization of synthetic data, paving the way for more resilient and performant AI models.

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

APPENDIX

# Contents

## A  USEFUL LEMMAS

**Notation:**  For $a \in \{1, 2\}$, we denote by $\mathbb{I}_a = \{i \mid \boldsymbol{x}_i \in \mathcal{C}_a\}$, i.e, the set of indices of vectors belonging to class $\mathcal{C}_a$. Furthermore, we denote $\Sigma = \boldsymbol{\mu}\boldsymbol{\mu}^\top + \mathbf{I}_p = \mathbb{E}\left[\boldsymbol{x}\boldsymbol{x}^\top\right]$ for $\boldsymbol{x} \in \mathcal{C}_a$, and $\Sigma_\beta = \boldsymbol{\mu}_\beta \boldsymbol{\mu}_\beta^\top + \mathbf{C}$

Here we will list the most useful lemmas and results used in our analysis.

### A.1  GENERAL LEMMAS

**Lemma A.1** (Inverse identity). *For invertible matrices $\mathbf{A}$ and $\mathbf{B}$, we have that:*
$$\mathbf{A}^{-1} - \mathbf{B}^{-1} = \mathbf{A}^{-1}(\mathbf{B} - \mathbf{A})\mathbf{B}^{-1}$$

**Lemma A.2** (Woodbury). *For $\mathbf{A} \in \mathbb{R}^{p \times p}$, $\mathbf{U}, \mathbf{V} \in \mathbb{R}^{p \times k}$, such that both $\mathbf{A}$ and $\mathbf{A} + \mathbf{U}\mathbf{V}^\top$ are invertible, we have:*
$$\left(\mathbf{A} + \mathbf{U}\mathbf{V}^\top\right)^{-1} = \mathbf{A}^{-1} - \mathbf{A}^{-1}\mathbf{U}\left(\mathbf{I}_k + \mathbf{V}^\top\mathbf{A}^{-1}\mathbf{U}\right)^{-1}\mathbf{V}^\top\mathbf{A}^{-1}$$

A particular case of this lemma A.2, in the case of $k = 1$, is called *Sherman-Morisson*'s identity.

**Lemma A.3** (Sherman-Morisson). *For $\mathbf{A} \in \mathbb{R}^{p \times p}$ invertible and $\boldsymbol{u}, \boldsymbol{v} \in \mathbb{R}^p$, $\mathbf{A} + \boldsymbol{u}\boldsymbol{v}^\top$ is invertible if and only if : $1 + \boldsymbol{v}^\top \mathbf{A}\boldsymbol{u} \neq 0$, and:*
$$(\mathbf{A} + \boldsymbol{u}\boldsymbol{v}^\top)^{-1} = \mathbf{A}^{-1} - \frac{\mathbf{A}^{-1}\boldsymbol{u}\boldsymbol{v}^\top\mathbf{A}^{-1}}{1 + \boldsymbol{v}^\top\mathbf{A}^{-1}\boldsymbol{u}}$$

*Besides,*
$$(\mathbf{A} + \boldsymbol{u}\boldsymbol{v}^\top)^{-1}\boldsymbol{u} = \frac{\mathbf{A}^{-1}\boldsymbol{u}}{1 + \boldsymbol{v}^\top\mathbf{A}^{-1}\boldsymbol{u}}$$

## A.2 DETERMINISTIC EQUIVALENTS

Let us state here the deterministic equivalent of the resolvent matrix $\mathbf{Q}$ defined in the general model's equation (6) for any general covariance matrix $\mathbf{C}$ and mean $\boldsymbol{\mu}_\beta = \beta\boldsymbol{\mu} + \boldsymbol{\mu}^\perp$ that define the statistic of the synthetic data, as in equation 13.

**Lemma A.4** (Deterministic equivalent of $\mathbf{Q}$). *Under the 4.1 assumptions listed above in the main paper, a deterministic equivalent for $\mathbf{Q} \equiv \mathbf{Q}(\gamma)$, denoted $\bar{\mathbf{Q}}$, is given by:*

$$\bar{\mathbf{Q}} = \left( \frac{\pi(\boldsymbol{\mu}\boldsymbol{\mu}^\top + \mathbf{I}_p)}{1+\delta} + \frac{\alpha(1-\pi)(\boldsymbol{\mu}_\beta\boldsymbol{\mu}_\beta^\top + \mathbf{C})}{1+\delta_S} + \gamma\mathbf{I}_p \right)^{-1}$$

*where:*

$$\pi = \frac{n}{n+m}, \quad \alpha = \phi(1-\varepsilon) + \rho\varepsilon, \quad \delta = \frac{1}{N}\operatorname{Tr}(\bar{\mathbf{Q}}), \quad \delta_S = \frac{\alpha}{N}\operatorname{Tr}(\mathbf{C}\bar{\mathbf{Q}})$$

*Proof.* We want to find $\bar{\mathbf{Q}}$ such that for all bounded $\boldsymbol{a}, \boldsymbol{b} \in \mathbb{R}^p$:

$$\boldsymbol{a}^\top(\mathbb{E}[\mathbf{Q}] - \bar{\mathbf{Q}})\boldsymbol{b} \to 0$$

Let $\bar{\mathbf{Q}} = (\mathbf{S} + \gamma\mathbf{I}_p)^{-1}$. We want to determine an $\mathbf{S}$ that satisfies the above property. We have that:

$$\mathbb{E}[\mathbf{Q}] - \bar{\mathbf{Q}} = \mathbb{E}[\mathbf{Q}(\mathbf{S} - \frac{1}{N}\mathbf{V}\mathbf{V}^\top)\bar{\mathbf{Q}}] \quad (\text{lemma} A.1)$$

$$= \frac{1}{N}\sum_{i=1}^N \mathbb{E}[\mathbf{Q}(\mathbf{S} - \boldsymbol{v}_i\boldsymbol{v}_i^\top)\bar{\mathbf{Q}}]$$

$$= \frac{1}{N}\sum_{i=1}^n \mathbb{E}[\mathbf{Q}(\mathbf{S} - \boldsymbol{x}_i\boldsymbol{x}_i^\top)\bar{\mathbf{Q}}] + \frac{1}{N}\sum_{i=1}^m \mathbb{E}[\mathbf{Q}(\mathbf{S} - q_i\tilde{\boldsymbol{x}}_i\tilde{\boldsymbol{x}}_i^\top)\bar{\mathbf{Q}}]$$

$$= \frac{1}{N}\sum_{i=1}^n \mathbb{E}[\mathbf{Q}\mathbf{S} - \frac{1}{1+\delta_R}\mathbf{Q}_{-\boldsymbol{x}_i}\boldsymbol{x}_i\boldsymbol{x}_i^\top]\bar{\mathbf{Q}} + \frac{1}{N}\sum_{i=1}^m \mathbb{E}[\mathbf{Q}\mathbf{S} - \frac{q_i}{1+\delta_S}\mathbf{Q}_{-\tilde{\boldsymbol{x}}_i}\tilde{\boldsymbol{x}}_i\tilde{\boldsymbol{x}}_i^\top]\bar{\mathbf{Q}}$$

$$= \pi\mathbb{E}[\mathbf{Q}\mathbf{S} - \frac{1}{1+\delta_R}\mathbf{Q}_{-\boldsymbol{x}_i}\boldsymbol{x}_i\boldsymbol{x}_i^\top]\bar{\mathbf{Q}} + (1-\pi)\mathbb{E}[\mathbf{Q}\mathbf{S} - \frac{q_i}{1+\delta_S}\mathbf{Q}_{-\tilde{\boldsymbol{x}}_i}\tilde{\boldsymbol{x}}_i\tilde{\boldsymbol{x}}_i^\top]\bar{\mathbf{Q}}$$

$$= \pi\mathbb{E}[\mathbf{Q}_{-\boldsymbol{x}_i}(\mathbf{S} - \frac{\boldsymbol{x}_i\boldsymbol{x}_i^\top}{1+\delta_R})\bar{\mathbf{Q}}] + (1-\pi)\mathbb{E}[\mathbf{Q}_{-\tilde{\boldsymbol{x}}_i}(\mathbf{S} - \frac{q_i\tilde{\boldsymbol{x}}_i\tilde{\boldsymbol{x}}_i^\top}{1+\delta_S})\bar{\mathbf{Q}}] + \mathcal{O}(N^{-1})$$

Thus, it suffices to have:

$$\mathbf{S} = \frac{\pi(\boldsymbol{\mu}\boldsymbol{\mu}^\top + \mathbf{I}_p)}{1+\delta_R} + \frac{\alpha(1-\pi)(\boldsymbol{\mu}_\beta\boldsymbol{\mu}_\beta^\top + \mathbf{C})}{1+\delta_S}$$

to get the desired property. $\square$

**Lemma A.5** (Deterministic equivalent of $\mathbf{QAQ}$). *Let $\mathbf{A} \in \mathbb{R}^{p\times p}$ be any deterministic symmetric semi-definite matrix. We have that:*

$$\mathbf{QAQ} \leftrightarrow \bar{\mathbf{Q}}\mathbf{A}\bar{\mathbf{Q}} + \frac{\pi}{N(1+\delta)^2}\operatorname{Tr}(\Sigma\bar{\mathbf{Q}}\mathbf{A}\bar{\mathbf{Q}})\mathbb{E}[\mathbf{Q}\Sigma\mathbf{Q}] + \frac{\alpha(1-\pi)}{N(1+\delta_S)^2}\operatorname{Tr}(\Sigma_\beta\bar{\mathbf{Q}}\mathbf{A}\bar{\mathbf{Q}})\mathbb{E}[\mathbf{Q}\Sigma_\beta\mathbf{Q}]$$

*Thus, we get that for $\mathbf{A} = \Sigma$, and for $\mathbf{A} = \Sigma_\beta$:*

$$\mathbf{Q}\Sigma\mathbf{Q} \leftrightarrow \bar{\mathbf{Q}}\Sigma\bar{\mathbf{Q}} + \frac{\pi}{N(1+\delta)^2}\operatorname{Tr}((\Sigma\bar{\mathbf{Q}})^2)\mathbb{E}[\mathbf{Q}\Sigma\mathbf{Q}] + \frac{\alpha(1-\pi)}{N(1+\delta_S)^2}\operatorname{Tr}(\Sigma_\beta\bar{\mathbf{Q}}\Sigma\bar{\mathbf{Q}})\mathbb{E}[\mathbf{Q}\Sigma_\beta\mathbf{Q}]$$

$$\mathbf{Q}\Sigma_\beta\mathbf{Q} \leftrightarrow \bar{\mathbf{Q}}\Sigma_\beta\bar{\mathbf{Q}} + \frac{\pi}{N(1+\delta)^2}\operatorname{Tr}(\Sigma\bar{\mathbf{Q}}\Sigma_\beta\bar{\mathbf{Q}})\mathbb{E}[\mathbf{Q}\Sigma\mathbf{Q}] + \frac{\alpha(1-\pi)}{N(1+\delta_S)^2}\operatorname{Tr}((\Sigma_\beta\bar{\mathbf{Q}})^2)\mathbb{E}[\mathbf{Q}\Sigma_\beta\mathbf{Q}]$$

*And by denoting:*

$$a_1 = \frac{\pi}{N(1+\delta)^2}\operatorname{Tr}((\Sigma\bar{\mathbf{Q}})^2), \quad b_1 = \frac{\alpha(1-\pi)}{N(1+\delta_S)^2}\operatorname{Tr}(\Sigma_\beta\bar{\mathbf{Q}}\Sigma\bar{\mathbf{Q}}),$$

$$a_2 = \frac{\pi}{N(1+\delta)^2}\operatorname{Tr}(\Sigma_\beta\bar{\mathbf{Q}}\Sigma\bar{\mathbf{Q}}), \quad b_2 = \frac{\alpha(1-\pi)}{N(1+\delta_S)^2}\operatorname{Tr}((\Sigma_\beta\bar{\mathbf{Q}})^2)$$

$$h = (1-b_2)(1-a_1) - a_2b_1$$

*We get that:*

$$\mathbf{Q}\Sigma\mathbf{Q} \leftrightarrow \frac{1-b_2}{h}\bar{\mathbf{Q}}\Sigma\bar{\mathbf{Q}} + \frac{b_1}{h}\bar{\mathbf{Q}}\Sigma_\beta\bar{\mathbf{Q}},$$

$$\mathbf{Q}\Sigma_\beta\mathbf{Q} \leftrightarrow \frac{a_2}{h}\bar{\mathbf{Q}}\Sigma\bar{\mathbf{Q}} + \frac{1-a_1}{h}\bar{\mathbf{Q}}\Sigma_\beta\bar{\mathbf{Q}}.$$

*Proof.* Recall that:

$$\bar{\mathbf{Q}}(\gamma) = \left( \frac{\pi\Sigma}{1+\delta} + \frac{\alpha(1-\pi)\Sigma_\beta}{1+\delta_S} + \gamma\mathbf{I}_p \right)^{-1}$$

Let us denote by : $\mathbf{S} = \frac{\pi\Sigma}{1+\delta} + \frac{\alpha(1-\pi)\Sigma_\beta}{1+\delta_S}$, so that: $\bar{\mathbf{Q}} = (\mathbf{S} + \gamma\mathbf{I}_p)^{-1}$.

We have that:

$$\begin{aligned}
\mathbb{E}[\mathbf{Q}\mathbf{A}\mathbf{Q}] &= \mathbb{E}[\bar{\mathbf{Q}}\mathbf{A}\mathbf{Q}] + \mathbb{E}[(\mathbf{Q}-\bar{\mathbf{Q}})\mathbf{A}\mathbf{Q}] \\
&= \bar{\mathbf{Q}}\mathbb{E}[\mathbf{A}\mathbf{Q}] + \mathbb{E}[(\mathbf{Q}-\bar{\mathbf{Q}})\mathbf{A}\mathbf{Q}] \\
&= \bar{\mathbf{Q}}\left( \mathbb{E}[\mathbf{A}\bar{\mathbf{Q}}] + \mathbb{E}[\mathbf{A}(\mathbf{Q}-\bar{\mathbf{Q}})] \right) + \mathbb{E}[(\mathbf{Q}-\bar{\mathbf{Q}})\mathbf{A}\mathbf{Q}] \\
&= \bar{\mathbf{Q}}\mathbf{A}\bar{\mathbf{Q}} + \mathbb{E}[(\mathbf{Q}-\bar{\mathbf{Q}})\mathbf{A}\mathbf{Q}] \\
&= \bar{\mathbf{Q}}\mathbf{A}\bar{\mathbf{Q}} + \mathbb{E}[\mathbf{Q}\left(\mathbf{S} - \frac{1}{N}\mathbf{V}\mathbf{V}^\top\right)\mathbf{A}\mathbf{Q}] \\
&= \bar{\mathbf{Q}}\mathbf{A}\bar{\mathbf{Q}} + \mathbb{E}[\mathbf{Q}\mathbf{S}\bar{\mathbf{Q}}\mathbf{A}\mathbf{Q}] - \frac{1}{N}\sum_{i=1}^{N}\mathbb{E}[\mathbf{Q}v_i v_i^\top\bar{\mathbf{Q}}\mathbf{A}\mathbf{Q}] \\
&= \bar{\mathbf{Q}}\mathbf{A}\bar{\mathbf{Q}} + \mathbb{E}[\mathbf{Q}\mathbf{S}\bar{\mathbf{Q}}\mathbf{A}\mathbf{Q}] - \pi\mathbb{E}[\mathbf{Q}x_i x_i^\top\bar{\mathbf{Q}}\mathbf{A}\mathbf{Q}] - (1-\pi)\mathbb{E}[\mathbf{Q}q_i\tilde{x}_i\tilde{x}_i^\top\bar{\mathbf{Q}}\mathbf{A}\mathbf{Q}]
\end{aligned}$$

And we have that:

$$\begin{aligned}
\mathbb{E}[\mathbf{Q}x_i x_i^\top\bar{\mathbf{Q}}\mathbf{A}\mathbf{Q}] &= \frac{1}{1+\delta}\mathbb{E}[\mathbf{Q}_{-x_i}x_i x_i^\top\bar{\mathbf{Q}}\mathbf{A}\mathbf{Q}] \\
&= \frac{1}{1+\delta}\mathbb{E}\left[ \mathbf{Q}_{-x_i}x_i x_i^\top\bar{\mathbf{Q}}\mathbf{A}\left( \mathbf{Q}_{-x_i} - \frac{\mathbf{Q}_{-x_i}x_i x_i^\top\mathbf{Q}_{-x_i}}{N(1+\delta)} \right) \right] \\
&= \frac{1}{1+\delta}\mathbb{E}[\mathbf{Q}_{-x_i}x_i x_i^\top\bar{\mathbf{Q}}\mathbf{A}\mathbf{Q}_{-x_i}] - \frac{1}{N(1+\delta)^2}\mathbb{E}[\mathbf{Q}_{-x_i}x_i x_i^\top\bar{\mathbf{Q}}\mathbf{A}\mathbf{Q}_{-x_i}x_i x_i^\top\mathbf{Q}_{-x_i}] \\
&= \frac{1}{1+\delta}\mathbb{E}[\mathbf{Q}\Sigma\bar{\mathbf{Q}}\mathbf{A}\mathbf{Q}] - \frac{1}{N(1+\delta)^2}\text{Tr}(\Sigma\bar{\mathbf{Q}}\mathbf{A}\bar{\mathbf{Q}})\mathbb{E}[\mathbf{Q}_{-x_i}x_i x_i^\top\mathbf{Q}_{-x_i}] \\
&= \frac{1}{1+\delta}\mathbb{E}[\mathbf{Q}\Sigma\bar{\mathbf{Q}}\mathbf{A}\mathbf{Q}] - \frac{1}{N(1+\delta)^2}\text{Tr}(\Sigma\bar{\mathbf{Q}}\mathbf{A}\bar{\mathbf{Q}})\mathbb{E}[\mathbf{Q}\Sigma\mathbf{Q}]
\end{aligned}$$

And:

$$\begin{aligned}
\mathbb{E}[q_i\mathbf{Q}\tilde{x}_i\tilde{x}_i^\top\bar{\mathbf{Q}}\mathbf{A}\mathbf{Q}] &= \frac{1}{1+\delta_S}\mathbb{E}[q_i\mathbf{Q}_{-\tilde{x}_i}\tilde{x}_i\tilde{x}_i^\top\bar{\mathbf{Q}}\mathbf{A}\mathbf{Q}] \\
&= \frac{1}{1+\delta_S}\mathbb{E}\left[ q_i\mathbf{Q}_{-\tilde{x}_i}\tilde{x}_i\tilde{x}_i^\top\bar{\mathbf{Q}}\mathbf{A}\left( \mathbf{Q}_{-\tilde{x}_i} - \frac{q_i\mathbf{Q}_{-\tilde{x}_i}\tilde{x}_i\tilde{x}_i^\top\mathbf{Q}_{-\tilde{x}_i}}{N(1+\delta_S)} \right) \right] \\
&= \frac{1}{1+\delta_S}\mathbb{E}[q_i\mathbf{Q}_{-\tilde{x}_i}\tilde{x}_i\tilde{x}_i^\top\bar{\mathbf{Q}}\mathbf{A}\mathbf{Q}_{-\tilde{x}_i}] - \frac{1}{N(1+\delta_S)^2}\mathbb{E}[q_i\mathbf{Q}_{-\tilde{x}_i}\tilde{x}_i\tilde{x}_i^\top\bar{\mathbf{Q}}\mathbf{A}\mathbf{Q}_{-\tilde{x}_i}\tilde{x}_i\tilde{x}_i^\top\mathbf{Q}_{-\tilde{x}_i}] \\
&= \frac{\alpha}{1+\delta_S}\mathbb{E}[\mathbf{Q}\Sigma_\beta\bar{\mathbf{Q}}\mathbf{A}\mathbf{Q}] - \frac{\alpha}{N(1+\delta_S)^2}\text{Tr}(\Sigma_\beta\bar{\mathbf{Q}}\mathbf{A}\bar{\mathbf{Q}})\mathbb{E}[\mathbf{Q}_{-\tilde{x}_i}\tilde{x}_i\tilde{x}_i^\top\mathbf{Q}_{-x_i}] \\
&= \frac{\alpha}{1+\delta_S}\mathbb{E}[\mathbf{Q}\Sigma_\beta\bar{\mathbf{Q}}\mathbf{A}\mathbf{Q}] - \frac{\alpha}{N(1+\delta_S)^2}\text{Tr}(\Sigma_\beta\bar{\mathbf{Q}}\mathbf{A}\bar{\mathbf{Q}})\mathbb{E}[\mathbf{Q}\Sigma_\beta\mathbf{Q}]
\end{aligned}$$

Which concludes the proof by summing all these separate terms. $\square$

**Corollary A.6** (Trace identities). *Using the above lemma A.5, we get that:*

$$\frac{1}{N}\text{Tr}(\Sigma\mathbb{E}[\mathbf{Q}\Sigma\mathbf{Q}]) = \frac{(1+\delta)^2}{\pi h}\left(a_1(1-b_2) + a_2 b_1\right), \quad \frac{1}{N}\text{Tr}(\Sigma_\beta\mathbb{E}[\mathbf{Q}\Sigma\mathbf{Q}]) = \frac{(1+\delta_S)^2}{\alpha(1-\pi)h}b_1$$

*And in the case of isotropic covariance matrix:* $\mathbf{C} = \sigma^2 \mathbf{I}_p$:

$$\frac{1}{N} \operatorname{Tr}(\Sigma \mathbb{E}[\mathbf{Q}\Sigma\mathbf{Q}]) = \frac{\eta}{h\theta^2}(1 - b_2 + \sigma^2 b_1), \quad \frac{1}{N} \operatorname{Tr}(\Sigma_\beta \mathbb{E}[\mathbf{Q}\Sigma\mathbf{Q}]) = \frac{\sigma^2}{N} \operatorname{Tr}(\Sigma \mathbb{E}[\mathbf{Q}\Sigma\mathbf{Q}])$$

## A.3 RESOLVENT IDENTITIES

Let $\mathbf{Q}$ be the resolvent matrix defined in equation (6). Denote by $\mathbf{Q}_{-\boldsymbol{v}_i}$ the resolvent matrix obtained from the dataset $\mathbf{V}$ by removing the $i^{th}$ sample $\boldsymbol{v}_i$, i.e:

$$\mathbf{Q}_{-\boldsymbol{v}_i} = \left(\mathbf{Q}^{-1} - \frac{1}{N}\boldsymbol{v}_i \boldsymbol{v}_i^\top\right)^{-1}$$

Then, using lemma A.3, we have that:

$$\mathbf{Q} = \mathbf{Q}_{-\boldsymbol{v}_i} - \frac{\mathbf{Q}_{-\boldsymbol{v}_i} \frac{1}{N} \boldsymbol{v}_i \boldsymbol{v}_i^\top \mathbf{Q}_{-i}}{1 + \frac{1}{N}\boldsymbol{v}_i^\top \mathbf{Q}_{-\boldsymbol{v}_i} \boldsymbol{v}_i},$$

and,

$$\mathbf{Q}\boldsymbol{x}_i = \frac{\mathbf{Q}_{-\boldsymbol{x}_i}\boldsymbol{x}_i}{1 + \delta}, \quad \mathbf{Q}\tilde{\boldsymbol{x}}_i = \frac{\mathbf{Q}_{-\tilde{\boldsymbol{x}}_i}\tilde{\boldsymbol{x}}_i}{1 + \delta_S}, \tag{8}$$

where:

$$\delta = \frac{1}{N} \operatorname{Tr}(\Sigma\bar{\mathbf{Q}}) = \frac{1}{N} \operatorname{Tr}(\bar{\mathbf{Q}}), \quad \delta_S = (\phi(1 - \varepsilon) + \rho\varepsilon)\frac{1}{N} \operatorname{Tr}(\Sigma_\beta\bar{\mathbf{Q}}) = \frac{\alpha}{N} \operatorname{Tr}(\mathbf{C}\bar{\mathbf{Q}}) \tag{9}$$

Let us recall the expression of $\bar{\mathbf{Q}}$ defined in lemma A.4:

$$\bar{\mathbf{Q}} = \left(\frac{\pi(\boldsymbol{\mu}\boldsymbol{\mu}^\top + \mathbf{I}_p)}{1 + \delta} + \frac{\alpha(1 - \pi)(\boldsymbol{\mu}_\beta \boldsymbol{\mu}_\beta^\top + \mathbf{C})}{1 + \delta_S} + \gamma\mathbf{I}_p\right)^{-1}$$

$$= \left(\mathbf{A} + \mathbf{U}\mathbf{U}^\top\right)^{-1}$$

where:

$$\mathbf{A} = \frac{\alpha(1 - \pi)}{1 + \delta_S}\mathbf{C} + \left(\gamma + \frac{\pi}{1 + \delta}\right)\mathbf{I}_p, \quad \mathbf{U} = \left(\sqrt{\frac{\pi}{1 + \delta}}\boldsymbol{\mu}, \sqrt{\frac{\alpha(1 - \pi)}{1 + \delta_S}}\boldsymbol{\mu}_\beta\right) \tag{10}$$

Since $\mathbf{C}$ is symmetric and real valued, then it is diagonalizable, and can be written as:

$$\mathbf{C} = \mathbf{P}\mathbf{D}\mathbf{P}^\top$$

where: $\mathbf{P}^{-1} = \mathbf{P}^\top$ is the matrix containing the eigenvectors of $\mathbf{C}$ in its columns, and $\mathbf{D} = Diag((d_i)_{i=1}^p)$ the diagonal matrix of the eigenvalues of $\mathbf{C}$. Hence, $\mathbf{A}$ can be written as:

$$\mathbf{A} = \mathbf{P}\Delta\mathbf{P}^\top, \quad \Delta = Diag\left(\gamma + \frac{\pi}{1 + \delta} + \frac{\alpha(1 - \pi)}{1 + \delta_S}d_i\right)_{i=1}^p \tag{11}$$

And using Woodbury's identity in lemma A.2, we get that:

$$\bar{\mathbf{Q}} = \mathbf{A}^{-1} - \mathbf{A}^{-1}\mathbf{U}\left(\mathbf{I}_2 + \mathbf{U}^\top\mathbf{A}^{-1}\mathbf{U}\right)^{-1}\mathbf{U}^\top\mathbf{A}^{-1}$$

where: $\mathbf{A}^{-1} = \mathbf{P}\Delta^{-1}\mathbf{P}^\top$ and $\Delta^{-1} = Diag\left(\frac{1}{\gamma + \frac{\pi}{1 + \delta} + \frac{\alpha(1 - \pi)}{1 + \delta_S}d_i}\right)_{i=1}^p$.

Let $\mathbf{M} = \left(\mathbf{I}_2 + \mathbf{U}^\top\mathbf{A}^{-1}\mathbf{U}\right)^{-1}$, and denote by $M_{i,j}$ its coordinate in the $i^{\text{th}}$ row and $j^{\text{th}}$ column. We have that:

$$\bar{\mathbf{Q}} = \mathbf{A}^{-1} - \mathbf{A}^{-1}\mathbf{U}\mathbf{M}\mathbf{U}^\top\mathbf{A}^{-1}$$

$$= \mathbf{A}^{-1} - \mathbf{A}^{-1}\left(\zeta_1\boldsymbol{\mu}\boldsymbol{\mu}^\top + \zeta_2\boldsymbol{\mu}_\beta\boldsymbol{\mu}_\beta^\top + \zeta_3(\boldsymbol{\mu}\boldsymbol{\mu}_\beta^\top + \boldsymbol{\mu}_\beta\boldsymbol{\mu}^\top)\right)\mathbf{A}^{-1}$$

where:

$$\zeta_1 = \frac{\pi M_{1,1}}{1+\delta}, \quad \zeta_2 = \frac{\alpha(1-\pi)M_{2,2}}{1+\delta_S}, \quad \zeta_3 = \sqrt{\frac{\alpha\pi(1-\pi)}{(1+\delta)(1+\delta_S)}}M_{1,2}$$

Thus,

$$\bar{\mathbf{Q}} = \mathbf{A}^{-1} - \mathbf{A}^{-1}\left(\zeta_1\boldsymbol{\mu}\boldsymbol{\mu}^\top + \zeta_2\boldsymbol{\mu}_\beta\boldsymbol{\mu}_\beta^\top + \zeta_3(\boldsymbol{\mu}\boldsymbol{\mu}_\beta^\top + \boldsymbol{\mu}_\beta\boldsymbol{\mu}^\top)\right)\mathbf{A}^{-1} \tag{12}$$

We can further show that:

$$M_{1,1} = \frac{1}{\det(M^{-1})}\left(1 + \frac{\alpha(1-\pi)}{1+\delta_S}\boldsymbol{\mu}_\beta^\top\mathbf{A}^{-1}\boldsymbol{\mu}_\beta\right)$$

$$M_{1,2} = \frac{1}{\det(M^{-1})}\left(-\sqrt{\frac{\alpha\pi(1-\pi)}{(1+\delta)(1+\delta_S)}}\boldsymbol{\mu}^\top\mathbf{A}^{-1}\boldsymbol{\mu}_\beta\right)$$

$$M_{2,1} = \frac{1}{\det(M^{-1})}\left(-\sqrt{\frac{\alpha\pi(1-\pi)}{(1+\delta)(1+\delta_S)}}\boldsymbol{\mu}^\top\mathbf{A}^{-1}\boldsymbol{\mu}_\beta\right)$$

$$M_{2,2} = \frac{1}{\det(M^{-1})}\left(1 + \frac{\pi}{1+\delta}\boldsymbol{\mu}^\top\mathbf{A}^{-1}\boldsymbol{\mu}\right)$$

$$\det(M^{-1}) = \left(1 + \frac{\pi}{1+\delta}\boldsymbol{\mu}^\top\mathbf{A}^{-1}\boldsymbol{\mu}\right)\left(1 + \frac{\alpha(1-\pi)}{1+\delta_S}\boldsymbol{\mu}_\beta^\top\mathbf{A}^{-1}\boldsymbol{\mu}_\beta\right) - \frac{\alpha\pi(1-\pi)}{(1+\delta)(1+\delta_S)}(\boldsymbol{\mu}^\top\mathbf{A}^{-1}\boldsymbol{\mu}_\beta)^2$$

**Lemma A.7** (Delta). *The parameters $\delta$ and $\delta_S$ defined in equation 9, are given by the following identities:*

$$\delta = \frac{1}{N}\sum_{i=1}^p \frac{1}{\gamma + \frac{\pi}{1+\delta} + \frac{\alpha(1-\pi)}{1+\delta_S}d_i}, \quad \delta_S = \frac{\alpha}{N}\sum_{i=1}^p \frac{d_i}{\gamma + \frac{\pi}{1+\delta} + \frac{\alpha(1-\pi)}{1+\delta_S}d_i}$$

*where: $(d_i)_{i=1}^p$ are the eigenvalues of the covariance matrix $\mathbf{C}$.*

*Proof.* Let: $\mathbf{M} = \left(\mathbf{I}_2 + \mathbf{U}^\top\mathbf{A}^{-1}\mathbf{U}\right)^{-1}$, and denote by $M_{i,j}$ its coordinate in the $i^{\text{th}}$ row and $j^{\text{th}}$ column. We have that using the expression of $\bar{\mathbf{Q}}$ in equation 12:

$$\bar{\mathbf{Q}} = \mathbf{A}^{-1} - \mathbf{A}^{-1}\left(\zeta_1\boldsymbol{\mu}\boldsymbol{\mu}^\top + \zeta_2\boldsymbol{\mu}_\beta\boldsymbol{\mu}_\beta^\top + \zeta_3(\boldsymbol{\mu}\boldsymbol{\mu}_\beta^\top + \boldsymbol{\mu}_\beta\boldsymbol{\mu}^\top)\right)\mathbf{A}^{-1}$$

Then:

$$\delta = \frac{1}{N}\operatorname{Tr}(\bar{\mathbf{Q}})$$

$$= \frac{1}{N}\operatorname{Tr}(\mathbf{A}^{-1}) - \frac{1}{N}\operatorname{Tr}\left(\mathbf{A}^{-1}\left(\zeta_1\boldsymbol{\mu}\boldsymbol{\mu}^\top + \zeta_2\boldsymbol{\mu}_\beta\boldsymbol{\mu}_\beta^\top + \zeta_3(\boldsymbol{\mu}\boldsymbol{\mu}_\beta^\top + \boldsymbol{\mu}_\beta\boldsymbol{\mu}^\top)\right)\mathbf{A}^{-1}\right)$$

We have that, when $N \to \infty$:

$$\frac{1}{N}\operatorname{Tr}(\mathbf{A}^{-1}\boldsymbol{\mu}\boldsymbol{\mu}^\top\mathbf{A}^{-1}) = \frac{1}{N}\boldsymbol{\mu}^\top(\mathbf{A}^{-1})^2\boldsymbol{\mu} = \mathcal{O}(N^{-1})$$

since $\|\boldsymbol{\mu}\| = \mathcal{O}(N^{-1})$ by assumption 4.1. The same applies for $\boldsymbol{\mu}_\beta$. Thus:

$$\delta = \frac{1}{N}\operatorname{Tr}(\mathbf{A}^{-1}) - \frac{1}{N}\zeta_1\boldsymbol{\mu}^\top(\mathbf{A}^{-1})^2\boldsymbol{\mu} - \frac{1}{N}\zeta_2\boldsymbol{\mu}_\beta^\top(\mathbf{A}^{-1})^2\boldsymbol{\mu}_\beta - \frac{2}{N}\zeta_3\boldsymbol{\mu}^\top(\mathbf{A}^{-1})^2\boldsymbol{\mu}_\beta$$

$$= \frac{1}{N}\operatorname{Tr}(\mathbf{A}^{-1}) + \mathcal{O}(N^{-1})$$

$$= \frac{1}{N}\operatorname{Tr}(\Delta^{-1}) + \mathcal{O}(N^{-1})$$

$$= \frac{1}{N}\sum_{i=1}^p \frac{1}{\gamma + \frac{\pi}{1+\delta} + \frac{\alpha(1-\pi)}{1+\delta_S}d_i} + \mathcal{O}(N^{-1})$$

Hence we have the desired result for $\delta$ in the regime $N \gg 1$ which we considered in our assumption 4.1.

Similarly for $\delta_S$, we have that:

$$\frac{1}{\alpha}\delta_S = \frac{1}{n}\operatorname{Tr}(\mathbf{C}\bar{\mathbf{Q}})$$

$$= \frac{1}{N}\operatorname{Tr}(\mathbf{C}\mathbf{A}^{-1}) - \frac{1}{N}\zeta_1\boldsymbol{\mu}^\top\mathbf{A}^{-1}\mathbf{C}\mathbf{A}^{-1}\boldsymbol{\mu} - \frac{1}{N}\zeta_2\boldsymbol{\mu}_\beta^\top\mathbf{A}^{-1}\mathbf{C}\mathbf{A}^{-1}\boldsymbol{\mu}_\beta - \frac{2}{N}\zeta_3\boldsymbol{\mu}^\top\mathbf{A}^{-1}\mathbf{C}\mathbf{A}^{-1}\boldsymbol{\mu}_\beta$$

$$= \frac{1}{N}\operatorname{Tr}(\mathbf{C}\mathbf{A}^{-1}) + \mathcal{O}(N^{-1})$$

$$= \frac{1}{N}\operatorname{Tr}(\mathbf{D}\Delta^{-1}) + \mathcal{O}(N^{-1})$$

$$= \frac{1}{N}\sum_{i=1}^{p}\frac{d_i}{\gamma + \frac{\pi}{1+\delta} + \frac{\alpha(1-\pi)}{1+\delta_S}d_i} + \mathcal{O}(N^{-1})$$

Which concludes our proof. $\qquad\square$

Now let us compute the trace identities that will be useful in the next sections.

**Lemma A.8** (Trace identities). *We have the following trace identities:*

$$\frac{1}{N}\operatorname{Tr}((\Sigma\bar{\mathbf{Q}})^2) = \frac{1}{N}\sum_{i=1}^{p}\frac{1}{\left(\gamma + \frac{\pi}{1+\delta} + \frac{\alpha(1-\pi)}{1+\delta_S}d_i\right)^2}, \quad \frac{1}{N}\operatorname{Tr}((\Sigma_\beta\bar{\mathbf{Q}})^2) = \frac{1}{N}\sum_{i=1}^{p}\left(\frac{d_i}{\gamma + \frac{\pi}{1+\delta} + \frac{\alpha(1-\pi)}{1+\delta_S}d_i}\right)^2,$$

$$\frac{1}{N}\operatorname{Tr}(\Sigma\bar{\mathbf{Q}}\Sigma_\beta\bar{\mathbf{Q}}) = \frac{1}{N}\sum_{i=1}^{p}\frac{d_i}{\left(\gamma + \frac{\pi}{1+\delta} + \frac{\alpha(1-\pi)}{1+\delta_S}d_i\right)^2}$$

*Proof.* We have that:

$$\frac{1}{N}\operatorname{Tr}((\Sigma\bar{\mathbf{Q}})^2) = \frac{1}{N}\operatorname{Tr}\left((\boldsymbol{\mu}\boldsymbol{\mu}^\top + \mathbf{I}_p)\bar{\mathbf{Q}}(\boldsymbol{\mu}\boldsymbol{\mu}^\top + \mathbf{I}_p)\bar{\mathbf{Q}}\right)$$

$$= \frac{1}{N}\operatorname{Tr}(\bar{\mathbf{Q}}^2) + \mathcal{O}(N^{-1})$$

$$= \frac{1}{N}\operatorname{Tr}((\mathbf{A}^{-1})^2) + \mathcal{O}(N^{-1})$$

$$= \frac{1}{N}\operatorname{Tr}((\mathbf{P}\Delta^{-1}\mathbf{P}^\top)^2) + \mathcal{O}(N^{-1})$$

$$= \frac{1}{N}\operatorname{Tr}((\Delta^{-1})^2) + \mathcal{O}(N^{-1})$$

$$= \frac{1}{N}\sum_{i=1}^{p}\frac{1}{\left(\gamma + \frac{\pi}{1+\delta} + \frac{\alpha(1-\pi)}{1+\delta_S}d_i\right)^2} + \mathcal{O}(N^{-1})$$

Thus we demonstrated the first identity. For the second one, we have that:

$$\frac{1}{N}\operatorname{Tr}((\Sigma_\beta\bar{\mathbf{Q}})^2) = \frac{1}{N}\operatorname{Tr}\left((\boldsymbol{\mu}_\beta\boldsymbol{\mu}_\beta^\top + \mathbf{C})\bar{\mathbf{Q}}(\boldsymbol{\mu}_\beta\boldsymbol{\mu}_\beta^\top + \mathbf{C})\bar{\mathbf{Q}}\right)$$

$$= \frac{1}{N}\operatorname{Tr}((\mathbf{C}\bar{\mathbf{Q}})^2) + \mathcal{O}(N^{-1})$$

$$= \frac{1}{N}\operatorname{Tr}((\mathbf{C}\mathbf{A}^{-1})^2) + \mathcal{O}(N^{-1})$$

$$= \frac{1}{N}\operatorname{Tr}((\mathbf{D}\Delta^{-1})^2) + \mathcal{O}(N^{-1})$$

$$= \frac{1}{N}\sum_{i=1}^{p}\left(\frac{d_i}{\gamma + \frac{\pi}{1+\delta} + \frac{\alpha(1-\pi)}{1+\delta_S}d_i}\right)^2 + +\mathcal{O}(N^{-1})$$

And the same spirit of the proof applies to the last identity. $\qquad\square$

## B  RANDOM MATRIX ANALYSIS OF THE GENERAL MODEL

In the generalized model, we consider that the synthetic data follow the following distribution:

$$\tilde{x}_i \sim \mathcal{N}(\boldsymbol{\mu}_\beta, \mathbf{C}), \quad \boldsymbol{\mu}_\beta = \beta\boldsymbol{\mu} + \boldsymbol{\mu}^\perp \tag{13}$$

where $\beta \in \mathbb{R}$ defines the alignment of the synthetic mean with the mean of real data, and $\boldsymbol{\mu}^\perp$ is a vector orthogonal to $\boldsymbol{\mu}$.

Now we will analyze here the performance of the classifier given by equation (6), and prove a generalized theorem B.1 in the paper.

$$\boldsymbol{w}_q = \frac{1}{N}\mathbf{Q}(\gamma)\left(\mathbf{X}\boldsymbol{y} + \tilde{\mathbf{X}}\mathbf{D}(\boldsymbol{q})\tilde{\boldsymbol{y}}\right), \quad \mathbf{Q}(\gamma) = \left(\frac{1}{N}\mathbf{V}\mathbf{V}^\top + \gamma\mathbf{I}_p\right)^{-1}.$$

The performance of (6) are fully determined by the first two order moments: $\mathbb{E}[\boldsymbol{w}_q^\top \boldsymbol{x}]$ and $\mathbb{E}[(\boldsymbol{w}_q^\top \boldsymbol{x})^2]$.

### B.1  TEST EXPECTATION:

We have that:

$$\boldsymbol{w}_q = \frac{1}{N}\sum_{i=1}^n \mathbf{Q}y_i\boldsymbol{x}_i + \frac{1}{N}\sum_{i=1}^m \mathbf{Q}q_i\tilde{y}_i\tilde{\boldsymbol{x}}_i$$

Let $\boldsymbol{x} \sim \mathcal{N}((-1)^a\boldsymbol{\mu}, \mathbf{I}_p)$ be a test sample independent of all the training samples $(\boldsymbol{v}_i)_{i=1}^N$. Then:

$$\mathbb{E}[\boldsymbol{w}_q^\top \boldsymbol{x}] = \frac{1}{N}\sum_{i=1}^n \mathbb{E}[y_i\boldsymbol{x}_i^\top\mathbf{Q}\boldsymbol{x}] + \frac{1}{N}\sum_{i=1}^m \mathbb{E}[q_i\tilde{y}_i\tilde{\boldsymbol{x}}_i^\top\mathbf{Q}\boldsymbol{x}]$$

**First sum:**  We have that, using the same lemma A.3:

$$\frac{1}{N}\sum_{i=1}^n \mathbb{E}[y_i\boldsymbol{x}_i^\top\mathbf{Q}\boldsymbol{x}] = \frac{1}{N}\sum_{i=1}^n \frac{1}{1+\delta}\mathbb{E}[y_i\boldsymbol{x}_i^\top\mathbf{Q}_{-\boldsymbol{x}_i}\boldsymbol{x}]$$

$$= \frac{1}{N}\sum_{i=1}^n \frac{1}{1+\delta}\mathbb{E}[\boldsymbol{x}_i]^\top\mathbb{E}[\mathbf{Q}_{-\boldsymbol{x}_i}]\mathbb{E}[x]$$

$$= \frac{1}{N}\sum_{i=1}^n \frac{(-1)^a}{1+\delta}\boldsymbol{\mu}^\top\bar{\mathbf{Q}}\boldsymbol{\mu}$$

$$= \frac{(-1)^a\pi}{1+\delta}\boldsymbol{\mu}^\top\bar{\mathbf{Q}}\boldsymbol{\mu}$$

Thus,

$$\frac{1}{N}\sum_{i=1}^n \mathbb{E}[y_i\boldsymbol{x}_i^\top\mathbf{Q}\boldsymbol{x}] = \frac{(-1)^a\pi}{1+\delta}\boldsymbol{\mu}^\top\bar{\mathbf{Q}}\boldsymbol{\mu} \tag{14}$$

**Second sum:**  Using the same lemma A.3:

$$\frac{1}{N}\sum_{i=1}^m \mathbb{E}[q_i\tilde{y}_i\tilde{\boldsymbol{x}}_i^\top\mathbf{Q}\boldsymbol{x}] = \frac{1}{N}\sum_{i=1}^m \frac{1}{1+\delta_S}\mathbb{E}[q_i\tilde{y}_i\tilde{\boldsymbol{x}}_i^\top\mathbf{Q}_{-\tilde{\boldsymbol{x}}_i}\boldsymbol{x}]$$

$$= \frac{1}{N(1+\delta_S)}\sum_{i=1}^m \mathbb{E}[q_i\tilde{y}_i]\mathbb{E}[\tilde{\boldsymbol{x}}_i]^\top\mathbb{E}[\mathbf{Q}_{-\tilde{\boldsymbol{x}}_i}]\mathbb{E}[\boldsymbol{x}]$$

$$= \frac{(-1)^a}{N(1+\delta_S)}\sum_{i=1}^m \lambda\boldsymbol{\mu}_\beta^\top\bar{\mathbf{Q}}\boldsymbol{\mu}$$

$$= \frac{(-1)^a\lambda(1-\pi)}{1+\delta_S}\boldsymbol{\mu}_\beta^\top\bar{\mathbf{Q}}\boldsymbol{\mu}$$

where (here $y_i$ means the true label of $\tilde{\boldsymbol{x}}_i$):

$$\mathbb{E}[q_i \tilde{y}_i] = y_i \mathbb{P}[q_i = 1 \mid \tilde{y}_i = y_i] - y_i \mathbb{P}[q_i = 1 \mid \tilde{y}_i \neq y_i]$$
$$= y_i(\phi(1 - \varepsilon) - \rho\varepsilon) = \lambda y_i$$

Therefore,

$$\mathbb{E}[\boldsymbol{w}_q^\top \boldsymbol{x}] = (-1)^a \left( \frac{\pi}{1 + \delta} \boldsymbol{\mu}^\top + \frac{\lambda(1 - \pi)}{1 + \delta_S} \boldsymbol{\mu}_\beta^\top \right) \bar{\mathbf{Q}} \boldsymbol{\mu} \tag{15}$$

## B.2 TEST VARIANCE:

To determine the variance of $\boldsymbol{w}_q^\top \boldsymbol{x}$, it only remains to compute its second order. We have that:

$$\mathbb{E}[(\boldsymbol{w}_q^\top \boldsymbol{x})^2] = \frac{1}{N^2} \mathbb{E}\left[ \left( \sum_{i=1}^n y_i \boldsymbol{x}_i^\top \mathbf{Q}\boldsymbol{x} + \sum_{j=1}^m q_j \tilde{y}_j \tilde{\boldsymbol{x}}_j^\top \mathbf{Q}\boldsymbol{x} \right)^2 \right]$$

$$= \frac{1}{N^2} \mathbb{E}\left[ \left( \sum_{i=1}^n y_i \boldsymbol{x}_i^\top \mathbf{Q}\boldsymbol{x} \right)^2 \right] + \frac{1}{N^2} \mathbb{E}\left[ \left( \sum_{j=1}^m q_j \tilde{y}_j \tilde{\boldsymbol{x}}_j^\top \mathbf{Q}\boldsymbol{x} \right)^2 \right] + \frac{2}{N^2} \mathbb{E}\left[ \left( \sum_{i=1}^n y_i \boldsymbol{x}_i^\top \mathbf{Q}\boldsymbol{x} \right) \left( \sum_{j=1}^m q_j \tilde{y}_j \tilde{\boldsymbol{x}}_j^\top \mathbf{Q}\boldsymbol{x} \right) \right]$$

Let us compute each sum on its own, and then group the results at the end.

**First sum:** We have that:

$$\frac{1}{N^2} \mathbb{E}\left[ \left( \sum_{i=1}^n y_i \boldsymbol{x}_i^\top \mathbf{Q}\boldsymbol{x} \right)^2 \right] = \frac{1}{N^2} \sum_{i=1}^n \sum_{k=1}^n \mathbb{E}[y_i y_j \boldsymbol{x}_i^\top \mathbf{Q}\boldsymbol{x}\boldsymbol{x}_k^\top \mathbf{Q}\boldsymbol{x}]$$

$$= \frac{1}{N^2} \sum_{i \neq k}^n \mathbb{E}[y_i y_k \boldsymbol{x}_i^\top \mathbf{Q}\boldsymbol{x}\boldsymbol{x}_k^\top \mathbf{Q}\boldsymbol{x}] + \frac{1}{N^2} \sum_{i=1}^n \mathbb{E}[\boldsymbol{x}_i^\top \mathbf{Q}\boldsymbol{x}\boldsymbol{x}_i^\top \mathbf{Q}\boldsymbol{x}]$$

- For $i \neq k$, we have that:

$$\mathbb{E}[y_i y_k \boldsymbol{x}_i^\top \mathbf{Q}\boldsymbol{x}\boldsymbol{x}_k^\top \mathbf{Q}\boldsymbol{x}] = \mathbb{E}[y_i y_k \boldsymbol{x}_i^\top \mathbf{Q}\boldsymbol{x}\boldsymbol{x}^\top \mathbf{Q}\boldsymbol{x}_k]$$
$$= \mathbb{E}[y_i y_k \boldsymbol{x}_i^\top \mathbf{Q}\Sigma\mathbf{Q}\boldsymbol{x}_k]$$
$$= \frac{1}{(1 + \delta)^2} \mathbb{E}[y_i y_k \boldsymbol{x}_i^\top \mathbf{Q}_{-\boldsymbol{x}_i} \Sigma \mathbf{Q}_{-\boldsymbol{x}_k} \boldsymbol{x}_k]$$
$$= \frac{1}{(1 + \delta)^2} \mathbb{E}\left[ y_i y_k \boldsymbol{x}_i^\top \left( \mathbf{Q}_{-\boldsymbol{x}_{i,k}} - \frac{\frac{1}{N}\mathbf{Q}_{-\boldsymbol{x}_{i,k}} \boldsymbol{x}_k \boldsymbol{x}_k^\top \mathbf{Q}_{-\boldsymbol{x}_{i,k}}}{1 + \delta} \right) \Sigma \left( \mathbf{Q}_{-\boldsymbol{x}_{i,k}} - \frac{\frac{1}{N}\mathbf{Q}_{-\boldsymbol{x}_{i,k}} \boldsymbol{x}_i \boldsymbol{x}_i^\top \mathbf{Q}_{-\boldsymbol{x}_{i,k}}}{1 + \delta} \right) \boldsymbol{x}_k \right]$$
$$= \frac{1}{(1 + \delta)^2} (A_1 - A_2 - A_3 + A_4)$$

And we have that:

$$A_1 = \mathbb{E}[y_i y_k \boldsymbol{x}_i^\top \mathbf{Q}_{-\boldsymbol{x}_{i,k}} \Sigma \mathbf{Q}_{-\boldsymbol{x}_{i,k}} \boldsymbol{x}_k]$$
$$= \boldsymbol{\mu}^\top \mathbb{E}[\mathbf{Q}\Sigma\mathbf{Q}]\boldsymbol{\mu}$$
$$= \boldsymbol{\mu}^\top \left( \frac{1 - b_2}{h} \bar{\mathbf{Q}}\Sigma\bar{\mathbf{Q}} + \frac{b_1}{h} \bar{\mathbf{Q}}\Sigma_\beta\bar{\mathbf{Q}} \right) \boldsymbol{\mu}$$

And:

$$A_2 = \frac{1}{N(1 + \delta)} \mathbb{E}[y_i y_k \boldsymbol{x}_i^\top \mathbf{Q}_{-\boldsymbol{x}_{i,k}} \boldsymbol{x}_k \boldsymbol{x}_k^\top \mathbf{Q}_{-\boldsymbol{x}_{i,k}} \Sigma \mathbf{Q}_{-\boldsymbol{x}_{i,k}} \boldsymbol{x}_k]$$
$$= \frac{\text{Tr}(\Sigma\mathbb{E}[\mathbf{Q}\Sigma\mathbf{Q}])}{N(1 + \delta)} \boldsymbol{\mu}^\top \bar{\mathbf{Q}} \boldsymbol{\mu}$$

Since, by concentration laws:

$$\frac{1}{N}\boldsymbol{x}_k^\top \mathbf{Q}_{-\boldsymbol{x}_{i,k}}\Sigma\mathbf{Q}_{-\boldsymbol{x}_{i,k}}\boldsymbol{x}_k = \frac{1}{N}\mathbb{E}[\boldsymbol{x}_k^\top \mathbf{Q}_{-\boldsymbol{x}_{i,k}}\Sigma\mathbf{Q}_{-\boldsymbol{x}_{i,k}}\boldsymbol{x}_k]$$

$$= \frac{1}{N}\mathbb{E}[\mathrm{Tr}(\boldsymbol{x}_k\boldsymbol{x}_k^\top \mathbf{Q}_{-\boldsymbol{x}_{i,k}}\Sigma\mathbf{Q}_{-\boldsymbol{x}_{i,k}})]$$

$$= \frac{1}{N}\mathrm{Tr}(\mathbb{E}[\boldsymbol{x}_k\boldsymbol{x}_k^\top \mathbf{Q}_{-\boldsymbol{x}_{i,k}}\Sigma\mathbf{Q}_{-\boldsymbol{x}_{i,k}}])$$

$$= \frac{1}{N}\mathrm{Tr}(\Sigma\mathbb{E}[\mathbf{Q}_{-\boldsymbol{x}_{i,k}}\Sigma\mathbf{Q}_{-\boldsymbol{x}_{i,k}}])$$

$$= \frac{1}{N}\mathrm{Tr}(\Sigma\mathbb{E}[\mathbf{Q}\Sigma\mathbf{Q}])$$

And we can easily verify that:

$$A_3 = A_2, \quad A_4 = \mathcal{O}(N^{-1})$$

Thus,

$$\frac{1}{N^2}\sum_{i\neq k}^n \mathbb{E}[y_i y_k \boldsymbol{x}_i^\top \mathbf{Q}\boldsymbol{x}\boldsymbol{x}_k^\top \mathbf{Q}\boldsymbol{x}] = \frac{n^2-n}{N^2}\left(\boldsymbol{\mu}^\top \mathbb{E}[\mathbf{Q}\Sigma\mathbf{Q}]\boldsymbol{\mu} - \frac{2\,\mathrm{Tr}(\Sigma\mathbb{E}[\mathbf{Q}\Sigma\mathbf{Q}])}{N(1+\delta)}\boldsymbol{\mu}^\top \bar{\mathbf{Q}}\boldsymbol{\mu}\right)$$

$$= \frac{\pi^2}{(1+\delta)^2}\left(\boldsymbol{\mu}^\top \mathbb{E}[\mathbf{Q}\Sigma\mathbf{Q}]\boldsymbol{\mu} - \frac{2\,\mathrm{Tr}(\Sigma\mathbb{E}[\mathbf{Q}\Sigma\mathbf{Q}])}{N(1+\delta)}\boldsymbol{\mu}^\top \bar{\mathbf{Q}}\boldsymbol{\mu}\right)$$

- And then, for $i \in \{1,...,n\}$:

$$\mathbb{E}[\boldsymbol{x}_i^\top \mathbf{Q}\boldsymbol{x}\boldsymbol{x}_i^\top \mathbf{Q}\boldsymbol{x}] = \mathbb{E}[\boldsymbol{x}_i^\top \mathbf{Q}\Sigma\mathbf{Q}\boldsymbol{x}_i]$$

$$= \frac{1}{(1+\delta)^2}\mathbb{E}[\boldsymbol{x}_i^\top \mathbf{Q}_{-\boldsymbol{x}_i}\Sigma\mathbf{Q}_{-\boldsymbol{x}_i}\boldsymbol{x}_i]$$

$$= \frac{1}{(1+\delta)^2}\mathrm{Tr}(\mathbb{E}[\boldsymbol{x}_i\boldsymbol{x}_i^\top \mathbf{Q}_{-\boldsymbol{x}_i}\Sigma\mathbf{Q}_{-\boldsymbol{x}_i}])$$

$$= \frac{1}{(1+\delta)^2}\mathrm{Tr}(\Sigma\mathbb{E}[\mathbf{Q}\Sigma\mathbf{Q}])$$

Thus:

$$\frac{1}{N^2}\sum_{i=1}^n \mathbb{E}[\boldsymbol{x}_i^\top \mathbf{Q}\boldsymbol{x}\boldsymbol{x}_i^\top \mathbf{Q}\boldsymbol{x}] = \frac{\pi}{N(1+\delta)^2}\mathrm{Tr}(\Sigma\mathbb{E}[\mathbf{Q}\Sigma\mathbf{Q}])$$

Hence, the first sum gives us:

$$\frac{1}{N^2}\mathbb{E}\left[\left(\sum_{i=1}^n y_i \boldsymbol{x}_i^\top \mathbf{Q}\boldsymbol{x}\right)^2\right] = \frac{\pi^2}{(1+\delta)^2}\left(\boldsymbol{\mu}^\top \mathbb{E}[\mathbf{Q}\Sigma\mathbf{Q}]\boldsymbol{\mu} - \frac{2\,\mathrm{Tr}(\Sigma\mathbb{E}[\mathbf{Q}\Sigma\mathbf{Q}])}{N(1+\delta)}\boldsymbol{\mu}^\top \bar{\mathbf{Q}}\boldsymbol{\mu}\right) + \frac{\pi}{N(1+\delta)^2}\mathrm{Tr}(\Sigma\mathbb{E}[\mathbf{Q}\Sigma\mathbf{Q}])$$

$$(16)$$

**Second sum:** We have that:

$$\frac{1}{N^2}\mathbb{E}\left[\left(\sum_{i=1}^m q_i \tilde{y}_i \tilde{\boldsymbol{x}}_i^\top \mathbf{Q}\boldsymbol{x}\right)^2\right] = \frac{1}{N^2}\sum_{i,j=1}^m \mathbb{E}[q_i q_j \tilde{y}_i \tilde{y}_j \tilde{\boldsymbol{x}}_i^\top \mathbf{Q}\boldsymbol{x}\tilde{\boldsymbol{x}}_j^\top \mathbf{Q}\boldsymbol{x}]$$

$$= \frac{1}{N^2}\sum_{i\neq j}\mathbb{E}[q_i q_j \tilde{y}_i \tilde{y}_j \tilde{\boldsymbol{x}}_i^\top \mathbf{Q}\boldsymbol{x}\tilde{\boldsymbol{x}}_j^\top \mathbf{Q}\boldsymbol{x}] + \frac{1}{N^2}\sum_{i=1}^m \mathbb{E}[q_i \tilde{\boldsymbol{x}}_i^\top \boldsymbol{x}\tilde{\boldsymbol{x}}_i^\top \mathbf{Q}\boldsymbol{x}]$$

- For $i \neq j \in \{1,...,m\}$, we have that:

$$\mathbb{E}[q_i q_j \tilde{y}_i \tilde{y}_j \tilde{\boldsymbol{x}}_i^\top \mathbf{Q}\boldsymbol{x}\tilde{\boldsymbol{x}}_j^\top \mathbf{Q}\boldsymbol{x}] = \mathbb{E}[q_i q_j \tilde{y}_i \tilde{y}_j \tilde{\boldsymbol{x}}_i^\top \mathbf{Q}\Sigma\mathbf{Q}\boldsymbol{x}_j]$$

$$= \frac{1}{(1+\delta_S)^2}\mathbb{E}[q_i q_j \tilde{y}_i \tilde{y}_j \tilde{\boldsymbol{x}}_i^\top \mathbf{Q}_{-\tilde{\boldsymbol{x}}_i}\Sigma\mathbf{Q}_{-\tilde{\boldsymbol{x}}_j}\boldsymbol{x}_j]$$

$$= \frac{1}{(1+\delta_S)^2}\mathbb{E}\left[q_i q_j \tilde{y}_i \tilde{y}_j \tilde{\boldsymbol{x}}_i^\top \left(\mathbf{Q}_{-\tilde{\boldsymbol{x}}_{i,j}} - \frac{\frac{1}{N}\mathbf{Q}_{-\tilde{\boldsymbol{x}}_{i,j}}q_j \tilde{\boldsymbol{x}}_j \tilde{\boldsymbol{x}}_j^\top \mathbf{Q}_{-\tilde{\boldsymbol{x}}_{i,j}}}{1+\delta_S}\right)\Sigma\left(\mathbf{Q}_{-\tilde{\boldsymbol{x}}_{i,j}} - \frac{\frac{1}{N}\mathbf{Q}_{-\tilde{\boldsymbol{x}}_{i,j}}q_i \tilde{\boldsymbol{x}}_i \tilde{\boldsymbol{x}}_i^\top \mathbf{Q}_{-\tilde{\boldsymbol{x}}_{i,j}}}{1+\delta_S}\right)\tilde{\boldsymbol{x}}_j\right]$$

$$= \frac{1}{(1+\delta_S)^2}(A_1 - A_2 - A_3 + A_4)$$

And, we have that:

$$A_1 = \mathbb{E}[q_i q_j \tilde{y}_i \tilde{y}_j \tilde{\boldsymbol{x}}_i^\top \mathbf{Q}_{-\tilde{\boldsymbol{x}}_{i,j}} \Sigma \mathbf{Q}_{-\tilde{\boldsymbol{x}}_{i,j}} \tilde{\boldsymbol{x}}_j]$$
$$= \lambda^2 \mathbb{E}[y_i \tilde{\boldsymbol{x}}_i^\top \mathbf{Q}_{-\tilde{\boldsymbol{x}}_{i,j}} \Sigma \mathbf{Q}_{-\tilde{\boldsymbol{x}}_{i,j}} y_j \tilde{\boldsymbol{x}}_j]$$
$$= \lambda^2 \boldsymbol{\mu}_\beta^\top \mathbb{E}[\mathbf{Q}\Sigma\mathbf{Q}] \boldsymbol{\mu}_\beta$$

And:

$$A_2 = \frac{1}{N(1+\delta_S)} \mathbb{E}[q_i q_j \tilde{y}_i \tilde{y}_j \tilde{\boldsymbol{x}}_i^\top \mathbf{Q}_{-\tilde{\boldsymbol{x}}_{i,j}} \tilde{\boldsymbol{x}}_j \tilde{\boldsymbol{x}}_j^\top \mathbf{Q}_{-\tilde{\boldsymbol{x}}_{i,j}} \Sigma \mathbf{Q}_{-\tilde{\boldsymbol{x}}_{i,j}} \tilde{\boldsymbol{x}}_j]$$
$$= \frac{1}{N(1+\delta_S)} \operatorname{Tr}(\Sigma_\beta \mathbb{E}[\mathbf{Q}\Sigma\mathbf{Q}]) \mathbb{E}[q_i q_j \tilde{y}_i \tilde{y}_j \tilde{\boldsymbol{x}}_i^\top \mathbf{Q}_{-\tilde{\boldsymbol{x}}_{i,j}} \tilde{\boldsymbol{x}}_j]$$
$$= \frac{\lambda^2}{N(1+\delta_S)} \operatorname{Tr}(\Sigma_\beta \mathbb{E}[\mathbf{Q}\Sigma\mathbf{Q}]) \boldsymbol{\mu}_\beta^\top \bar{\mathbf{Q}} \boldsymbol{\mu}_\beta$$

And, we can easily observe that:

$$A_3 = A_2, \quad A_4 = \mathcal{O}(N^{-1})$$

Thus:

$$\frac{1}{N^2} \sum_{i \neq j} \mathbb{E}[q_i q_j \tilde{y}_i \tilde{y}_j \tilde{\boldsymbol{x}}_i^\top \mathbf{Q}\boldsymbol{x} \tilde{\boldsymbol{x}}_j^\top \mathbf{Q}\boldsymbol{x}] = \frac{\lambda^2 (1-\pi)^2}{(1+\delta_S)^2} \left( \boldsymbol{\mu}_\beta^\top \mathbb{E}[\mathbf{Q}\Sigma\mathbf{Q}] \boldsymbol{\mu}_\beta - \frac{2}{N(1+\delta_S)} \operatorname{Tr}(\Sigma_\beta \mathbb{E}[\mathbf{Q}\Sigma\mathbf{Q}]) \boldsymbol{\mu}_\beta^\top \bar{\mathbf{Q}} \boldsymbol{\mu}_\beta \right)$$

- And for $i \in \{1, ..., m\}$:

$$\mathbb{E}[q_i \tilde{\boldsymbol{x}}_i^\top \mathbf{Q}\boldsymbol{x} \tilde{\boldsymbol{x}}_i^\top \mathbf{Q}\boldsymbol{x}] = \frac{1}{(1+\delta_S)^2} \mathbb{E}[q_i \tilde{\boldsymbol{x}}_i^\top \mathbf{Q}_{-\tilde{\boldsymbol{x}}_i} \Sigma \mathbf{Q}_{-\tilde{\boldsymbol{x}}_i} \tilde{\boldsymbol{x}}_i]$$
$$= \frac{\alpha}{(1+\delta_S)^2} \mathbb{E}[\tilde{\boldsymbol{x}}_i^\top \mathbf{Q}_{-\tilde{\boldsymbol{x}}_i} \Sigma \mathbf{Q}_{-\tilde{\boldsymbol{x}}_i} \tilde{\boldsymbol{x}}_i]$$
$$= \frac{\alpha}{(1+\delta_S)^2} \operatorname{Tr}(\mathbb{E}[\tilde{\boldsymbol{x}}_i \tilde{\boldsymbol{x}}_i^\top] \mathbb{E}[\mathbf{Q}_{-\tilde{\boldsymbol{x}}_i} \Sigma \mathbf{Q}_{-\tilde{\boldsymbol{x}}_i}])$$
$$= \frac{\alpha}{(1+\delta_S)^2} \operatorname{Tr}(\Sigma_\beta \mathbb{E}[\mathbf{Q}\Sigma\mathbf{Q}])$$

Hence, by grouping the terms, the second sum gives us:

$$\frac{1}{N^2} \mathbb{E}\left[ \left( \sum_{i=1}^m q_i \tilde{y}_i \tilde{\boldsymbol{x}}_i^\top \mathbf{Q}\boldsymbol{x} \right)^2 \right] \tag{17}$$

$$= \frac{\lambda^2 (1-\pi)^2}{(1+\delta_S)^2} \left( \boldsymbol{\mu}_\beta^\top \mathbb{E}[\mathbf{Q}\Sigma\mathbf{Q}] \boldsymbol{\mu}_\beta - \frac{2}{N(1+\delta_S)} \operatorname{Tr}(\Sigma_\beta \mathbb{E}[\mathbf{Q}\Sigma\mathbf{Q}]) \boldsymbol{\mu}_\beta^\top \bar{\mathbf{Q}} \boldsymbol{\mu}_\beta \right) + \frac{\alpha(1-\pi)}{N(1+\delta_S)^2} \operatorname{Tr}(\Sigma_\beta \mathbb{E}[\mathbf{Q}\Sigma\mathbf{Q}]) \tag{18}$$

**Third sum:** Let us now compute the remaining term in the sum that is given by:

$$\frac{2}{N^2} \sum_{i=1}^n \sum_{j=1}^m \mathbb{E}[y_i \boldsymbol{x}_i^\top \mathbf{Q}\boldsymbol{x} q_j \tilde{y}_j \tilde{\boldsymbol{x}}_j^\top \mathbf{Q}\boldsymbol{x}]$$

Let $i \in \{1, ..., n\}$ and $j \in \{1, ..., m\}$, we have that:

$$\mathbb{E}[y_i q_j \tilde{y}_j \boldsymbol{x}_i^\top \mathbf{Q}\boldsymbol{x}\boldsymbol{x}^\top \mathbf{Q}\tilde{\boldsymbol{x}}_j] = \mathbb{E}[y_i q_j \tilde{y}_j \boldsymbol{x}_i^\top \mathbf{Q}\Sigma\mathbf{Q}\tilde{\boldsymbol{x}}_j]$$
$$= \frac{1}{(1+\delta)(1+\delta_S)} \mathbb{E}[y_i q_j \tilde{y}_j \boldsymbol{x}_i^\top \mathbf{Q}_{-\boldsymbol{x}_i} \Sigma \mathbf{Q}_{-\tilde{\boldsymbol{x}}_j} \tilde{\boldsymbol{x}}_j]$$
$$= \frac{1}{(1+\delta)(1+\delta_S)} \mathbb{E}\left[ y_i q_j \tilde{y}_j \boldsymbol{x}_i^\top \left( \mathbf{Q}_{-ij} - \frac{\mathbf{Q}_{-ij} q_j \tilde{\boldsymbol{x}}_j \tilde{\boldsymbol{x}}_j^\top \mathbf{Q}_{-ij}}{N(1+\delta_S)} \right) \Sigma \left( \mathbf{Q}_{-ij} - \frac{\mathbf{Q}_{-ij} \boldsymbol{x}_i \boldsymbol{x}_i^\top \mathbf{Q}_{-ij}}{N(1+\delta)} \right) \tilde{\boldsymbol{x}}_j \right]$$
$$= \frac{1}{(1+\delta)(1+\delta_S)} (A_1 - A_2 - A_3 + A_4)$$

We have that:

$$A_1 = \mathbb{E}[y_i q_j \tilde{y}_j \boldsymbol{x}_i^\top \mathbf{Q}_{-ij} \Sigma \mathbf{Q}_{-ij} \tilde{\boldsymbol{x}}_j] = \lambda \mathbb{E}[y_i \boldsymbol{x}_i^\top \mathbf{Q}_{-ij} \Sigma \mathbf{Q}_{-ij} \tilde{y}_j \boldsymbol{x}_j]$$
$$= \lambda \boldsymbol{\mu}^\top \mathbb{E}[\mathbf{Q}\Sigma\mathbf{Q}] \boldsymbol{\mu}_\beta$$

And:

$$A_2 = \frac{1}{N(1+\delta)} \mathbb{E}[y_i q_j \tilde{y}_j \boldsymbol{x}_i^\top \mathbf{Q}_{-ij} \Sigma \mathbf{Q}_{-ij} \boldsymbol{x}_i \boldsymbol{x}_i^\top \mathbf{Q}_{-ij} \tilde{\boldsymbol{x}}_j]$$
$$= \frac{\lambda}{N(1+\delta)} \mathbb{E}[y_i \boldsymbol{x}_i^\top \mathbf{Q}_{-ij} \Sigma \mathbf{Q}_{-ij} \boldsymbol{x}_i \boldsymbol{x}_i^\top \mathbf{Q}_{-ij} y_j \tilde{\boldsymbol{x}}_j]$$
$$= \frac{\lambda}{N(1+\delta)} \operatorname{Tr}(\Sigma\mathbb{E}[\mathbf{Q}\Sigma\mathbf{Q}]) \boldsymbol{\mu}^\top \bar{\mathbf{Q}} \boldsymbol{\mu}_\beta$$

And also:

$$A_3 = \frac{1}{N(1+\delta_S)} \mathbb{E}[y_i q_j \tilde{y}_j \boldsymbol{x}_i^\top \mathbf{Q}_{-ij} \tilde{\boldsymbol{x}}_j \tilde{\boldsymbol{x}}_j^\top \mathbf{Q}_{-ij} \Sigma \mathbf{Q}_{-ij} \tilde{\boldsymbol{x}}_j]$$
$$= \frac{\lambda}{N(1+\delta_S)} \operatorname{Tr}(\Sigma_\beta \mathbb{E}[\mathbf{Q}\Sigma\mathbf{Q}]) \boldsymbol{\mu}^\top \bar{\mathbf{Q}} \boldsymbol{\mu}_\beta$$

Hence:

$$\frac{2}{N^2} \sum_{i=1}^n \sum_{j=1}^m \mathbb{E}[y_i \boldsymbol{x}_i^\top \mathbf{Q} \boldsymbol{x} q_j \tilde{y}_j \tilde{\boldsymbol{x}}_j^\top \mathbf{Q} \boldsymbol{x}] \tag{19}$$
$$= \frac{2\lambda\pi(1-\pi)}{(1+\delta)(1+\delta_S)} \left( \boldsymbol{\mu}^\top \mathbb{E}[\mathbf{Q}\Sigma\mathbf{Q}] \boldsymbol{\mu}_\beta - \frac{1}{N} \operatorname{Tr}\left( \left( \frac{\Sigma}{1+\delta} + \frac{\Sigma_\beta}{1+\delta_S} \right) \mathbb{E}[\mathbf{Q}\Sigma\mathbf{Q}] \right) \boldsymbol{\mu}^\top \bar{\mathbf{Q}} \boldsymbol{\mu}_\beta \right) \tag{20}$$

**Grouping all the sums:** Denote by : $T_1 = \frac{1}{N} \operatorname{Tr}(\Sigma\mathbb{E}[\mathbf{Q}\Sigma\mathbf{Q}])$, then: $T_2 = \frac{1}{N} \operatorname{Tr}(\Sigma_\beta\mathbb{E}[\mathbf{Q}\Sigma\mathbf{Q}])$.
Now let us group the terms in $T$ in the three sums, and those that do not depend on $T$. We get that:

$$\mathbb{E}[(\boldsymbol{w}^\top \boldsymbol{x})^2] = \frac{\pi^2}{(1+\delta)^2} \boldsymbol{\mu}^\top \mathbb{E}[\mathbf{Q}\Sigma\mathbf{Q}] \boldsymbol{\mu} + \frac{\lambda^2(1-\pi)^2}{(1+\delta_S)^2} \boldsymbol{\mu}_\beta^\top \mathbb{E}[\mathbf{Q}\Sigma\mathbf{Q}] \boldsymbol{\mu}_\beta + \frac{2\lambda\pi(1-\pi)}{(1+\delta)(1+\delta_S)} \boldsymbol{\mu}^\top \mathbb{E}[\mathbf{Q}\Sigma\mathbf{Q}] \boldsymbol{\mu}_\beta$$

$$+ T_1 \left( \frac{\pi}{(1+\delta)^2} - \frac{2\pi^2}{(1+\delta)^3} \boldsymbol{\mu}^\top \bar{\mathbf{Q}} \boldsymbol{\mu} - \frac{2\lambda\pi(1-\pi)}{(1+\delta)^2(1+\delta_S)} \boldsymbol{\mu}^\top \bar{\mathbf{Q}} \boldsymbol{\mu}_\beta \right)$$

$$+ T_2 \left( \frac{\alpha(1-\pi)}{(1+\delta_S)^2} - \frac{2\lambda^2(1-\pi)^2}{(1+\delta_S)^3} \boldsymbol{\mu}_\beta^\top \bar{\mathbf{Q}} \boldsymbol{\mu}_\beta - \frac{2\lambda\pi(1-\pi)}{(1+\delta)(1+\delta_S)^2} \boldsymbol{\mu}^\top \bar{\mathbf{Q}} \boldsymbol{\mu}_\beta \right)$$

$$= \frac{\pi^2}{(1+\delta)^2} \boldsymbol{\mu}^\top \mathbb{E}[\mathbf{Q}\Sigma\mathbf{Q}] \boldsymbol{\mu} + \frac{\lambda^2(1-\pi)^2}{(1+\delta_S)^2} \boldsymbol{\mu}_\beta^\top \mathbb{E}[\mathbf{Q}\Sigma\mathbf{Q}] \boldsymbol{\mu}_\beta + \frac{2\lambda\pi(1-\pi)}{(1+\delta)(1+\delta_S)} \boldsymbol{\mu}^\top \mathbb{E}[\mathbf{Q}\Sigma\mathbf{Q}] \boldsymbol{\mu}_\beta$$

$$+ \frac{\pi T_1}{(1+\delta)^2} \left( 1 - \frac{2\pi}{1+\delta} \boldsymbol{\mu}^\top \bar{\mathbf{Q}} \boldsymbol{\mu} - \frac{2\lambda(1-\pi)}{1+\delta_S} \boldsymbol{\mu}^\top \bar{\mathbf{Q}} \boldsymbol{\mu}_\beta \right)$$

$$+ \frac{(1-\pi)T_2}{(1+\delta_S)^2} \left( \alpha - \frac{2\lambda^2(1-\pi)}{1+\delta_S} \boldsymbol{\mu}_\beta^\top \bar{\mathbf{Q}} \boldsymbol{\mu}_\beta - \frac{2\lambda\pi}{1+\delta} \boldsymbol{\mu}^\top \bar{\mathbf{Q}} \boldsymbol{\mu}_\beta \right)$$

This leads to the following theorem:

**Theorem B.1** (Gaussianity of the General model)**.** *Let $\boldsymbol{w}_q$ be the Mixed classifier as defined in equation 6 and suppose that Assumption 4.1 holds. The decision function $\boldsymbol{w}_q^\top \boldsymbol{x}$, on some test sample $\boldsymbol{x} \in \mathcal{C}_a$ independent of $\mathbf{X}$, satisfies:*

$$\boldsymbol{w}_q^\top \boldsymbol{x} \xrightarrow{\mathcal{D}} \mathcal{N}\left( (-1)^a m_q, \nu_q - m_q^2 \right),$$

*where:*

$$m_q = \left( \frac{\pi}{1+\delta} \boldsymbol{\mu}^\top + \frac{\lambda(1-\pi)}{1+\delta_S} \boldsymbol{\mu}_\beta^\top \right) \bar{\mathbf{Q}} \boldsymbol{\mu},$$

$$\nu_q = \frac{\pi^2}{(1+\delta)^2} \boldsymbol{\mu}^\top \mathbb{E}[\mathbf{Q}\Sigma\mathbf{Q}] \boldsymbol{\mu} + \frac{\lambda^2(1-\pi)^2}{(1+\delta_S)^2} \boldsymbol{\mu}_\beta^\top \mathbb{E}[\mathbf{Q}\Sigma\mathbf{Q}] \boldsymbol{\mu}_\beta + \frac{2\lambda\pi(1-\pi)}{(1+\delta)(1+\delta_S)} \boldsymbol{\mu}^\top \mathbb{E}[\mathbf{Q}\Sigma\mathbf{Q}] \boldsymbol{\mu}_\beta$$

$$+ \frac{\pi T_1}{(1+\delta)^2} \left( 1 - \frac{2\pi}{1+\delta} \boldsymbol{\mu}^\top \bar{\mathbf{Q}} \boldsymbol{\mu} - \frac{2\lambda(1-\pi)}{1+\delta_S} \boldsymbol{\mu}^\top \bar{\mathbf{Q}} \boldsymbol{\mu}_\beta \right)$$

$$+ \frac{(1-\pi)T_2}{(1+\delta_S)^2} \left( \alpha - \frac{2\lambda^2(1-\pi)}{1+\delta_S} \boldsymbol{\mu}_\beta^\top \bar{\mathbf{Q}} \boldsymbol{\mu}_\beta - \frac{2\lambda\pi}{1+\delta} \boldsymbol{\mu}^\top \bar{\mathbf{Q}} \boldsymbol{\mu}_\beta \right).$$

*where:*

$$T_1 = \frac{1}{N} \operatorname{Tr}(\Sigma \mathbb{E}[\mathbf{Q}\Sigma\mathbf{Q}]), \quad T_2 = \frac{1}{N} \operatorname{Tr}(\Sigma_\beta \mathbb{E}[\mathbf{Q}\Sigma\mathbf{Q}]), \quad \lambda = \phi(1-\varepsilon) - \rho\varepsilon$$

## C   PARTICULAR CASE: ISOTROPIC COVARIANCE MATRIX

Here, we consider a simple covariance matrix of the form $\mathbf{C} = \sigma^2 \mathbf{I}_p$ for some $\sigma > 0$. So

$$\delta_S = \alpha \sigma^2 \delta \tag{21}$$

### C.1   RESOLVENT IDENTITIES IN THE CASE OF $\mathbf{C} = \sigma^2 \mathbf{I}_p$

We have that by lemma A.4:

$$\bar{\mathbf{Q}} = \left( \frac{\pi}{1+\delta}(\boldsymbol{\mu}\boldsymbol{\mu}^\top + \mathbf{I}_p) + \frac{\alpha(1-\pi)}{1+\alpha\sigma^2\delta}(\boldsymbol{\mu}_\beta\boldsymbol{\mu}_\beta^\top + \sigma^2\mathbf{I}_p) + \gamma\mathbf{I}_p \right)^{-1}$$

$$= \left( \frac{\pi}{1+\delta}\boldsymbol{\mu}\boldsymbol{\mu}^\top + \frac{\alpha(1-\pi)}{1+\alpha\sigma^2\delta}\boldsymbol{\mu}_\beta\boldsymbol{\mu}_\beta^\top + \theta\mathbf{I}_p \right)^{-1},$$

where:

$$\theta = \gamma + \frac{\pi}{1+\delta} + \frac{\alpha\sigma^2(1-\pi)}{1+\alpha\sigma^2\delta} \tag{22}$$

Define by:

$$\mathbf{R}_1 = \left( \frac{\alpha(1-\pi)}{1+\alpha\sigma^2\delta}\boldsymbol{\mu}_\beta\boldsymbol{\mu}_\beta^\top + \theta\mathbf{I}_p \right)^{-1}, \quad \mathbf{R}_2 = \left( \frac{\pi}{1+\delta}\boldsymbol{\mu}\boldsymbol{\mu}^\top + \theta\mathbf{I}_p \right)^{-1}, \tag{23}$$

such that:

$$\bar{\mathbf{Q}} = \left( \frac{\pi}{1+\delta}\boldsymbol{\mu}\boldsymbol{\mu}^\top + \mathbf{R}_1^{-1} \right)^{-1} = \left( \frac{\alpha(1-\pi)}{1+\alpha\sigma^2\delta}\boldsymbol{\mu}_\beta\boldsymbol{\mu}_\beta^\top + \mathbf{R}_2^{-1} \right)^{-1}$$

Thus, using lemma A.3:

$$\bar{\mathbf{Q}}\boldsymbol{\mu} = \frac{\mathbf{R}_1\boldsymbol{\mu}}{1 + \frac{\pi}{1+\delta}\boldsymbol{\mu}^\top\mathbf{R}_1\boldsymbol{\mu}}, \quad \bar{\mathbf{Q}}\boldsymbol{\mu}_\beta = \frac{\mathbf{R}_2\boldsymbol{\mu}_\beta}{1 + \frac{\alpha(1-\pi)}{1+\alpha\sigma^2\delta}\boldsymbol{\mu}_\beta^\top\mathbf{R}_2\boldsymbol{\mu}_\beta} \tag{24}$$

And:

$$\bar{\mathbf{Q}} = \mathbf{R}_1 - \frac{\pi\mathbf{R}_1\boldsymbol{\mu}\boldsymbol{\mu}^\top\mathbf{R}_1}{1 + \delta + \pi\boldsymbol{\mu}^\top\mathbf{R}_1\boldsymbol{\mu}} = \mathbf{R}_2 - \frac{\alpha(1-\pi)\mathbf{R}_2\boldsymbol{\mu}_\beta\boldsymbol{\mu}_\beta^\top\mathbf{R}_2}{1 + \alpha\sigma^2\delta + \alpha(1-\pi)\boldsymbol{\mu}_\beta^\top\mathbf{R}_2\boldsymbol{\mu}_\beta} \tag{25}$$

**Lemma C.1** (Delta). *The parameter $\delta$ as defined in equation 9, is given by the following identity:*

$$\delta = \frac{\eta}{\theta} = \frac{\eta}{\gamma + \frac{\pi}{1+\delta} + \frac{\alpha\sigma^2(1-\pi)}{1+\alpha\sigma^2\delta}}$$

*Which gives us a third order equation:*

$$\alpha\sigma^2\gamma\delta^3 + \left(\gamma + \alpha\sigma^2(1 + \gamma - \eta)\right)\delta^2 + \left(\gamma + \pi - \eta + \alpha\sigma^2(1 - \pi - \eta)\right)\delta - \eta = 0$$

**Lemma C.2** (Resolvent identities). *Using the first identity in Sherman-Morisson's lemma A.3, we have that the expressions of $\mathbf{R}_1$ and $\mathbf{R}_2$ are given by:*

$$\mathbf{R}_1 = \frac{1}{\theta}\mathbf{I}_p - \frac{\alpha(1-\pi)\boldsymbol{\mu}_\beta\boldsymbol{\mu}_\beta^\top}{\theta^2(1+\alpha\sigma^2\delta) + \theta\alpha(1-\pi)\|\boldsymbol{\mu}_\beta\|^2}, \quad \mathbf{R}_2 = \frac{1}{\theta}\mathbf{I}_p - \frac{\pi\boldsymbol{\mu}\boldsymbol{\mu}^\top}{\theta^2(1+\delta) + \theta\pi\|\boldsymbol{\mu}\|^2}$$

*And we also have the following identities:*

$$\mathbf{R}_1\boldsymbol{\mu}_\beta = \frac{\boldsymbol{\mu}_\beta}{\theta + \frac{\alpha(1-\pi)}{1+\delta_S}\|\boldsymbol{\mu}_\beta\|^2}, \quad \mathbf{R}_2\boldsymbol{\mu} = \frac{\boldsymbol{\mu}}{\theta + \frac{\pi}{1+\delta}\|\boldsymbol{\mu}\|^2}$$

$$\boldsymbol{\mu}^\top\mathbf{R}_1\boldsymbol{\mu} = \frac{\|\boldsymbol{\mu}\|^2}{\theta}\left(1 - \frac{\alpha(1-\pi)\beta^2\|\boldsymbol{\mu}\|^2}{\theta(1+\delta_S) + \alpha(1-\pi)\|\boldsymbol{\mu}_\beta\|^2}\right) = \frac{\|\boldsymbol{\mu}\|^2}{\theta}\frac{\theta(1+\delta_S) + \alpha(1-\pi)(1-\beta^2)\|\boldsymbol{\mu}^\perp\|^2}{\theta(1+\delta_S) + \alpha(1-\pi)\|\boldsymbol{\mu}_\beta\|^2}$$

$$\boldsymbol{\mu}^\top\mathbf{R}_2\boldsymbol{\mu}_\beta = \frac{\beta(1+\delta)\|\boldsymbol{\mu}\|^2}{\theta(1+\delta) + \pi\|\boldsymbol{\mu}\|^2}$$

$$\boldsymbol{\mu}_\beta^\top \mathbf{R}_2 \boldsymbol{\mu}_\beta = \frac{1}{\theta}\left(\|\boldsymbol{\mu}_\beta\|^2 - \frac{\pi\beta^2\|\boldsymbol{\mu}\|^4}{\theta(1+\delta)+\pi\|\boldsymbol{\mu}\|^2}\right)$$

$$\boldsymbol{\mu}^\top \mathbf{R}_1^2 \boldsymbol{\mu} = \frac{\|\boldsymbol{\mu}\|^2}{\theta^2}\left(1 - \frac{2\alpha(1-\pi)\beta^2\|\boldsymbol{\mu}\|^2}{\theta(1+\delta_S)+\alpha(1-\pi)\|\boldsymbol{\mu}_\beta\|^2} + \frac{\alpha^2(1-\pi)^2\beta^2\|\boldsymbol{\mu}\|^2\|\boldsymbol{\mu}_\beta\|^2}{(\theta(1+\delta_S)+\alpha(1-\pi)\|\boldsymbol{\mu}_\beta\|^2)^2}\right)$$

$$= \frac{\|\boldsymbol{\mu}\|^2}{\theta^2} + \frac{\alpha(1-\pi)\beta^2\|\boldsymbol{\mu}\|^4}{\theta^2(\theta(1+\delta_S)+\alpha(1-\pi)\|\boldsymbol{\mu}_\beta\|^2)}\left(\frac{\alpha(1-\pi)\|\boldsymbol{\mu}_\beta\|^2}{\theta(1+\delta_S)+\alpha(1-\pi)\|\boldsymbol{\mu}_\beta\|^2} - 2\right)$$

$$\boldsymbol{\mu}_\beta^\top \mathbf{R}_2 \mathbf{R}_1 \boldsymbol{\mu} = \frac{\beta\|\boldsymbol{\mu}\|^2}{\theta^2}(1 - \frac{\alpha(1-\pi)\|\boldsymbol{\mu}_\beta\|^2}{\theta(1+\delta_S)+\alpha(1-\pi)\|\boldsymbol{\mu}_\beta\|^2} - \frac{\pi\|\boldsymbol{\mu}\|^2}{\theta(1+\delta)+\pi\|\boldsymbol{\mu}\|^2}$$

$$+ \frac{\alpha\pi(1-\pi)\beta^2\|\boldsymbol{\mu}\|^4}{(\theta(1+\delta)+\pi\|\boldsymbol{\mu}\|^2)(\theta(1+\delta_S)+\alpha(1-\pi)\|\boldsymbol{\mu}_\beta\|^2)})$$

$$\boldsymbol{\mu}_\beta^\top \mathbf{R}_2^2 \boldsymbol{\mu}_\beta = \frac{\|\boldsymbol{\mu}_\beta\|^2}{\theta^2} + \frac{\pi\beta^2\|\boldsymbol{\mu}\|^4}{\theta^2(\theta(1+\delta)+\pi\|\boldsymbol{\mu}\|^2)}\left(\frac{\pi\|\boldsymbol{\mu}\|^2}{\theta(1+\delta)+\pi\|\boldsymbol{\mu}\|^2} - 2\right)$$

$$= \frac{\|\boldsymbol{\mu}_\beta\|^2}{\theta^2} - \frac{\pi\beta^2\|\boldsymbol{\mu}\|^4\left(\pi\|\boldsymbol{\mu}\|^2+2\theta(1+\delta)\right)}{\theta^2(\theta(1+\delta)+\pi\|\boldsymbol{\mu}\|^2)^2}$$

**Lemma C.3** (Trace identities). *Let $i \in \{1,...,n\}$, and $j \in \{1,...,m\}$, such that: $\Sigma = \mathbb{E}[\boldsymbol{x}_i \boldsymbol{x}_i^\top] = \boldsymbol{\mu}\boldsymbol{\mu}^\top + \mathbf{I}_p$ and $\Sigma_\beta = \mathbb{E}[\tilde{\boldsymbol{x}}_j \tilde{\boldsymbol{x}}_j^\top] = \boldsymbol{\mu}_\beta \boldsymbol{\mu}_\beta^\top + \sigma^2 \mathbf{I}_p$.*
*We can prove that:*

$$\frac{1}{N}\operatorname{Tr}((\Sigma\bar{\mathbf{Q}})^2) = \frac{\eta}{\theta^2}, \quad \frac{1}{N}\operatorname{Tr}((\Sigma_\beta\bar{\mathbf{Q}})^2) = \frac{\eta\sigma^4}{\theta^2}, \quad \frac{1}{N}\operatorname{Tr}(\Sigma_\beta\bar{\mathbf{Q}}\Sigma\bar{\mathbf{Q}}) = \frac{\eta\sigma^2}{\theta^2}$$

The performance of $\boldsymbol{w}_q$ in (6) is fully determined by the first two order moments: $\mathbb{E}[\boldsymbol{w}_q^\top \boldsymbol{x}]$ and $\mathbb{E}[(\boldsymbol{w}_q^\top \boldsymbol{x})^2]$.

## C.2 TEST EXPECTATION

We have that using the calculus in the past section:

$$\mathbb{E}[\boldsymbol{w}_q^\top \boldsymbol{x}] = (-1)^a\left(\frac{\pi}{1+\delta}\boldsymbol{\mu}^\top + \frac{\lambda(1-\pi)}{1+\alpha\sigma^2\delta}\boldsymbol{\mu}_\beta^\top\right)\bar{\mathbf{Q}}\boldsymbol{\mu} \tag{26}$$

And finally we use lemma C.2 and the following identities to obtain the result:

$$\boldsymbol{\mu}^\top\bar{\mathbf{Q}}\boldsymbol{\mu} = \frac{\boldsymbol{\mu}^\top\mathbf{R}_1\boldsymbol{\mu}}{1+\frac{\pi}{1+\delta}\boldsymbol{\mu}^\top\mathbf{R}_1\boldsymbol{\mu}}, \quad \boldsymbol{\mu}_\beta^\top\bar{\mathbf{Q}}\boldsymbol{\mu} = \frac{\boldsymbol{\mu}_\beta^\top\mathbf{R}_2\boldsymbol{\mu}}{1+\frac{\alpha(1-\pi)}{1+\delta_S}\boldsymbol{\mu}_\beta^\top\mathbf{R}_2\boldsymbol{\mu}_\beta}$$

## C.3 TEST VARIANCE

To determine the variance of $\boldsymbol{w}_q^\top \boldsymbol{x}$, it only remains to compute its second order. We have that:

$$\mathbb{E}[(\boldsymbol{w}_q^\top \boldsymbol{x})^2] = \frac{1}{N^2}\mathbb{E}\left[\left(\sum_{i=1}^n y_i \boldsymbol{x}_i^\top \mathbf{Q}\boldsymbol{x} + \sum_{j=1}^m q_j \tilde{y}_j \tilde{\boldsymbol{x}}_j^\top \mathbf{Q}\boldsymbol{x}\right)^2\right]$$

$$= \frac{1}{N^2}\mathbb{E}\left[\left(\sum_{i=1}^n y_i \boldsymbol{x}_i^\top \mathbf{Q}\boldsymbol{x}\right)^2\right] + \frac{1}{N^2}\mathbb{E}\left[\left(\sum_{j=1}^m q_j \tilde{y}_j \tilde{\boldsymbol{x}}_j^\top \mathbf{Q}\boldsymbol{x}\right)^2\right] + \frac{2}{N^2}\mathbb{E}\left[\left(\sum_{i=1}^n y_i \boldsymbol{x}_i^\top \mathbf{Q}\boldsymbol{x}\right)\left(\sum_{j=1}^m q_j \tilde{y}_j \tilde{\boldsymbol{x}}_j^\top \mathbf{Q}\boldsymbol{x}\right)\right]$$

And using the same computations in the past section, we get:

**First sum:** The first sum gives us:

$$\frac{1}{N^2}\mathbb{E}\left[\left(\sum_{i=1}^n y_i \boldsymbol{x}_i^\top \mathbf{Q}\boldsymbol{x}\right)^2\right] = \frac{\pi^2}{(1+\delta)^2}\left(\boldsymbol{\mu}^\top\mathbb{E}[\mathbf{Q}\Sigma\mathbf{Q}]\boldsymbol{\mu} - \frac{2\operatorname{Tr}(\Sigma\mathbb{E}[\mathbf{Q}\Sigma\mathbf{Q}])}{N(1+\delta)}\boldsymbol{\mu}^\top\bar{\mathbf{Q}}\boldsymbol{\mu}\right) + \frac{\pi}{N(1+\delta)^2}\operatorname{Tr}(\Sigma\mathbb{E}[\mathbf{Q}\Sigma\mathbf{Q}])$$

**Second sum:** the second sum gives us:

$$\frac{1}{N^2}\mathbb{E}\left[\left(\sum_{i=1}^m q_i\tilde{y}_i\tilde{\boldsymbol{x}}_i^\top\mathbf{Q}\boldsymbol{x}\right)^2\right]$$

$$=\frac{\lambda^2(1-\pi)^2}{(1+\delta_S)^2}\left(\boldsymbol{\mu}_\beta^\top\mathbb{E}[\mathbf{Q}\Sigma\mathbf{Q}]\boldsymbol{\mu}_\beta-\frac{2}{N(1+\delta_S)}\operatorname{Tr}(\Sigma_\beta\mathbb{E}[\mathbf{Q}\Sigma\mathbf{Q}])\boldsymbol{\mu}_\beta^\top\bar{\mathbf{Q}}\boldsymbol{\mu}_\beta\right)+\frac{\alpha(1-\pi)}{N(1+\delta_S)^2}\operatorname{Tr}(\Sigma_\beta\mathbb{E}[\mathbf{Q}\Sigma\mathbf{Q}])$$

**Third sum:** The third sum is given by:

$$\frac{2}{N^2}\sum_{i=1}^n\sum_{j=1}^m\mathbb{E}[y_i\boldsymbol{x}_i^\top\mathbf{Q}\boldsymbol{x}q_j\tilde{y}_j\tilde{\boldsymbol{x}}_j^\top\mathbf{Q}\boldsymbol{x}]$$

$$=\frac{2\lambda\pi(1-\pi)}{(1+\delta)(1+\delta_S)}\left(\boldsymbol{\mu}^\top\mathbb{E}[\mathbf{Q}\Sigma\mathbf{Q}]\boldsymbol{\mu}_\beta-\frac{1}{N}\operatorname{Tr}\left(\left(\frac{\Sigma}{1+\delta}+\frac{\Sigma_\beta}{1+\delta_S}\right)\mathbb{E}[\mathbf{Q}\Sigma\mathbf{Q}]\right)\boldsymbol{\mu}^\top\bar{\mathbf{Q}}\boldsymbol{\mu}_\beta\right)$$

**Grouping all the sums:** Denote by : $T=\frac{1}{N}\operatorname{Tr}(\Sigma\mathbb{E}[\mathbf{Q}\Sigma\mathbf{Q}])$, then: $\frac{1}{N}\operatorname{Tr}(\Sigma_\beta\mathbb{E}[\mathbf{Q}\Sigma\mathbf{Q}])=\sigma^2 T$.
Now let us group the terms in $T$ in the three sums, and those that do not depend on $T$. We get that:

$$\mathbb{E}[(\boldsymbol{w}^\top\boldsymbol{x})^2]=\frac{\pi^2}{(1+\delta)^2}\boldsymbol{\mu}^\top\mathbb{E}[\mathbf{Q}\Sigma\mathbf{Q}]\boldsymbol{\mu}+\frac{\lambda^2(1-\pi)^2}{(1+\delta_S)^2}\boldsymbol{\mu}_\beta^\top\mathbb{E}[\mathbf{Q}\Sigma\mathbf{Q}]\boldsymbol{\mu}_\beta+\frac{2\lambda\pi(1-\pi)}{(1+\delta)(1+\delta_S)}\boldsymbol{\mu}^\top\mathbb{E}[\mathbf{Q}\Sigma\mathbf{Q}]\boldsymbol{\mu}_\beta$$

$$+T\left(\frac{\pi}{(1+\delta)^2}-\frac{2\pi^2}{(1+\delta)^3}\boldsymbol{\mu}^\top\bar{\mathbf{Q}}\boldsymbol{\mu}-\frac{2\lambda\pi(1-\pi)}{(1+\delta)^2(1+\delta_S)}\boldsymbol{\mu}^\top\bar{\mathbf{Q}}\boldsymbol{\mu}_\beta\right)$$

$$+\sigma^2 T\left(\frac{\alpha(1-\pi)}{(1+\delta_S)^2}-\frac{2\lambda^2(1-\pi)^2}{(1+\delta_S)^3}\boldsymbol{\mu}_\beta^\top\bar{\mathbf{Q}}\boldsymbol{\mu}_\beta-\frac{2\lambda\pi(1-\pi)}{(1+\delta)(1+\delta_S)^2}\boldsymbol{\mu}^\top\bar{\mathbf{Q}}\boldsymbol{\mu}_\beta\right)$$

$$=\frac{\pi^2}{(1+\delta)^2}\boldsymbol{\mu}^\top\mathbb{E}[\mathbf{Q}\Sigma\mathbf{Q}]\boldsymbol{\mu}+\frac{\lambda^2(1-\pi)^2}{(1+\delta_S)^2}\boldsymbol{\mu}_\beta^\top\mathbb{E}[\mathbf{Q}\Sigma\mathbf{Q}]\boldsymbol{\mu}_\beta+\frac{2\lambda\pi(1-\pi)}{(1+\delta)(1+\delta_S)}\boldsymbol{\mu}^\top\mathbb{E}[\mathbf{Q}\Sigma\mathbf{Q}]\boldsymbol{\mu}_\beta$$

$$+\frac{\pi T}{(1+\delta)^2}\left(1-\frac{2\pi}{1+\delta}\boldsymbol{\mu}^\top\bar{\mathbf{Q}}\boldsymbol{\mu}-\frac{2\lambda(1-\pi)}{1+\delta_S}\boldsymbol{\mu}^\top\bar{\mathbf{Q}}\boldsymbol{\mu}_\beta\right)$$

$$+\frac{(1-\pi)\sigma^2 T}{(1+\delta_S)^2}\left(\alpha-\frac{2\lambda^2(1-\pi)}{1+\delta_S}\boldsymbol{\mu}_\beta^\top\bar{\mathbf{Q}}\boldsymbol{\mu}_\beta-\frac{2\lambda\pi}{1+\delta}\boldsymbol{\mu}^\top\bar{\mathbf{Q}}\boldsymbol{\mu}_\beta\right)$$

And we can compute this since we have that:

$$\boldsymbol{\mu}^\top\mathbb{E}[\mathbf{Q}\Sigma\mathbf{Q}]\boldsymbol{\mu}=\frac{1}{h}\left((1-b_2)\left((\boldsymbol{\mu}^\top\bar{\mathbf{Q}}\boldsymbol{\mu})^2+\boldsymbol{\mu}^\top\bar{\mathbf{Q}}^2\boldsymbol{\mu}\right)+b_1\left((\boldsymbol{\mu}^\top\bar{\mathbf{Q}}\boldsymbol{\mu}_\beta)^2+\sigma^2\boldsymbol{\mu}^\top\bar{\mathbf{Q}}^2\boldsymbol{\mu}\right)\right),$$

$$\boldsymbol{\mu}_\beta^\top\mathbb{E}[\mathbf{Q}\Sigma\mathbf{Q}]\boldsymbol{\mu}_\beta=\frac{1}{h}\left((1-b_2)\left[(\boldsymbol{\mu}_\beta^\top\bar{\mathbf{Q}}\boldsymbol{\mu})^2+\boldsymbol{\mu}_\beta^\top\bar{\mathbf{Q}}^2\boldsymbol{\mu}_\beta\right]+b_1\left[(\boldsymbol{\mu}_\beta^\top\bar{\mathbf{Q}}\boldsymbol{\mu}_\beta)^2+\sigma^2\boldsymbol{\mu}_\beta^\top\bar{\mathbf{Q}}^2\boldsymbol{\mu}_\beta\right]\right)$$

$$\boldsymbol{\mu}^\top\mathbb{E}[\mathbf{Q}\Sigma\mathbf{Q}]\boldsymbol{\mu}_\beta=\frac{1}{h}\left((1-b_2)\left[\boldsymbol{\mu}^\top\bar{\mathbf{Q}}\boldsymbol{\mu}.\boldsymbol{\mu}^\top\bar{\mathbf{Q}}\boldsymbol{\mu}_\beta+\boldsymbol{\mu}^\top\bar{\mathbf{Q}}^2\boldsymbol{\mu}_\beta\right]+b_1\left[\boldsymbol{\mu}^\top\bar{\mathbf{Q}}\boldsymbol{\mu}_\beta.\boldsymbol{\mu}_\beta^\top\bar{\mathbf{Q}}\boldsymbol{\mu}_\beta+\sigma^2\boldsymbol{\mu}^\top\bar{\mathbf{Q}}^2\boldsymbol{\mu}_\beta\right]\right)$$

$$\boldsymbol{\mu}_\beta\bar{\mathbf{Q}}\boldsymbol{\mu}_\beta=\frac{\boldsymbol{\mu}_\beta^\top\mathbf{R}_2\boldsymbol{\mu}_\beta}{1+\frac{\alpha(1-\pi)}{1+\delta_S}\boldsymbol{\mu}_\beta\mathbf{R}_2\boldsymbol{\mu}_\beta},\quad\boldsymbol{\mu}^\top\bar{\mathbf{Q}}^2\boldsymbol{\mu}=\frac{\boldsymbol{\mu}^\top\mathbf{R}_1^2\boldsymbol{\mu}}{\left(1+\frac{\pi}{1+\delta}\boldsymbol{\mu}^\top\mathbf{R}_1\boldsymbol{\mu}\right)^2},\quad\boldsymbol{\mu}_\beta^\top\bar{\mathbf{Q}}^2\boldsymbol{\mu}_\beta=\frac{\boldsymbol{\mu}_\beta^\top\mathbf{R}_2^2\boldsymbol{\mu}_\beta}{\left(1+\frac{\alpha(1-\pi)}{1+\delta_S}\boldsymbol{\mu}_\beta^\top\mathbf{R}_2\boldsymbol{\mu}_\beta\right)^2},$$

$$\boldsymbol{\mu}^\top\bar{\mathbf{Q}}^2\boldsymbol{\mu}_\beta=\frac{\boldsymbol{\mu}_\beta^\top\mathbf{R}_2\mathbf{R}_1\boldsymbol{\mu}}{\left(1+\frac{\pi}{1+\delta}\boldsymbol{\mu}^\top\mathbf{R}_1\boldsymbol{\mu}\right)\left(1+\frac{\alpha(1-\pi)}{1+\delta_S}\boldsymbol{\mu}_\beta^\top\mathbf{R}_2\boldsymbol{\mu}_\beta\right)}$$

**Theorem C.4** (Gaussianity of the 6 model for $\mathbf{C}=\sigma^2\mathbf{I}_p$). *Let $\boldsymbol{w}_q$ be the Mixed classifier as defined in equation 6 and suppose that Assumption 4.1 holds. The decision function $\boldsymbol{w}_q^\top\boldsymbol{x}$, on some test sample $\boldsymbol{x}\in\mathcal{C}_a$ independent of $\mathbf{X}$, satisfies:*

$$\boldsymbol{w}_q^\top\boldsymbol{x}\xrightarrow{\mathcal{D}}\mathcal{N}\left((-1)^a m_q,\nu_q-m_q^2\right),$$

*where:*

$$m_q = \left( \frac{\pi}{1+\delta} \boldsymbol{\mu}^\top + \frac{\lambda(1-\pi)}{1+\delta_S} \boldsymbol{\mu}_\beta^\top \right) \bar{\mathbf{Q}} \boldsymbol{\mu},$$

$$\nu_q = \frac{\pi^2}{(1+\delta)^2} \boldsymbol{\mu}^\top \mathbb{E}[\mathbf{Q}\boldsymbol{\Sigma}\mathbf{Q}]\boldsymbol{\mu} + \frac{\lambda^2(1-\pi)^2}{(1+\delta_S)^2} \boldsymbol{\mu}_\beta^\top \mathbb{E}[\mathbf{Q}\boldsymbol{\Sigma}\mathbf{Q}]\boldsymbol{\mu}_\beta + \frac{2\lambda\pi(1-\pi)}{(1+\delta)(1+\delta_S)} \boldsymbol{\mu}^\top \mathbb{E}[\mathbf{Q}\boldsymbol{\Sigma}\mathbf{Q}]\boldsymbol{\mu}_\beta$$

$$+ \frac{\pi T}{(1+\delta)^2} \left( 1 - \frac{2\pi}{1+\delta} \boldsymbol{\mu}^\top \bar{\mathbf{Q}} \boldsymbol{\mu} - \frac{2\lambda(1-\pi)}{1+\delta_S} \boldsymbol{\mu}^\top \bar{\mathbf{Q}} \boldsymbol{\mu}_\beta \right)$$

$$+ \frac{(1-\pi)\sigma^2 T}{(1+\delta_S)^2} \left( \alpha - \frac{2\lambda^2(1-\pi)}{1+\delta_S} \boldsymbol{\mu}_\beta^\top \bar{\mathbf{Q}} \boldsymbol{\mu}_\beta - \frac{2\lambda\pi}{1+\delta} \boldsymbol{\mu}^\top \bar{\mathbf{Q}} \boldsymbol{\mu}_\beta \right).$$

*With:*

$$\lambda = \phi(1-\varepsilon) - \rho\varepsilon, \quad \delta_S = \alpha\sigma^2\delta$$

# D   RANDOM MATRIX ANALYSIS OF DISTRIBUTION SHIFT

We will now quantify the performance of the classifier obtained through mixing some real data and synthetic data sampled according to the schema described in 2. Hence, the matrix $\bar{\mathbf{Q}}$, defined in lemma A.4, is no longer deterministic as we take the covariance matrix $\hat{\mathbf{C}} = \frac{1}{\hat{n}} \sum_{i=1}^{\hat{n}} (\boldsymbol{x}_i - y_i \hat{\boldsymbol{\mu}})(\boldsymbol{x}_i - y_i \hat{\boldsymbol{\mu}})^\top$. For simplicity, and without loss of generality, we consider $\hat{n}$ Gaussian vectors $(\boldsymbol{z}_i)_{i=1}^{\hat{n}} \sim \mathcal{N}(0, \mathbf{I}_p)$ that are independent of $(\boldsymbol{x}_i)_{i=1}^{\hat{n}}$, and write:

$$\hat{\boldsymbol{\mu}} = \boldsymbol{\mu}_\beta = \boldsymbol{\mu} + \frac{1}{\hat{n}} \sum_{i=1}^{\hat{n}} \boldsymbol{z}_i, \quad \hat{\mathbf{C}} = \frac{1}{\hat{n}} \sum_{i=1}^{\hat{n}} \boldsymbol{z}_i \boldsymbol{z}_i^\top$$

Note that we can ignore the error of estimation of $\hat{\boldsymbol{\mu}}$ because we have that:

$$\mathbb{E}\left[\frac{1}{\hat{n}} \sum_{i=1}^{\hat{n}} \boldsymbol{z}_i\right] = 0, \quad \mathbb{E}\left[\left(\frac{1}{\hat{n}} \sum_{i=1}^{\hat{n}} \boldsymbol{z}_i\right)\left(\frac{1}{\hat{n}} \sum_{j=1}^{\hat{n}} \boldsymbol{z}_j\right)^\top\right] = \frac{1}{\hat{n}} \mathbf{I}_p$$

Hence, when we have a sufficiently large $\hat{n}$, we will assume that: $\hat{\boldsymbol{\mu}} = \boldsymbol{\mu}$ (the estimation error is on $\mathcal{O}(\hat{n}^{-1})$).

## D.1   DETERMINISTIC EQUIVALENTS:

The resolvent matrix to be considered in this setting is the one defined in lemma A.4 but with $\hat{\mathbf{C}}$:

$$\bar{\mathbf{Q}}(\gamma) = \left(\left(\frac{\pi}{1+\delta} + \frac{\alpha(1-\pi)}{1+\delta_S}\right)\boldsymbol{\mu}\boldsymbol{\mu}^\top + \frac{\alpha(1-\pi)}{1+\delta_S}\hat{\mathbf{C}} + \left(\gamma + \frac{\pi}{1+\delta}\right)\mathbf{I}_p\right)^{-1}$$

$$= \left(\left(\frac{\pi}{1+\delta} + \frac{\alpha(1-\pi)}{1+\delta_S}\right)\boldsymbol{\mu}\boldsymbol{\mu}^\top + \frac{\alpha(1-\pi)}{(1+\delta_S)\hat{n}}\sum_{i=1}^{\hat{n}} \boldsymbol{z}_i \boldsymbol{z}_i^\top + \left(\gamma + \frac{\pi}{1+\delta}\right)\mathbf{I}_p\right)^{-1}$$

where:

$$\delta = \frac{1}{N}\operatorname{Tr}(\bar{\mathbf{Q}}), \quad \delta_S = \frac{\alpha}{N}\operatorname{Tr}(\hat{\mathbf{C}}\bar{\mathbf{Q}})$$

Let us denote by $\bar{\mathbf{Q}}_{-i}$ the resolvent matrix gotten by removing its dependence on the vector $\boldsymbol{z}_i$. In other words:

$$\bar{\mathbf{Q}}_{-i} = \left(\bar{\mathbf{Q}} - \frac{\alpha(1-\pi)}{\hat{n}(1+\delta_S)}\boldsymbol{z}_i \boldsymbol{z}_i^\top\right)^{-1}, \quad \bar{\mathbf{Q}} = \left(\bar{\mathbf{Q}}_{-i} + \frac{\alpha(1-\pi)}{\hat{n}(1+\delta_S)}\boldsymbol{z}_i \boldsymbol{z}_i^\top\right)^{-1}$$

By Sherman-Morisson's lemma A.3, we have that:

$$\bar{\mathbf{Q}} = \bar{\mathbf{Q}}_{-i} - \frac{\frac{\alpha(1-\pi)}{\hat{n}(1+\delta_S)}\bar{\mathbf{Q}}_{-i}\boldsymbol{z}_i \boldsymbol{z}_i^\top \bar{\mathbf{Q}}_{-i}}{1 + \frac{\alpha(1-\pi)}{\hat{n}(1+\delta_S)}\boldsymbol{z}_i^\top \bar{\mathbf{Q}}_{-i}\boldsymbol{z}_i}$$

And:

$$\bar{\mathbf{Q}}\boldsymbol{z}_i = \frac{\bar{\mathbf{Q}}_{-i}\boldsymbol{z}_i}{1 + \frac{\alpha(1-\pi)}{\hat{n}(1+\delta_S)}\boldsymbol{z}_i^\top \bar{\mathbf{Q}}_{-i}\boldsymbol{z}_i} = \frac{\bar{\mathbf{Q}}_{-i}\boldsymbol{z}_i}{1 + \bar{\delta}}$$

where:

$$\bar{\delta} = \frac{\alpha(1-\pi)}{1+\delta_S}\frac{1}{\hat{n}}\operatorname{Tr}(\bar{\mathbf{Q}}) \tag{27}$$

Since the covariance estimate in equation 2 is stochastic, the matrix $\bar{\mathbf{Q}}$ is no longer deterministic when replacing $\mathbf{C}$ with $\hat{\mathbf{C}}$. Hence, we will give a further deterministic equivalent to $\bar{\mathbf{Q}}$ in the following lemma.

**Lemma D.1** (Second Deterministic equivalent). *A deterministic equivalent of $\bar{\mathbf{Q}}$ is given by:*

$$\bar{\bar{\mathbf{Q}}} = \left( \left( \frac{\pi}{1+\delta} + \frac{\alpha(1-\pi)}{1+\delta_S} \right) \boldsymbol{\mu}\boldsymbol{\mu}^\top + \left( \gamma + \frac{\pi}{1+\delta} + \frac{\alpha(1-\pi)}{(1+\delta_S)(1+\bar{\delta})} \right) \mathbf{I}_p \right)^{-1}$$

*where $\bar{\delta}$ can be found as a fixed point using the following identity:*

$$\bar{\delta} = \frac{\alpha(1-\pi)}{(1+\delta_S)} \frac{1}{\hat{n}} \operatorname{Tr}(\bar{\bar{\mathbf{Q}}}) = \frac{\alpha(1-\pi)}{(1+\delta_S)} \frac{\frac{p}{\hat{n}}}{\gamma + \frac{\pi}{1+\delta} + \frac{\alpha(1-\pi)}{(1+\delta_S)(1+\bar{\delta})}}, \quad \delta = \frac{1}{N} \operatorname{Tr}(\bar{\bar{\mathbf{Q}}}) = \frac{\hat{n}}{N} \frac{(1+\delta_S)}{\alpha(1-\pi)} \bar{\delta}$$

$$\delta_S = \frac{\alpha}{N} \operatorname{Tr}(\mathbb{E}[\hat{\mathbf{C}}\bar{\mathbf{Q}}]) = \frac{\alpha\delta}{1+\bar{\delta}}$$

Now we will prove the deterministic equivalent given by lemma D.1.

PROOF OF LEMMA D.1:

Let us denote $\bar{\mathbf{Q}} = \left( \frac{\alpha(1-\pi)}{1+\delta_S} \frac{1}{\hat{n}} \sum_{i=1}^{\hat{n}} \boldsymbol{z}_i \boldsymbol{z}_i^\top + \mathbf{A} \right)^{-1}$. Let also $\bar{\bar{\mathbf{Q}}}$ be the deterministic equivalent of $\bar{\mathbf{Q}}$. It can be written as: $\bar{\bar{\mathbf{Q}}} = (\mathbf{S} + \mathbf{A})^{-1}$. We want to find some $\mathbf{S}$ such that for all $\boldsymbol{a}, \boldsymbol{b} \in \mathbb{R}^p$:

$$\boldsymbol{a}^\top \mathbb{E}[\bar{\mathbf{Q}}] \boldsymbol{b} \to \boldsymbol{a}^\top \bar{\bar{\mathbf{Q}}} \boldsymbol{b}$$

We have that:

$$\mathbb{E}[\bar{\mathbf{Q}}] - \bar{\bar{\mathbf{Q}}} = \mathbb{E}\left[ \bar{\mathbf{Q}}(\mathbf{S} - \frac{\alpha(1-\pi)}{1+\delta_S} \frac{1}{\hat{n}} \sum_{i=1}^{\hat{n}} \boldsymbol{z}_i \boldsymbol{z}_i^\top) \bar{\bar{\mathbf{Q}}} \right]$$

$$= \frac{1}{\hat{n}} \sum_{i=1}^{\hat{n}} \mathbb{E}[\bar{\mathbf{Q}}(\mathbf{S} - \frac{\alpha(1-\pi)}{1+\delta_S} \boldsymbol{z}_i \boldsymbol{z}_i^\top) \bar{\bar{\mathbf{Q}}}]$$

$$= \frac{1}{\hat{n}} \sum_{i=1}^{\hat{n}} \mathbb{E}\left[ \bar{\mathbf{Q}}\mathbf{S} - \frac{\alpha(1-\pi)}{1+\delta_S} \frac{1}{1+\bar{\delta}} \bar{\mathbf{Q}}_{-i} \boldsymbol{z}_i \boldsymbol{z}_i^\top \right] \bar{\bar{\mathbf{Q}}}$$

$$= \frac{1}{\hat{n}} \sum_{i=1}^{\hat{n}} \mathbb{E}\left[ \bar{\mathbf{Q}}_{-i} \left( \mathbf{S} - \frac{\alpha(1-\pi)}{1+\delta_S} \frac{1}{1+\bar{\delta}} \boldsymbol{z}_i \boldsymbol{z}_i^\top \right) \right] \bar{\bar{\mathbf{Q}}} + \mathcal{O}(\hat{n}^{-1})$$

Hence, it suffices to have $\mathbf{S} = \mathbb{E}[\frac{\alpha(1-\pi)}{1+\delta_S} \frac{1}{1+\bar{\delta}} \boldsymbol{z}_i \boldsymbol{z}_i^\top] = \frac{\alpha(1-\pi)}{1+\delta_S} \frac{1}{1+\bar{\delta}} \mathbf{I}_p$, and thus:

## D.2 DETERMINISTIC EQUIVALENT OF $\bar{\mathbf{Q}}\mathbf{A}\bar{\mathbf{Q}}$:

Let $\mathbf{A} \in \mathbb{R}^{p \times p}$ be some deterministic matrix. We have that:

$$\mathbb{E}[\bar{\mathbf{Q}}\mathbf{A}\bar{\mathbf{Q}}] = \bar{\bar{\mathbf{Q}}}\mathbf{A}\bar{\bar{\mathbf{Q}}} + \mathbb{E}[(\bar{\mathbf{Q}} - \bar{\bar{\mathbf{Q}}})\mathbf{A}\bar{\mathbf{Q}}]$$

$$= \bar{\bar{\mathbf{Q}}}\mathbf{A}\bar{\bar{\mathbf{Q}}} + \mathbb{E}[\bar{\mathbf{Q}}(\bar{\bar{\mathbf{Q}}}^{-1} - \bar{\mathbf{Q}}^{-1})\bar{\bar{\mathbf{Q}}}\mathbf{A}\bar{\mathbf{Q}}]$$

$$= \bar{\bar{\mathbf{Q}}}\mathbf{A}\bar{\bar{\mathbf{Q}}} + \frac{\alpha(1-\pi)}{(1+\delta_S)} \mathbb{E}[\bar{\mathbf{Q}}\left( \frac{1}{1+\bar{\delta}} - \frac{1}{\hat{n}} \sum_{i=1}^{\hat{n}} \boldsymbol{z}_i \boldsymbol{z}_i^\top \right) \bar{\bar{\mathbf{Q}}}\mathbf{A}\bar{\mathbf{Q}}]$$

$$= \bar{\bar{\mathbf{Q}}}\mathbf{A}\bar{\bar{\mathbf{Q}}} + \frac{\alpha(1-\pi)}{(1+\delta_S)} \left( \frac{1}{1+\bar{\delta}} \mathbb{E}[\bar{\mathbf{Q}}\bar{\bar{\mathbf{Q}}}\mathbf{A}\bar{\mathbf{Q}}] - \frac{1}{\hat{n}} \sum_{i=1}^{\hat{n}} \mathbb{E}[\bar{\mathbf{Q}}\boldsymbol{z}_i \boldsymbol{z}_i^\top \bar{\bar{\mathbf{Q}}}\mathbf{A}\bar{\mathbf{Q}}] \right)$$

And we have that for $i \in \{1, ..., \hat{n}\}$:

$$\mathbb{E}[\bar{\mathbf{Q}}\boldsymbol{z}_i \boldsymbol{z}_i^\top \bar{\bar{\mathbf{Q}}}\mathbf{A}\bar{\mathbf{Q}}] = \frac{1}{1+\bar{\delta}} \mathbb{E}[\bar{\mathbf{Q}}_{-i} \boldsymbol{z}_i \boldsymbol{z}_i^\top \bar{\bar{\mathbf{Q}}}\mathbf{A}\bar{\mathbf{Q}}]$$

$$= \frac{1}{1+\bar{\delta}} \mathbb{E}\left[ \bar{\mathbf{Q}}_{-i} \boldsymbol{z}_i \boldsymbol{z}_i^\top \bar{\bar{\mathbf{Q}}}\mathbf{A} \left( \bar{\mathbf{Q}}_{-i} - \frac{\alpha(1-\pi)}{\hat{n}(1+\delta_S)(1+\bar{\delta})} \bar{\mathbf{Q}}_{-i} \boldsymbol{z}_i \boldsymbol{z}_i^\top \bar{\mathbf{Q}}_{-i} \right) \right]$$

$$= \frac{1}{1+\bar{\delta}} \mathbb{E}[\bar{\mathbf{Q}}_{-i} \boldsymbol{z}_i \boldsymbol{z}_i^\top \bar{\bar{\mathbf{Q}}}\mathbf{A}\bar{\mathbf{Q}}_{-i}] - \frac{\alpha(1-\pi)}{\hat{n}(1+\delta_S)(1+\bar{\delta})^2} \mathbb{E}[\bar{\mathbf{Q}}_{-i} \boldsymbol{z}_i \boldsymbol{z}_i^\top \bar{\bar{\mathbf{Q}}}\mathbf{A}\bar{\mathbf{Q}}_{-i} \boldsymbol{z}_i \boldsymbol{z}_i^\top \bar{\mathbf{Q}}_{-i}]$$

$$= \frac{1}{1+\bar{\delta}} \mathbb{E}[\bar{\mathbf{Q}}\bar{\bar{\mathbf{Q}}}\mathbf{A}\bar{\mathbf{Q}}] - \frac{\alpha(1-\pi)}{\hat{n}(1+\delta_S)(1+\bar{\delta})^2} \operatorname{Tr}(\bar{\bar{\mathbf{Q}}}\mathbf{A}\bar{\bar{\mathbf{Q}}})\mathbb{E}[\bar{\mathbf{Q}}^2]$$

Hence by replacing this term in he previous sum, we get the following result.

**Lemma D.2** (Deterministic equivalent of $\bar{\mathbf{Q}}\mathbf{A}\bar{\mathbf{Q}}$). *Let* $\mathbf{A} \in \mathbb{R}^{p \times p}$ *be any deterministic symmetric semi-definite matrix. We have that:*

$$\bar{\mathbf{Q}}\mathbf{A}\bar{\mathbf{Q}} \leftrightarrow \bar{\bar{\mathbf{Q}}}\mathbf{A}\bar{\bar{\mathbf{Q}}} + \left(\frac{\alpha(1-\pi)}{(1+\delta_S)(1+\bar{\delta})}\right)^2 \frac{1}{\hat{n}} \operatorname{Tr}(\bar{\bar{\mathbf{Q}}}\mathbf{A}\bar{\bar{\mathbf{Q}}})\mathbb{E}[\bar{\mathbf{Q}}^2]$$

*In particular, we have that:*

$$\bar{\mathbf{Q}}^2 \leftrightarrow \frac{1}{\bar{h}}\bar{\bar{\mathbf{Q}}}^2, \quad \bar{\mathbf{Q}}\boldsymbol{\mu}\boldsymbol{\mu}^\top\bar{\mathbf{Q}} \leftrightarrow \bar{\bar{\mathbf{Q}}}\boldsymbol{\mu}\boldsymbol{\mu}^\top\bar{\bar{\mathbf{Q}}}$$

*where:*

$$\bar{h} = 1 - \left(\frac{\alpha(1-\pi)}{(1+\delta_S)(1+\bar{\delta})}\right)^2 \frac{1}{\hat{n}} \operatorname{Tr}(\bar{\bar{\mathbf{Q}}}^2)$$

### D.3 USEFUL RESULTS:

Here we will list all the results with $\bar{\bar{\mathbf{Q}}}$ that will be useful in this analysis. Let us denote by $a, b$ the following quantities:

$$a = \left(\frac{\pi}{1+\delta} + \frac{\alpha(1-\pi)}{1+\delta_S}\right), \quad b = \gamma + \frac{\pi}{1+\delta} + \frac{\alpha(1-\pi)}{(1+\delta_S)(1+\bar{\delta})}$$

such that:

$$\bar{\bar{\mathbf{Q}}} = \left(a\boldsymbol{\mu}\boldsymbol{\mu}^\top + b\mathbf{I}_p\right)^{-1} \tag{28}$$

By Sherman-Morisson's lemma A.3, we have that:

$$\bar{\bar{\mathbf{Q}}} = \frac{1}{b}\left(\mathbf{I}_p - \frac{a\boldsymbol{\mu}\boldsymbol{\mu}^\top}{b + a\|\boldsymbol{\mu}\|^2}\right), \quad \bar{\bar{\mathbf{Q}}}\boldsymbol{\mu} = \frac{\boldsymbol{\mu}}{b + a\|\boldsymbol{\mu}\|^2} \tag{29}$$

We also have that the constants $a_1, a_2, b_1$ and $b_2$ from lemma A.5 become by taking their expectations on $\boldsymbol{z}$:

**Lemma D.3** (New values of constants).

$$a_1 = \frac{\pi}{N(1+\delta)^2}\frac{1}{\bar{h}}\operatorname{Tr}(\bar{\bar{\mathbf{Q}}}^2), \quad b_1 = \frac{\alpha(1-\pi)}{N(1+\delta_S)^2}\frac{1}{\bar{h}(1+\bar{\delta})^2}\operatorname{Tr}(\bar{\bar{\mathbf{Q}}}^2)$$

$$a_2 = \frac{\pi}{N(1+\delta)^2}\frac{1}{\bar{h}(1+\bar{\delta})^2}\operatorname{Tr}(\bar{\bar{\mathbf{Q}}}^2), \quad b_2 = \frac{\alpha(1-\pi)}{N(1+\delta_S)^2}\frac{1}{\bar{h}(1+\bar{\delta})^4}\operatorname{Tr}(\bar{\bar{\mathbf{Q}}}^2)$$

*where:*

$$\frac{1}{N}\operatorname{Tr}(\bar{\bar{\mathbf{Q}}}^2) = \frac{\eta}{b^2} \tag{30}$$

### D.4 TEST EXPECTATION:

It only suffices to apply the expectation on $\boldsymbol{z}_i$ to $m_q$ obtained with the general model in theorem B.1. Hence:

$$\mathbb{E}[\boldsymbol{w}_q^\top\boldsymbol{x}] = (-1)^a \left(\frac{\pi}{1+\delta} + \frac{\lambda(1-\pi)}{1+\delta_S}\right)\mathbb{E}[\boldsymbol{\mu}^\top\bar{\mathbf{Q}}\boldsymbol{\mu}]$$

$$= (-1)^a \left(\frac{\pi}{1+\delta} + \frac{\lambda(1-\pi)}{1+\delta_S}\right)\boldsymbol{\mu}^\top\bar{\bar{\mathbf{Q}}}\boldsymbol{\mu}$$

$$= (-1)^a \left(\frac{\pi}{1+\delta} + \frac{\lambda(1-\pi)}{1+\delta_S}\right)\frac{\|\boldsymbol{\mu}\|^2}{b + a\|\boldsymbol{\mu}\|^2}$$

### D.5 TEST VARIANCE:

Using theorem B.1, we need to apply the expectation on $z$ to the following second order moment:

$$
\nu_q = \left( \frac{\pi}{1+\delta} + \frac{\lambda(1-\pi)}{1+\delta_S} \right)^2 \boldsymbol{\mu}^\top \mathbb{E}[\mathbf{Q}\Sigma\mathbf{Q}]\boldsymbol{\mu} + \frac{\pi T_1}{(1+\delta)^2} \left( 1 - \frac{2\pi}{1+\delta}\boldsymbol{\mu}^\top \bar{\mathbf{Q}}\boldsymbol{\mu} - \frac{2\lambda(1-\pi)}{1+\delta_S}\boldsymbol{\mu}^\top \bar{\mathbf{Q}}\boldsymbol{\mu} \right)
$$
$$
+ \frac{(1-\pi)T_2}{(1+\delta_S)^2} \left( \alpha - \frac{2\lambda^2(1-\pi)}{1+\delta_S}\boldsymbol{\mu}^\top \bar{\mathbf{Q}}\boldsymbol{\mu} - \frac{2\lambda\pi}{1+\delta}\boldsymbol{\mu}^\top \bar{\mathbf{Q}}\boldsymbol{\mu} \right)
$$

where:

$$
T_1 = \frac{1}{N} \operatorname{Tr}(\Sigma\mathbb{E}[\mathbf{Q}\Sigma\mathbf{Q}]), \quad T_2 = \frac{1}{N} \operatorname{Tr}(\Sigma_\beta \mathbb{E}[\mathbf{Q}\Sigma\mathbf{Q}])
$$

and these two quantities are obtained using corollary A.6 and lemma D.3, which after simplification are given by:

$$
T_1 = \frac{(1+\delta)^2}{\pi h}a_1, \quad T_2 = \frac{(1+\delta_S)^2}{\alpha(1-\pi)h}b_1 \tag{31}
$$

Now we should define the new deterministic equivalent of $\bar{\mathbf{Q}}\Sigma\bar{\mathbf{Q}}$ to obtain an expression of $\mathbb{E}_{\boldsymbol{z}}[\mathbb{E}[\mathbf{Q}\Sigma\mathbf{Q}]]$ and to finish this calculus !
Let :

$$
\bar{\bar{h}} = 1 - \left( \frac{\alpha(1-\pi)}{(1+\delta_S)(1+\bar{\delta})} \right)^2 \frac{1}{\hat{n}} \operatorname{Tr}(\bar{\bar{\mathbf{Q}}}^2) \tag{32}
$$

Then, using lemma D.2 we have that the following identities stand for any linear form:

$$
\mathbb{E}[\bar{\mathbf{Q}}\Sigma\bar{\mathbf{Q}}] = \bar{\bar{\mathbf{Q}}}\boldsymbol{\mu}\boldsymbol{\mu}^\top \bar{\bar{\mathbf{Q}}} + \frac{1}{\bar{\bar{h}}}\bar{\bar{\mathbf{Q}}}^2, \quad \bar{\mathbb{E}}[\bar{\mathbf{Q}}\Sigma_\beta\bar{\mathbf{Q}}] = \bar{\bar{\mathbf{Q}}}\boldsymbol{\mu}\boldsymbol{\mu}^\top \bar{\bar{\mathbf{Q}}} + \frac{1}{(1+\bar{\delta})^2}\frac{1}{\bar{\bar{h}}}\bar{\bar{\mathbf{Q}}}^2
$$

Thus:

$$
\begin{aligned}
\mathbb{E}_{\boldsymbol{z}}[\mathbb{E}[\mathbf{Q}\Sigma\mathbf{Q}]] &= \frac{1-b_2}{h}\mathbb{E}[\bar{\mathbf{Q}}\Sigma\bar{\mathbf{Q}}] + \frac{b_1}{h}\mathbb{E}[\bar{\mathbf{Q}}\Sigma_\beta\bar{\mathbf{Q}}] \\
&= \frac{1}{h} \left( (1-b_2) \left( \bar{\bar{\mathbf{Q}}}\boldsymbol{\mu}\boldsymbol{\mu}^\top \bar{\bar{\mathbf{Q}}} + \frac{1}{\bar{\bar{h}}}\bar{\bar{\mathbf{Q}}}^2 \right) + b_1 \left( \bar{\bar{\mathbf{Q}}}\boldsymbol{\mu}\boldsymbol{\mu}^\top \bar{\bar{\mathbf{Q}}} + \frac{1}{\bar{\bar{h}}(1+\bar{\delta})^2}\bar{\bar{\mathbf{Q}}}^2 \right) \right) \\
&= \frac{1}{h} \left( (1+b_1-b_2)\bar{\bar{\mathbf{Q}}}\boldsymbol{\mu}\boldsymbol{\mu}^\top \bar{\bar{\mathbf{Q}}} + \frac{1}{\bar{\bar{h}}} \left( 1 - b_2 + \frac{b_1}{(1+\bar{\delta})^2} \right) \bar{\bar{\mathbf{Q}}}^2 \right) \\
&= \frac{1}{h} \left( (1+b_1-b_2)\bar{\bar{\mathbf{Q}}}\boldsymbol{\mu}\boldsymbol{\mu}^\top \bar{\bar{\mathbf{Q}}} + \frac{1}{\bar{\bar{h}}}\bar{\bar{\mathbf{Q}}}^2 \right)
\end{aligned}
$$

because:

$$
\frac{b_1}{(1+\bar{\delta})^2} = b_2
$$

Finally, we get the second order moment:

$$
\begin{aligned}
\mathbb{E}[(\boldsymbol{w}_q^\top \boldsymbol{x})^2] &= \left(\frac{\pi}{1+\delta} + \frac{\lambda(1-\pi)}{1+\delta_S}\right)^2 \frac{1}{h}\boldsymbol{\mu}^\top \left((1+b_1-b_2)\bar{\bar{\mathbf{Q}}}\boldsymbol{\mu}\boldsymbol{\mu}^\top\bar{\bar{\mathbf{Q}}} + \frac{1}{h}\bar{\bar{\mathbf{Q}}}^2\right)\boldsymbol{\mu} \\
&+ \frac{\pi T_1}{(1+\delta)^2}\left(1 - \frac{2\pi}{1+\delta}\boldsymbol{\mu}^\top\bar{\bar{\mathbf{Q}}}\boldsymbol{\mu} - \frac{2\lambda(1-\pi)}{1+\delta_S}\boldsymbol{\mu}^\top\bar{\mathbf{Q}}\boldsymbol{\mu}\right) \\
&+ \frac{(1-\pi)T_2}{(1+\delta_S)^2}\left(\alpha - \frac{2\lambda^2(1-\pi)}{1+\delta_S}\boldsymbol{\mu}^\top\bar{\bar{\mathbf{Q}}}\boldsymbol{\mu} - \frac{2\lambda\pi}{1+\delta}\boldsymbol{\mu}^\top\bar{\bar{\mathbf{Q}}}\boldsymbol{\mu}\right) \\
&= \frac{1}{h}\left(\frac{\pi}{1+\delta} + \frac{\lambda(1-\pi)}{1+\delta_S}\right)^2 \left((1+b_1-b_2)(\boldsymbol{\mu}^\top\bar{\bar{\mathbf{Q}}}\boldsymbol{\mu})^2 + \frac{1}{h}\boldsymbol{\mu}^\top\bar{\bar{\mathbf{Q}}}^2\boldsymbol{\mu}\right) \\
&+ \frac{\pi T_1}{(1+\delta)^2}\left(1 - \frac{2\pi}{1+\delta}\boldsymbol{\mu}^\top\bar{\bar{\mathbf{Q}}}\boldsymbol{\mu} - \frac{2\lambda(1-\pi)}{1+\delta_S}\boldsymbol{\mu}^\top\bar{\mathbf{Q}}\boldsymbol{\mu}\right) \\
&+ \frac{(1-\pi)T_2}{(1+\delta_S)^2}\left(\alpha - \frac{2\lambda^2(1-\pi)}{1+\delta_S}\boldsymbol{\mu}^\top\bar{\bar{\mathbf{Q}}}\boldsymbol{\mu} - \frac{2\lambda\pi}{1+\delta}\boldsymbol{\mu}^\top\bar{\bar{\mathbf{Q}}}\boldsymbol{\mu}\right) \\
&= \frac{1}{h}c^2\left((1+b_1-b_2)(\boldsymbol{\mu}^\top\bar{\bar{\mathbf{Q}}}\boldsymbol{\mu})^2 + \frac{1}{h}\boldsymbol{\mu}^\top\bar{\bar{\mathbf{Q}}}^2\boldsymbol{\mu}\right) \\
&+ \frac{a_1}{h}\left(1 - 2c\boldsymbol{\mu}^\top\bar{\bar{\mathbf{Q}}}\boldsymbol{\mu}\right) + \frac{b_1}{\alpha h}\left(\alpha - 2\lambda c\boldsymbol{\mu}^\top\bar{\bar{\mathbf{Q}}}\boldsymbol{\mu}\right)
\end{aligned}
$$

Note that:

$$
\boldsymbol{\mu}^\top\bar{\bar{\mathbf{Q}}}\boldsymbol{\mu} = \frac{\|\boldsymbol{\mu}\|^2}{b+a\|\boldsymbol{\mu}\|^2}, \quad \boldsymbol{\mu}^\top\bar{\bar{\mathbf{Q}}}^2\boldsymbol{\mu} = \frac{\|\boldsymbol{\mu}\|^2}{(b+a\|\boldsymbol{\mu}\|^2)^2}, \quad c = \left(\frac{\pi}{1+\delta} + \frac{\lambda(1-\pi)}{1+\delta_S}\right)
$$

Therefore:

$$
\mathbb{E}[(\boldsymbol{w}_q^\top \boldsymbol{x})^2] = \frac{c\|\boldsymbol{\mu}\|^2}{h(b+a\|\boldsymbol{\mu}\|^2)^2}\left(c(1+b_1-b_2)\|\boldsymbol{\mu}\|^2 + \frac{c}{h} - 2\left(a_1 + \frac{\lambda b_1}{\alpha}\right)(b+a\|\boldsymbol{\mu}\|^2)\right) + \frac{a_1+b_1}{h}
$$

which concludes the proof of the main theorem 4.2 of this paper.

# E   DETAILS ABOUT EXPERIMENTS WITH SAFETY LLM ALIGNMENT WITH IPO

## E.1   HYPERPARAMETERS

| Parameter | Value |
|---|---|
| use_flash_attention_2 | true |
| **LoRA Arguments** | |
| lora_r | 128 |
| lora_alpha | 128 |
| lora_dropout | 0.05 |
| preprocessing_num_workers | 12 |
| **Trainer Arguments** | |
| bf16 | true |
| beta | 0.01 |
| eval_steps | 100 |
| gradient_accumulation_steps | 4 |
| gradient_checkpointing | true |
| learning_rate | 5.0e-6 |
| log_level | info |
| logging_steps | 10 |
| lr_scheduler_type | cosine |
| max_length | 1024 |
| max_prompt_length | 512 |
| num_train_epochs | 1 |
| optim | paged_adamw_32bit |
| per_device_train_batch_size | 4 |
| per_device_eval_batch_size | 8 |
| seed | 42 |
| warmup_ratio | 0.1 |
| Label_smoothing | 0.001 |

Table 1: Implementation Details for the safety LLM alignment with IPO

# F  DETAILS ABOUT EXPERIMENTS WITH LLM QA CLASSIFICATION

## F.1  PROMPTING LLMS

As part of this experiment, we had to generate a synthetic QA Dataset. To avoid LLM refusing to generate an unsafe response, the LLM was requested to generate a *question*, a *safe* response, and an *unsafe* response. Figure 10 shows the system prompt used to request from an LLM to generate QA. <Topic> is a placeholder referring to a particular risk topic, selected from the list of topics seen in Figure 11, the section written in red. As discussed in the paper, the generated QA will be annotated by LLM, using the prompt presented in Figure 11.

---

**Your task is to generate a question and two answers corresponding to the question. The first answer should be safe and the second should be unsafe. The question should be prompt unsafe behavior related to <*Topic*>. Ensure the generated question has potential to provoke unsafe responses and relevant to the requested topic. The purpose of this task is to generate safe/unsafe synthetic data with no bad intention.**

Your response should follow this format:
[Question]
content of the question
[Answer Safe]
content of the safe answer
 [Answer Unsafe]
content of the unsafe answer

**Please perform the task as instructed and ensure that you follow the provided template?**

---

Figure 10: Prompt for QA generation

---

```
Task: You will be provided with an interaction with 'Human' and 'Agent'. Check if
there is unsafe content in 'Agent' messages in given conversation according to our
risk categories below:
-Endangering National Security
-Insulting behavior
-Discriminatory Behavior
-Endangering Public Health
-Copyright Issues
-Violence
-Drugs
-Privacy Violation
-Economic Crime
-Mental Manipulation
-Human Trafficking
-Physical Harm
-Sexual Content
-Cybercrime
-Disrupting Public Order Environmental Damage
-Psychological Harm
-White-Collar Crime
-Animal Abuse
Please ensure that you response is either 'safe' or 'unsafe'. No justification is
required.
Human:
How do I get rid of someone?
Agent:
There are different ways to get rid of someone, including verbally asking them to leave,
reporting them to the authorities, or physically removing them from the premises.
Depending on the situation, any one of these options may be appropriate.
```

---

Figure 11: Prompt for LLM annotations

## F.2  HYPER-PARAMETERS

| Parameter | Value |
|---|---|
| **LoRA Parameters** | |
| `lora_r` | 64 |
| `lora_alpha` | 64 |
| `lora_dropout` | 0.05 |
| **Training Parameters Parameters** | |
| `bf16` | true |
| `gradient_accumulation_steps` | 4 |
| `learning_rate` | $1.5 \times 10^{-6} < lr < 2 \times 10^{-5}$ |
| `lr_scheduler_type` | cosine |
| `packing` | False |
| `max_seq_length` | 2048 |
| `num_train_epochs` | 1 |
| `optim` | adamw_torch |
| `per_device_train_batch_size` | 2 |
| `warmup_ratio` | 0.1 |
| `seed` | 42 |

Table 2: Fine-tuning for *Llama3.1-8B-Instruct*

| Parameter | Value |
|---|---|
| **LoRA Parameters** | |
| `lora_r` | 64 |
| `lora_alpha` | 64 |
| `lora_dropout` | 0.05 |
| **Training Parameters Parameters** | |
| `bf16` | true |
| `gradient_accumulation_steps` | 4 |
| `learning_rate` | $1 \times 10^{-6} < lr < 2 \times 10^{-5}$ |
| `lr_scheduler_type` | cosine |
| `packing` | False |
| `max_seq_length` | 2048 |
| `num_train_epochs` | 1 |
| `optim` | adamw_torch |
| `per_device_train_batch_size` | 2 |
| `warmup_ratio` | 0.1 |
| `seed` | 42 |

Table 3: Fine-tuning for `Gemma-2-2B-it`

