# OpenReview forum: "Maximizing the Potential of Synthetic Data: Insights from Random Matrix Theory"
_ICLR.cc/2025/Conference — ICLR 2025 Poster_

### Official Review · Reviewer_qhaB · 2024-11-02

**Soundness:** 3
**Presentation:** 2
**Contribution:** 2
**Rating:** 5
**Confidence:** 4

**Summary:**

This paper builds upon Feng et al. that examined how using synthetic data with selection impacts final performance. Here, the authors extend that research to a setting that considers both distribution shifts in the feature space and the combination of real and synthetic data. This approach provides precise results without requiring an infinite number of synthetic samples. The experiments corroborate some theoretical insights.

**Strengths:**

1. The theoretical section is well-written, offering strong motivations and clear explanations.
2. Analyzing distribution shifts in the feature space is a valuable contribution compared with the previous work.
3. The paper provides a precise analysis of performance when training on mixed real and (selected) synthetic data with random matrix theory, expanding beyond previous work.

**Weaknesses:**

My primary concern is the applicability of the theoretical insights in practical settings. The theoretical predictions are based on binary Gaussian mixture data, with synthetic data generated by fitting another Gaussian mixture. While the authors provide performance predictions for models trained on mixed data, it is unclear how well these predictions, derived from Gaussian mixture models, will generalize to real-world data. Specifically, how can we estimate some of the key theoretical variables? How can we ensure that the supervision quality is sufficient to mitigate the negative effects of distribution shift?

Currently, the experimental section could be improved by drawing more explicit connections to the theoretical insights. It is generally expected that better supervision and a more accurate generator would lead to improved performance with selected data. However, what practical insights does the theory provide? Can the theoretical findings be leveraged to inform the design of good data selection methods?

The experiment part is not well-written, with lots of details missing. I will elaborate more in the questions.

**Questions:**

1. In Equation (2), the generative model assumes a prior with symmetric data sharing the same $\hat{\mu}$. How would the theory change if one Gaussian per class was fit instead?
2. On line 129, label noise is introduced as purely random noise. What if the label error depended on the input? For instance, in safety applications, a generative model might introduce label errors in specific regions of the distribution due to bias. Could such input-dependent noise be analyzed?
3. It would be helpful to discuss how label noise and feature noise correspond to various types of errors and shifts in real settings.

Missing details from the experiment:

4. What value of $\hat{n}$ is used across all experiments? The authors mention $\hat{n}$ varies in the MNIST experiment (line 400) but don’t provide further details.
5. In the LLM Safety experiment (line 411), it is stated, "we focus only on label noise." However, since data is generated by another language model, feature noise is likely present.
6. In Figure 7 (left), both feature noise and label noise should be included, as data are generated by a Gaussian fitted to the distribution. Could the authors provide error bars here? Given the simplicity of the MNIST setup with two-layer networks, this should be feasible.
7. In Figure 8, why does using $(\rho, \phi) = (0.5, 0.5)$ as weak supervision? According to the definition in Equation (5), the data selection remains random regardless of label accuracy, which seems equivalent to no supervision.
8. In the LLM Q&A Safety Generation experiment, is accuracy evaluated by the Llama-Guard-3.1 model? What weak supervision method is employed here? Why do the authors use two models to annotate labels? It would be helpful to include the evaluation results comparing the two labeling models with the “ground truth” from Llama-Guard-3.1 to assess the quality of synthetic labels. Additionally, how is supervised fine-tuning conducted with data containing both safe and unsafe answers?

---

> ### Author Response · Authors · 2024-11-20
> **Responses to Reviewer qhaB**
>
> We thank the reviewer for their thoughtful feedback.
>
> **Limitations of the Data Model:** Our data model, based on Gaussian mixtures with symmetric clusters, is a simplification chosen for analytical tractability. It allows us to derive explicit theoretical results using random matrix theory, capturing key aspects of high-dimensional data behavior, such as covariance estimation inconsistency and its impact on distribution shifts. These insights provide a foundation for future extensions to more complex distributions.
>
> **Fitting One Gaussian per Class:** Fitting one Gaussian per class with different means would introduce additional complexity. While possible, it would require careful handling of class-specific statistics and might lead to more intricate expressions in the theoretical results. We consider this an interesting direction for future work to better reflect scenarios with asymmetric class distributions.
>
> **Input-Dependent Label Noise:** Modeling label noise that depends on the input introduces significant analytical challenges and may be intractable within our framework. We focus on input-independent noise for tractability, providing foundational insights that can inform more complex models in future studies. Empirical studies of input-dependent noise are valuable but beyond the scope of our theoretical analysis.
>
> **Correspondence of Label and Feature Noise to Real-World Errors:** Label noise can arise from misannotations or errors in the labeling process, while feature noise can result from measurement errors or model approximations in synthetic data generation. Our model captures these through parameters $\varepsilon$ (label noise) and $\hat \eta$ (feature noise due to covariance estimation), helping understand their impact on performance.
>
> **Values of $\hat n$ in MNIST Experiments:**
>
> - **Best Stats Estimation:** $\hat n = 50{,}000$ (all training data).
> - **Weak Stats Estimation:** $\hat n = 1{,}000$.
> - **Very Weak Stats Estimation:** $\hat n = 100$.
> - **No Covariance Estimation:** Covariance set to identity.
>
> We will include these details in the revised paper.
>
> **LLM Safety Experiment and Presence of Feature Noise:** While we focus on label noise, feature noise is inherently present because the synthetic data are generated by another language model. This reflects real-world scenarios where both types of noise affect performance.
>
> **Error Bars in Figure 7:** Including error bars would enhance reliability. However, due to computational constraints, we prioritized adding more experiments, such as those involving LLMs. In future work, we aim to include error bars where feasible to improve the robustness of our experimental results.
>
> **Interpretation of $(\rho, \phi) = (0.5, 0.5)$ as Weak Supervision:** With these values, there is a 50% chance of selecting any synthetic data point, reducing the amount of noisy data included compared to "No supervision" with $(\rho, \phi) = (1, 1)$. This mitigates the negative impact of incorrect labels, which is why we consider it weak supervision.
>
> **LLM Q&A Safety Generation Experiment Details:** The accuracy is evaluated using Llama-Guard-3.1. We used two models to annotate labels to introduce variability and potential label noise, simulating realistic scenarios. Including a comparison of labeling models with ground truth is a valuable suggestion for future work. During supervised fine-tuning, the model is trained on data containing both safe and unsafe answers, learning to produce safe responses when given new prompts.

---

> ### Comment · Reviewer_qhaB · 2024-11-22
> **Further Questions**
>
> The reviewer thanks the author for the response.
>
> 1. **Insights for Practitioners**. My primary concern remains: could the author summarize the insights from this theoretical paper that could directly benefit practitioners? While I appreciate the theoretical contributions of this paper and understand that some results are derived under Gaussian data assumptions—which may approximate real high-dimensional data in certain cases—I feel the connection between theory and practical scenarios could be developed further. For example, the author states, "Our findings identify conditions where synthetic data could improve performance, focusing on the quality of the generative model and verification strategy." Could some of these conditions be verified in practice? Mixing real and synthetic data might lead to a U-shaped performance curve due to distribution shift. Could we predict whether such a U-shape would occur? Furthermore, with the random matrix analysis, what new insights are obtained compared to prior work by Feng et al. (2024)?
>
> 2. **Input-Dependent Label Noise**. From my understanding, the theoretical framework in Feng et al. (2024) encompasses input-dependent errors, whereas the current setting considers only input-independent noise. While the author provides stronger theoretical results, this seems to narrow the scope compared to previous work. As the author notes, "Label noise can arise from misannotations or errors in the labeling process", if the synthetic data are annotated using the same pipeline as the real data, the noise levels should be similar. However, if synthetic labels are generated by another model, input-dependent errors would arise. In the experiments, the authors write, "We used two models to annotate labels to introduce variability and potential label noise," implying that the introduced noise should also be input-dependent. This limitation should be explicitly discussed.
>
> 3. **Figure 7 Clarifications**. In Figure 7 (left), both feature noise and label noise are included. However, the subtitles of the two plots are somewhat misleading. The right plot is labeled "Feature Noise," while the left plot is labeled "Label noise $\epsilon=0.3$." This gives the impression that there is no feature noise in the left plot, which is incorrect. I suggest the author revise the subtitles for clarity.
>
> 4. **Interpretation of $(\rho, \phi) = (0.5, 0.5)$ as Weak Supervision**. Could the author clarify the meaning of the proportion of synthetic data, which is used as the x-axis in all figures? Is it the proportion of synthetic data in the entire training set before verification and selection? The paper assumes knowledge of which data are synthetic and which are real and the verification apply only to the synthetic portion then. If so, different lines with varying verification methods but the same proportion of synthetic data involve different total amounts of data being used for training. Why not control for the total number of training samples in each case? Initially, I assumed that the proportion of synthetic data referred to the training set after verification. Under that assumption, $(\rho, \phi) = (0.5, 0.5)$ would correspond to no supervision, i.e., $(\rho, \phi) = (1, 1)$. Even if verification applies only to synthetic data, the "weak supervision" here is actually randomly using half the synthetic data. Why is this termed supervision? Isn't it simply a way of altering the ratio between real and synthetic data, given the assumed knowledge of data sources?
>
> 5. In the caption of Figure 3, the authors write "The parameter $\epsilon$ is variable depending on the proportion of synthetic data by taking it equal to the misclassification error corresponding to training a classifier on synthetic data only". What does it mean?

---

> > ### Author Response · Authors · 2024-11-23
> > **Responses to new questions of reviewer qhaB**
> >
> > We thank the reviewer for their rigorous remarks and engagement with our work. Here we would like to tackle their concerns:
> >
> > **1- Insights for Practitioners:** Overall, in our paper, we highlighted the impact of a type of noise that was not considered in previous related work, which is feature noise. We believe that synthetic data does not necessarily have the same distribution as the real one, and this shift in the distributions, in addition to label noise, is what determines the quality of this synthetic data (the greater is this shift, the lower is the quality of synthetic data). In addition to our theoretical setting that describes how this shift is for Gaussian Mixture Models, our experiments in Figures 7 and 9 clearly show the impact of this type of noise on the performance of the trained models. Therefore, we shed light on taking this feature noise into account to mitigate the negative effects of synthetic data. But unfortunately, it is generally a very hard task to measure this shift when it comes to most real world
> > scenarios, as the true distribution of data is unknown and hard to determine, and this goes beyond the scope of our theoretical paper.
> >
> > **2- Input-dependent Label Noise:** Indeed, we agree with the reviewer that we assumed an input-independent label noise in our theoretical study, because we considered a label noise coming from a different ”labelling” model, that is not necessarily the one used for training, and this should guarantee a sort of independence between features and the labels. And in the Q&A experiments, we used labelling models (Mistral-Nemo and Qwen2-7B-Instruct) that are different from the ones that were finetuned and used to generate the synthetic prompts  (Llama3.1-8B-Instruct and Gemma-2B-it), and thus label noise in this experiment should be independent from the input, as it is determined by the performance of the labelling models.
> > However, we strongly agree with the reviewer that considering an input-dependent label noise is also relevant to most real case scenarios, and we are happy to consider it in a future work.
> >
> > **3- Figure 7 clarifications:** We thank the reviewer for pointing out this remark. Indeed, the synthetic data used in the experiments shown in the left plot of Figure 7 certainly contain feature noise, and the reason why we didn’t mention it is because this noise is minor in this setting as we do a good estimation of the statistics of the real data. However, we agree that this can be missleading to the readers, so we updated the titles of the plots in the new manuscript (see edited manuscript).
> >
> > **4- Interpretation of $(\rho, \phi ) = (0.5, 0.5)$ as Weak supervision:** We thank the reviewer for this interesting remark. We want to point out that the quantity $1 - \pi$ denotes the proportion of synthetic data used **before verification**. This can justify why we considered  $(\rho, \phi) =  (0.5, 0.5)$ to be a Weak supervision scheme, and the No supervision only corresponds in our case  to $(\rho, \phi) =  (1, 1)$. However, if the reviewer recommends it, we are happy to run another experiment with different values of $(\rho, \phi)$ (for instance $(0.6, 0.8)$) and report the results in a new version of the manuscript.
> >
> > **5- Figure 3 clarifications:** We thank the reviewer for pointing out this detail. Indeed, in that experiment, instead of using a fixed noise ratio $\varepsilon$ for every proportion $1 - \pi$ of synthetic data, we wanted to imitate what can be found in real case synthetic data generations. For that, we considered a variable noise ratio $\varepsilon(m)$ that depends on the number of synthetic samples, and that is equal to the missclassification error of a linear classifier trained solely on this synthetic data of
> > size $m$ (so that the label noise here would correspond to the error of a classifier trained on this data). The reviewer should know that this idea of taking a variable noise rate is just a choice of us, and we can certainly conduct the same experiment with a fixed $\varepsilon$ (which is easier actually) if the reviewer believes that it is more relevant (which is easier actually).
> >
> > We hope that we have adequately addressed all the reviewer's questions, and we remain open to any further clarifications.

---

> ### Comment · Reviewer_qhaB · 2024-11-28
>
> The reviewer thanks the authors for their further responses and clarifications. I still have some follow-up questions:
>
> 1. **Insights for Practitioners**: I agree that including feature shift in analyzing synthetic data is important; however, I would like to see more actionable insights beyond simply stating that a distributional shift exists and affects performance. Since the authors provide a rigorous characterization, could some insights be included on how supervision may influence this shift and thereby affect performance?
>
> Additionally, I do not fully agree with the statement: "the greater is this shift, the lower is the quality of synthetic data." Why should a shift directly correlate with lower quality? There could be beneficial distributional shifts. For example, in mathematical reasoning tasks, if the real distribution of problems spans different difficulty levels, it might be advantageous to augment more intermediate-level problems to help the model learn to solve harder problems. I hope this analogy can help the authors gain deeper insights into the distributional shifts and their potential impacts.
>
> 2. **Input-Dependent Label Noise**: In the Q&A experiments, why should the label noise in this experiment be independent of the input? Label noise is determined by the performance of the labeling models, which would naturally vary based on the input characteristics.
>
> 3. **Interpretation of $(\phi, \psi) = (0.5, 0.5)$ as Weak Supervision**: My concern here is that $(\phi, \psi) = (0.5, 0.5)$ does not actually involve supervision. It simply represents taking a random half of the synthetic data, as you already know which data is synthetic. To truly qualify as weak supervision, a setting such as $(\phi, \psi) = (0.6, 0.8)$ would involve some level of actual supervision.

---

> > ### Author Response · Authors · 2024-11-29
> > **Response by Authors**
> >
> > We thank the reviewer for their additional remarks and engagement with our work. Here we would like to further tackle their concerns:
> >
> > **1- Insights for Practitioners:** We thank the reviewer for their constructive remark. Indeed, using our main Theorem 4.2, we can analyze how the distribution shift and the label noise rate $\varepsilon$ affect the model's performance through the theoretical test accuracy expression $\Phi \left( (\nu - m^2)^{-\frac12} m \right)$ (as stated at the end of the theorem). For instance, this allows us to determine, for a given $\varepsilon$ and supervision parameters $(\rho, \phi)$, the optimal (or worst) proportion of synthetic data that maximizes (or minimizes) the theoretical test accuracy. Our analysis provides a framework to assess the influence of any parameter in the model on its final performance. However, we should note that our analysis is currently limited to the binary linear classification model. Extending these results to more complex models lies beyond the scope of this paper.
> >
> > **Why should a shift directly correlate with lower quality?**: We agree with the reviewer that some distribution shifts can indeed be beneficial in certain scenarios, such as the one they described. However, we would like to emphasize that the test data (unseen data) is assumed to follow the same distribution as the real training data, which is a fundamental assumption in most machine learning tasks. Therefore, in most cases, a significant distribution shift is likely to mislead the model from its original goal.
> > For example, consider training a binary classifier to distinguish between images of cats and dogs. Augmenting the dog's class with images of wolves may not mislead the model because wolves share many similarities with dogs, resulting in a small distribution shift. However, if we instead augment the dog's class with images of rabbits, this would likely mislead the classifier, as the distribution shift between dogs and rabbits is much greater than that between dogs and wolves.
> >
> > We hope this example clarifies our point about scenarios where a great distribution shift misleads the model (also observed in Figure 7 (right) in our manuscript).
> >
> > **2- Input-dependent Label Noise:** We thank the reviewer for their interesting remark. While we agree that feature noise may impact the labeling model, our analysis remains valid under this additional assumption. In fact, our framework accommodates any value of $\varepsilon$ (and, therefore, any labeling model). By interpreting $\varepsilon$ as the misclassification error of a labeling model affected by distribution shift, this scenario becomes a specific case of our general analysis. Consequently, our current setting is sufficiently general to handle such situations.
> >
> > **3- Interpretation of $(\rho, \phi) = (0.5, 0.5)$ as Weak Supervision:** We totally agree with the reviewer in this point and that the values of $(\rho, \phi) = (0.5, 0.5)$ do not fully reflect a weak supervision scenario, hence we will run another experiment with other values and report the results in the final manuscript.
> >
> > We hope that we have adequately addressed all the reviewer's questions, and we remain open to any further clarifications.

---

### Official Review · Reviewer_BEs2 · 2024-11-03

**Soundness:** 4
**Presentation:** 4
**Contribution:** 3
**Rating:** 6
**Confidence:** 5

**Summary:**

This paper extends important work by Feng et al. 24 and studies the effect of synthetesized data as part of the training data of models. It has two parts, a theoretical part that significantly extends the analysis of a mixture of two isotropic Gaussians from Feng et al. 24 from the case of infinite data at finite dimension to the case where both dimension and data tend to infinity with fixed ratio. Using tools from RMT they thus uncover a richer structure of transitions in a parameter space defined by the quality of synthetic data, the generator, label noise and verifier strength. A key novel element (which facilitates the analysis) is the assumption that the synthetic data comes from a model that estimates the mean and covariance matrix from initial (real) data (as opposed to a more general estimator). This allows to nicely study distribution shift of the features (not just the labels). Another extension over Feng et al.24 is the analysis of mixtures of real and synthetic data (though this is also done in "Strong Model Collapse" by Dohmatob et al., in a different and more general setting). 4 experiments nicely support the theory.

**Strengths:**

The paper lays out a coherent program and goes ahead to prove it with relatively sophisticated tools. The message is conveyed very clearly, the extensions over prior work, and in particular Feng et al 24 are clearly demarcated.
The consequences of theorems are explained well, despite the necessarily heavy math. This paper thus provides further evidence to what was laid out in Feng et al 2024: synthetic data can only boost performance when coupled with verification (pruning).
The paper does a masterful job of presenting the results and the experiments are a nice addition. The paper is also well written and is fun to read.

**Weaknesses:**

The main weakness is that a few crucial references are missing. In particular there are three papers mixing real and synthetic data that should be cited, [1] and [2], and [3], a more recent one (there is no fault in not citing the latter, as it is recent,
but the former date back over 8 months). In particular, a careful brief comparison to the [1] setting and results should be added, and [2] should be mentioned. Another paper that should be cited among the theoretical works for model collapse is [4]. On the other hand,
your reference to [5] should probably be completely removed as it is misleading (the analysis of combination of real and synthetic data doesn't compare to adding the same amount of real data properly - this has been debunked on several occasions and should not be propagated).

Specifics (mostly minor):
1. line 252: typo "usefull"
2. line 402: "emphasize on" - sounds wrong, reformulate
3. line 406 use \citep for Malartic
4. line 408: "from the the Antropic's" --> "from Antropic's"
5. line 483-484: discuss Llama versus Gemma to make clear that you regard one as a weaker supervisor than the other
6. line 518: "Conclusionn" - typo
7. line 527-28: you might want to discuss that label pruning can induce a distribution shift in the features (because the pruning could be biased towards keeping certain features alive) - see (and cite) the discussion in Feng et al. 2024 (https://arxiv.org/pdf/2406.07515 version bottom of p. 10 - top p 11)

I will be happy to reconsider my score once these have been addressed (and the questions).

[1] Ayush Jain, Andrea Montanari, Eren Sasoglu: Scaling laws for learning with real and surrogate data, https://arxiv.org/abs/2402.04376

[2] Bertrand, Q., Bose, A. J., Duplessis, A., Jiralerspong, M., and Gidel, G. On the stability of iterative retraining of generative models on their own data, ICLR 2024 spotlight

[3] Elvis Dohmatob, Yunzhen Feng, Arjun Subramonian, Julia Kempe: Strong Model Collapse, https://arxiv.org/abs/2410.04840

[4] Elvis Dohmatob, Yunzhen Feng, Pu Yang, Francois Charton, Julia Kempe: A Tale of Tails: Model Collapse as a Change of Scaling Laws, ICML 2024

[5] Gerstgrasser et al. 2024

**Questions:**

1) How does your work compare to Jain et al. [1] specifically? (It’s a different data model, of course, but qualitatively, are there similarities/differences and how do the methods differ?)
2) I am confused about the "No supervision" case in Fig. 6 (the Amazon data). While in all other experiments more synthetic data doesn't help, here the U-shape of the curve stands out. Can you discuss a little more why in this case the red curve starts going up?

---

> ### Author Response · Authors · 2024-11-20
> **Responses to Reviewer BEs2**
>
> We thank the reviewer for their feedback and for highlighting areas for improvement.
>
> **Comparison with Jain et al. (2024):** Our work differs primarily in the statistical modeling of synthetic data. Jain et al. (2024) model differences between real and synthetic data through mean shifts. We consider both mean and covariance estimation errors, acknowledging that covariance estimation is inconsistent in high-dimensional settings. We also include label noise in our analysis, which is not considered in their work. This allows us to study the combined effect of feature and label inaccuracies on model performance.
>
> **U-Shaped Curve in Figure 6:** The performance increase when the proportion of synthetic data exceeds 0.8 under "No supervision" is due to the large amount of synthetic data starting to outweigh the negative impact of label noise. With label noise $\varepsilon = 0.2$, 80% of the synthetic labels are correct. As more synthetic data is added, the correct labels provide sufficient signal for the model to improve. This effect diminishes when label noise is higher, as incorrect labels have a more detrimental impact.
>
> **Typos and Literature Inclusion:** We will carefully revise the paper to correct any typos and ensure all relevant literature, including the papers you mentioned, is appropriately cited and discussed.

---

> > ### Comment · Reviewer_BEs2 · 2024-11-21
> > **Thank you for your response**
> >
> > Thank you for these clarifications and the intent to adapt the manuscript (although it would have been nice to see the new version with changes marked).
> >
> > As for Jain et al. - would you also be able to comment on how your methods and techniques differ from theirs?
> >
> > I would also like to hear more in response to reviewer KXSN as the questions raised there are also of interest to me.
> >
> > Thank you!

---

> > > ### Author Response · Authors · 2024-11-23
> > > **Additional response and updates for reviewer BEs2**
> > >
> > > We thank the reviewer for their constructive guidelines, and we want to mention that we have updated our manuscript according to the reviewer's feedback.
> > >
> > > Here we want to add more details about their concerns:
> > >
> > > **Another difference with Jain el al.**: In addition to the differences in the theoretical model discussed earlier, a major distinction between our work and theirs lies in the experimental setup. In our experiments, we use a generation scheme to get the synthetic data for most of our tasks, contrary to their work where they take their synthetic data from external resources. For example we generated synthetic MNIST-like samples (see Figure 5 for some examples) to make the experiments described in Figure 7. On the other hand, Jain et al. used CIFAR-100 samples as surrogate data to train a classifier intended for CIFAR-10, rather than generating samples based on CIFAR-10.
> > >
> > > We hope that we have tackled all the reviewer's remarks, and remain open for any further clarification.

---

> > > > ### Comment · Reviewer_BEs2 · 2024-11-25
> > > > **Response to response**
> > > >
> > > > I thank the authors for their comments and the new manuscript. I see the typos corrected (marked in blue) but I don't see any other marked changes?
> > > >
> > > > My question on methods and techniques in Jain et al 24 referred to mathematical techniques, not the experiments.
> > > >
> > > > In any case, I maintain my score, though for the benefit of the AC I point out that this would be a 6.5 if such a score existed.

---

> ### Author Response · Authors · 2024-11-29
> **Response by Authors**
>
> We thank the reviewer for their engagement and support to our work, and for all the guidelines to improve our manuscript.
>
> We also want to point out that, in addition to correcting the typos (marked in blue), we also included the additional references in the related work section of the updated manuscript.
>
> **In the differences with Jain et al.:** We apologize for our misunderstanding of the reviewer’s question. Beyond the differences in synthetic data modeling (as described in our initial rebuttal) and the types of experiments (outlined above), we would like to highlight another key distinction from Jain et al. (2024): our analysis relies on Random Matrix Theory, which provides a deterministic theoretical performance of the model (Theorem 4.2). This performance depends solely on the characteristics of the training data (such as the mean and covariance in Gaussian Mixture Models) rather than the random nature of the data itself, as in Jain et al. (2024). Additionally, our single analysis holds for both low and high-dimensional regimes, whereas in Jain et al. (2024), they provide a different analysis for each regime, making our method easier to exploit and more generalisable.
>
> We hope our response was satisfactory to the reviewer and remain available for any further clarification.

---

### Official Review · Reviewer_m5rf · 2024-11-03

**Soundness:** 3
**Presentation:** 2
**Contribution:** 2
**Rating:** 3
**Confidence:** 3

**Summary:**

Synthetic data has been used in training large language models, but concerns about its quality have emerged in practical applications. This paper works on data pruning to select high-quality data and analyzing the performance of trained on a mixture of real and synthetic data in a high-dimensional setting. The authors provide both theoretical and experimental results for the benefits of using and verifying synthetic data.

**Strengths:**

1. This work presents analysis and insights with theoretical justification.
2. Through multiple tasks, the approach is shown its empirical effectiveness.

**Weaknesses:**

1. The paper lacks a clear definition of the quality of synthetic data.
2. The derived theory in the paper is strongly assumed based on Gaussian distribution, however, non-Gaussian data are common in text or image data used for LLMs.
3. The paper only focuses on label verification for synthetic data. In real-world applications, synthetic data generation often struggles with feature fidelity, addressing feature consistency or the quality of synthetic data in feature space is more beneficial.

**Questions:**

1.Figure 1: $\hat{\eta}$ = 0.1, $p$ = 500, and $\hat{n}$ = 500, given that $\hat{\eta}$ = $\frac{p}{\hat{n}}$, is this a typo?
2. Figure 3, in the figure of weak supervision, when the proportion of synthetic data is higher than 0.8, the performance increases, what is the reason for this? This trend is opposite to that when the proportion of synthetic data is lower than 0.8.  Similar observations also happen in Figure 6.
3. Can authors explain why $\hat{\eta}$ can represent a distribution shift?

---

> ### Author Response · Authors · 2024-11-20
> **Responses to Reviewer m5rf**
>
> We thank the reviewer for their valuable feedback.
>
> **Definition of Synthetic Data Quality:** In our paper, the quality of synthetic data is characterized by:
>
> 1. **Sample Size $\hat n$ for Estimating Statistics:** The parameter $\hat n$ represents the number of real data samples used to estimate the mean and covariance in our generative model. Larger $\hat n$ leads to more accurate estimates and higher-quality synthetic data.
>
> 2. **Label Noise Rate $\varepsilon$:** Indicates the fraction of incorrect labels in the synthetic data. Lower $\varepsilon$ means fewer label errors, contributing to better data quality.
>
> By considering both feature quality (through covariance estimation) and label accuracy, our model captures key aspects affecting synthetic data utility.
>
> **Use of Gaussian Distributions:** While real-world data often deviate from Gaussianity, we chose Gaussian mixtures for analytical tractability, enabling explicit formulas and rigorous analysis. Our findings extend qualitatively to non-Gaussian settings, supported by related work showing that results hold under broader conditions like sub-Gaussian distributions or bounded moments. Experiments with MNIST and LLMs, which are non-Gaussian, validate the applicability of our results.
>
> **Focus on Label Verification and Feature Fidelity:** Our analysis accounts for feature noise by incorporating distribution shifts due to covariance estimation errors. While we focus on label verification, including feature noise in our model provides insights into how feature inconsistency affects performance. Developing methods for feature verification is a promising direction for future work.
>
> **Typographical Error in Figure 1:** We apologize for the error and confirm that the correct value is $\hat n = 5 \times 10^3$. We will correct this in the revised paper.
>
> **Performance Increase with High Synthetic Data Proportion:** In Figure 3, under weak supervision, the performance increase when synthetic data proportion exceeds 0.8 occurs because the large volume of synthetic data begins to compensate for label noise and distribution shifts. The correct labels (even with some noise) provide sufficient signal for the model to improve as more data is added. This effect is more pronounced when label noise $\varepsilon$ is not excessively high. When $\varepsilon$ is high, adding more noisy data continues to degrade performance, as shown in Figure 8.
>
> **Interpretation of $\hat \eta$ as Distribution Shift:** The parameter $\hat \eta$ represents the ratio $p / \hat n$, capturing the extent of distribution shift due to covariance estimation errors. A higher $\hat \eta$ indicates greater estimation error in the covariance matrix, leading to a more significant distribution shift between synthetic and real data. This affects model performance, highlighting the importance of accurate covariance estimation in generating high-quality synthetic data.

---

> > ### Author Response · Authors · 2024-11-25
> > **Comment by authors**
> >
> > Dear reviewer m5rf,
> >
> > Given that the discussion deadline is soon, we would appreciate it if the reviewer could provide feedback on our rebuttal, particularly noting if their concerns have not been addressed.

---

> > ### Comment · Reviewer_m5rf · 2024-12-01
> >
> > Thanks for the authors’ response.
> >
> > - I’m not fully convinced by the setup regarding the quality of synthetic data. The distribution shift doesn’t fully account for the difference between real and synthetic data in the feature space.  For example, different styles of dog images, such as photorealistic vs sketch, may have distinct distributions but including them in training could not degrade the in-distribution model performance but potentially increase the out-of-distribution performance. While the verification from the label is one thing,  I think more considerations from the feature space make more sense for the synthetic data in real-world scenarios.

---

> > > ### Author Response · Authors · 2024-12-01
> > > **Response**
> > >
> > > We thank the reviewer for their feedback. While we understand the point about verification in feature space, we want to emphasize that our work makes significant theoretical advances by explicitly modeling noise in the feature space - a crucial aspect that previous works have not addressed. Our framework provides rigorous theoretical analysis and demonstrates smooth phase transitions both analytically and empirically, substantially generalizing existing results in the literature.
> > >
> > > The primary goal of theoretical work is to develop tractable mathematical models that capture essential aspects of real-world phenomena while remaining analyzable. Our simplified setting allows us to prove fundamental results about the interplay between synthetic data quality and learning outcomes. While feature verification is certainly an important direction, it would require substantially more complex analysis that is out of the scope of this paper.
> > >
> > > We appreciate the constructive nature of the reviewer's comments, which suggest they see value in our contributions while proposing interesting future directions. Given this generally positive feedback focusing on potential extensions rather than fundamental flaws, we were quite surprised by the low score assigned to the paper. We believe our work makes substantial theoretical progress on an important problem, laying crucial groundwork for the more complex scenarios the reviewer describes.

---

### Official Review · Reviewer_KXSN · 2024-11-13

**Soundness:** 3
**Presentation:** 4
**Contribution:** 3
**Rating:** 8
**Confidence:** 4

**Summary:**

This work considers the problem of maximizing the usefulness of synthetic data in the modern era of generative modelling whereby at least part of the training data is not from the real data distribution, but synthesized from another AI model. In the solvable setting of a mixture of isotropic Gaussians, the authors consider a linear model whose weights vector is a regression-style ridge estimator. The synthetic data distribution is taken as yet another Gaussian mixture whose parameters (i.e shape) is given by empirical estimates of the ground-truth Gaussian mixture: the amount of data on which these parameters are estimated serves as a proxy for the quality of the synthetic data. Optiinally, this synthetic data is mixed with data from the real data distribution.

In the "proportionate" high-dimensional scaling limit, the authors use classical tools from random matrix theory (RMT) to obtain exact analytic expressions for the classifier and its accuracy. In particular, they paper recovers the theoretical results of Feng. et al (2024) as special case (corresponding to training only on synthetic data).

**Strengths:**

- A nice theoretical setup for analysing the impact of synthetic data and pruning strategies is introduced.
- A complete analytic theory for this toy setting is obtained and its phenomological clearly discussed.
- Theoretical results from previous work Feng et al. (2024) are recovered as special cases.
- The calculations provided in the appendix could be of independent usefulness, beyond the specific problem considered in the paper. I should however immediately nuance this praise by noting that similar calculations have been done in  Liao and Couillet (2019) "A Large Dimensional Analysis of Least Squares Support Vector Machines", for more general kernel-based models.
- The paper is very clearly written.

**Weaknesses:**

- **The setup.** Fitting a regression model to use its weights as a linear classifier, nobody does this in practice. The authors should justify why such a model is reasonable / relevant, beyond the fact that it leads to a tractable analysis. Note that the underlying estimator is a linear version of the so-called least-squares support vector machine (LS-SVM) which was analyzed in Liao and Couillet (2019) "A Large Dimensional Analysis of Least Squares Support Vector Machines".
- **(Ir)relevance to practice.** In the asymptotic regime considered in this paper, mixing real and synthetic data only helps when the proportion of real data is bounded-away from zero. Also, the practitional needs to know which samples are real and which samples are synthetic. These constraints might not match what happens in practice. This observations have also been made in Dohmatob et al. (2024) "Strong Model Collapse".
- **Non-optimal mixing.** The weights in mixing strategy considered is a bit naive / sub-optimal. The right thing to do would be to consider a general mixing constant $\alpha \in (0,1)$ (the papers uses $\alpha=1-\alpha=1/2$), and derive a theory as a function of $\alpha$, alongside the other  constants in the theory, namely $\pi$, etc.. In general the optimal value of $\alpha$ will depend on the other constants. See Jain et al. (2024) "Scaling laws for learning with real and surrogate data" and Dohmatob et al. (2024) "Strong Model Collapse"

**Questions:**

Though I think the paper is potentially a very good paper, I still have a few worries as outlined in the **Weaknesses** section of this review and also the questions listed below (I'm open to changing my mind if addressed).

- Clarify relevance of the consider model (see **Wicknesses** section above)
- Clarify connection to literature, especially Jain et al. (2024) "Scaling laws for learning with real and surrogate data" and Dohmatob et al. (2024) "Strong Model Collapse" (see **Wicknesses** section above)
- Can you instantiate your results on a concrete setup and show the corresponding scaling laws (perhaps with some approximations of Gaussian CDF) ?

---

> ### Author Response · Authors · 2024-11-20
> **Responses to Reviewer KXSN**
>
> We thank the reviewer for their constructive remarks.
>
> **Model Relevance and Use of $L^2$ Loss:** While using a regression model as a linear classifier is not standard in practice, we chose this setup for several reasons:
>
> 1. **Analytical Tractability:** The linear classifier with an $L^2$ loss allows us to derive closed-form solutions and perform rigorous theoretical analysis using random matrix theory. This helps us understand the fundamental effects of synthetic data, label noise, and distribution shifts in high-dimensional settings.
>
> 2. **Isolating Effects:** By focusing on a linear model, we can isolate and study the impact of synthetic data quality and verification strategies without the complexities introduced by non-linear models. This simplification aids in deriving insights applicable to more complex models.
>
> 3. **Practical Scenarios:** Linear models are still relevant, especially when used as classifiers on features extracted from neural networks, which is common in data-limited situations.
>
> Regarding the connection to Liao and Couillet (2019), our work extends their analysis by incorporating a mixture of real and synthetic data, providing a framework to study the effect of synthetic data on model performance. While the $L^2$ loss is less common in classification, it provides analytical tractability. Our experiments demonstrate that our findings hold qualitatively when using other loss functions like cross-entropy.
>
> **Practical Relevance of Assumptions:** Practitioners often know which data is real and synthetic, especially when augmenting datasets. Our analysis covers various proportions of real and synthetic data through the parameter $\pi$, allowing us to study scenarios where real data is limited. Although we cannot directly analyze the case with $\pi = 0$ (training only on synthetic data) in Theorem 4.2 due to undefined quantities, we address this separately in Subsection 4.2 (Corollary 4.3), generalizing previous results to include high-dimensional settings.
>
> **Mixing Strategy with Mixing Parameter $\alpha$:** We appreciate the suggestion to consider a general mixing constant $\alpha$. We highlight that this can be easily incorporated into our current analysis using the exact same derivations. We will provide in the final version a general result incorporating the mixing parameter $\alpha$ as suggested by the reviewer and investigate how the optimal value of $\alpha$ depends on other constants like $\pi$, $\varepsilon$, and the verification quality.
>
> **Connection to Literature:**
>
> - **Jain et al. (2024):** Our work differs by considering both mean and covariance estimation errors in the synthetic data, acknowledging that covariance estimation is inconsistent in high-dimensional settings. We also incorporate label noise, which is not considered in their analysis.
>
> - **Dohmatob et al. (2024):** We extend their study by including both label and feature noise, providing a more comprehensive analysis of the synthetic data's impact on model performance.
>
> **Scaling Laws and Concrete Setup:** Our experiments illustrate the theoretical findings, showing how key parameters like the ratio $p/n$, label noise $\varepsilon$, and verification quality $(\rho, \phi)$ affect model performance. We would appreciate it if the reviewer could elaborate more on their question “Can you instantiate your results on a concrete setup and show the corresponding scaling laws?”.

---

> ### Comment · Reviewer_KXSN · 2024-11-21
> **A Few Comments**
>
> Thanks for the response. I'm not satisfied with a couple of points made.
>
> >*Practitioners often know which data is real and synthetic, especially when augmenting datasets.*
>
> This is simply not true, and will get worse over time, as more synthetic data pollution occurs; it's not possibly to reliable tell which part of the training corpus of a LLM is real and which part is synthetic (can't tell which images on the internet are real and which ones are synthetic). I'm not referring here to synthetic data generated and controlled by the practitioner; I mean synthetic data which finds its way in the training corpus (e.g because data was scrapped from the Internet, etc.).
>
> >*Practical Scenarios: Linear models are still relevant, especially when used as classifiers on features extracted from neural networks, which is common in data-limited situations.*
>
> The question is not whether linear models are relevant or irrelevant. I'm fine with linear models for theoretical analysis of phenomena in relevant regimes. The issue here is that nobody in ML fits a linear model via a **regression loss**, to use it as a classifier at inference time. From a technical standpoint, I think you can get just as much "analytical tractability" by working with an actual classification loss (e.g logistic), and using classical tools around Gaussian comparison inequalities (Gordon, etc.). The justification for L^2 loss here seems superficial.

---

> > ### Author Response · Authors · 2024-11-23
> > **Responses to comments of reviewer KXSN**
> >
> > We thank the reviewer for their interesting comments and engagement with our work. Below, we would like to answer their concerns:
> >
> > 1- **Knowing which data is synthetic and which is real:** We totally agree with the reviewer that in cases where we do not generate "explicitly" and intentionally the synthetic data, then knowing which samples are synthetic is a hard problem, and this goes beyond the scope of our paper. In fact, our main goal was to model the impact of this synthetic data that incorporates both feature and label noise, and to the best of our knowledge, making the assumption of knowing which data is synthetic is broadly considered in most of the theoretical related works.
> >
> > 2- **The usage of regression loss:** We agree with the reviewer that the $L^2$ loss (or regression loss) is not commonly used for classification tasks as the cross-entropy is more intuitive and theoretically grounded. However, it is still legitimate and sometimes better to use $L^2$ loss rather than the Cross-Entropy as was demonstrated in Hui et al., ”Evaluation of Neural Architectures Trained with Square Loss VS Cross-Entropy in Classification Tasks”, ICLR 2021 (https://arxiv.org/pdf/2006.07322), which shows that a wide range of NN models perform comparably or better in classification when trained with the squared loss, even after equalizing computational resources. Thus, in addition to easier theoretical tractability, this cited paper might serve as a justification for our choice of the squared loss.
> >
> > We hope that we have tackled all the reviewer's concerns, and we remain open for any further clarification.

---

> > > ### Comment · Reviewer_KXSN · 2024-11-29
> > > **Final comment**
> > >
> > > Thanks for the response. I'm not 100% satisfied with the response of the authors to some of the points I raised, but i've raised my score from 6 to 8 because I thing the paper still makes a non-trivial contribution to an interesting problem!

---

> ### Author Response · Authors · 2024-11-29
> **Final comment by Authors**
>
> We thank the reviewer for their engagement and thoughtful feedback throughout the rebuttal process. We also sincerely appreciate the reviewer's interest and support for our work, as well as the increase in their score.

---

### Author Response · Authors · 2024-11-29
**General Comment**

We thank all the reviewers for their active engagement and interest in our work. We are also grateful for their thoughtful and constructive feedback, as well as their valuable suggestions to improve our manuscript.

In this rebuttal, we have carefully addressed all the reviewers' questions and comments, and we hope to have resolved all their concerns.

Additionally, we have provided an updated version of the manuscript, which includes references to additional related work, corrections of typos (highlighted in blue), and updated figure subtitles (Figure 7).

---

### Meta-Review · Area_Chair_VBGD · 2024-12-18

**Metareview:**

The submission considers the impact of using synthetic data (e.g., samples from a pre-trained generative model) to train a model, and how one must incorporate a verification/pruning mechanism. This is investigated first from a theoretical point of view using a Gaussian mixture model, and then experimentally with a more realistic data setup. The analysis is more general than previous work, and recovers the result of Feng et al. (2024) as a special case. Moreover, the empirical analysis corroborates and extends the claims made about Gaussian mixture analysis to the more practical real-world setting. Reviewers also note that the presentation is excellent and more approachable than one might expect given the sophistication of the tools employed.

The main concerns outlined by the reviewers were related to the Gaussian mixture model used in the theoretical analysis. However, I agree with the authors and some of the reviewers that this is not a massive limitation, given the experimental validation and the possibility that the results can be generalised to the sub-Gaussian case.

**Additional Comments On Reviewer Discussion:**

The criticisms of reviewer BEs2, that there were a number of typos and missing discussion about/citations to related work, have been addressed in a revised manuscript. Reviewer qhaB identified several missing pieces of information pertaining to the experimental validation, which have been included in the latest revision.

---

### Decision · Program_Chairs · 2025-01-22

Accept (Poster)